# Spectral Multiplicity Entails Sample-wise Multiple Descent

## Abstract

In this paper, we study the generalization risk of ridge and ridgeless linear regression. We assume that the data features follow a multivariate normal distribution and that the spectrum of the covariance matrix consists of a given set of eigenvalues of proportionally growing multiplicity. We characterize the limiting bias and variance when the dimension and the number of training samples tend to infinity proportionally. Exact formulae for the bias and variance are derived using the random matrix theory and convex Gaussian min-max theorem. Based on these formulae, we study the sample-wise multiple descent phenomenon of the generalization risk curve, i.e., with more data, the generalization risk can be non-monotone, and specifically, can increase and then decrease multiple times with more training data samples. We prove that sample-wise multiple descent occurs when the spectrum of the covariance matrix is highly ill-conditioned. We also present numerical results to confirm the values of the bias and variance predicted by our theory and illustrate the multiple descent of the generalization risk curve. Moreover, we theoretically show that the ridge estimator with optimal regularization can result in a monotone generalization risk curve and thereby eliminate multiple descent under some assumptions.

## 1 Introduction

The double/multiple descent phenomenon attracted recent research attention due to (Belkin et al., 2019). This line of work focuses on the parameter-wise double/multiple descent phenomenon of the risk curve (Bartlett et al., 2020; Tsigler & Bartlett, 2020; Belkin et al., 2019; 2020; Chen et al., 2020a; Liang et al., 2020; Advani et al., 2020; Bös & Opper, 1998; Krogh & Hertz, 1992; Le Cun et al., 1991; Mei & Montanari, 2019; Opper et al., 1990; Vallet et al., 1989; Watkin et al., 1993). The classical learning theory shows that when the number of parameters (which reflects the model complexity) increases, the test error (generalization risk) first decreases due to more fitting power, and then increases due to overfitting. The generalization risk attains a peak at the interpolation threshold (the number of parameters equals the number of data points so that the model interpolates the data). This results in a U-shaped risk curve if we plot the test error versus the number of parameters. The double descent risk curve posits that the risk will decrease (again) if one further increases the model complexity beyond the interpolation threshold (Belkin et al., 2019). Thus there is a second descent in addition to the first one in the U-shaped stage of the curve. Belkin et al. (2019) presented empirical results and showed the existence of such double descent behavior in the random Fourier features model, the fully connected neural network, and the random forest model. Prior to (Belkin et al., 2019), earlier studies of the shape and features of the risk curve in a number of contexts include (Vallet et al., 1989; Opper et al., 1990; Le Cun et al., 1991; Krogh & Hertz, 1992; Bös & Opper, 1998; Watkin et al., 1993; Advani et al., 2020). Loog et al. (2020) presented a prehistory of the double descent phenomenon. Belkin et al. (2020) proved the double descent curve in the Gaussian model and the Fourier series model. Mei & Montanari (2019) theoretically established the double descent curve of the random features regression. Bartlett et al. (2020); Tsigler & Bartlett (2020) characterized the conditions for ridgeless and ridge linear regression problems, respectively, under which the minimum-norm interpolants achieve near-optimal generalization risk. Liang et al. (2020) showed that the test error of the minimum-norm interpolator of data in reproducing kernel Hilbert space is upper bounded by a multiple descent curve as the model complexity increases. They also presented a numerical result supporting that the test error itself exhibits a multiple descent curve.

Chen et al. (2020a) proved that the multiple descent curve does exist for the minimum-norm interpolator in linear regression and that the curve can be even designed.

Following the parameter-wise double descent, research interest extended to epoch-wise and sample-wise double descent (Nakkiran et al., 2020; Chen et al., 2020b; Min et al., 2021; Nakkiran et al., 2021). Nakkiran et al. (2020) observed from their numerical result that the generalization risk experiences a double descent as one keeps the model size fixed and increases the training time. They called this observation epoch-wise double descent. Nakkiran et al. (2020) also noted sample-wise non-monotonicity, which means that more data can hurt generalization. Nakkiran et al. (2021) proved that for isotropic features, optimally regularized ridge regression yields a monotonic generalization risk curve with more samples. Nakkiran et al. (2021) also showed that if the features are formed by projecting high-dimensional isotropic data to a random low-dimensional space (say, $d$-dimensional), the optimally regularized ridge regression has a monotonic generalization risk curve with increasing $d$ (the model size). Sample-wise non-monotonicity and double descent was also observed in (Chen et al., 2020b; Min et al., 2021) in adversarially trained models. C ompared to (Wu & Xu, 2020; ichi Amari et al., 2021; Dobriban & Wager, 2018; Richards et al., 2021), in what follows, we highlight our contributions and the differences from them. First, our major contribution is providing a rigorous proof for the existence of sample-wise (test error vs. the number of training samples) double and multiple descent in linear regression. However, (Richards et al., 2021) only mentioned parameter-wise double descent (test error vs. model capacity) in their related work. (ichi Amari et al., 2021) only mentioned epoch-wise (test error vs. training time) double descent in Appendix A.2. Neither (Richards et al., 2021) nor (ichi Amari et al., 2021) mentioned multiple descent. Second, we made and theoretically proved the observation that an ill-conditioned covariance matrix is a sufficient condition for the existence of sample-wise multiple descent. To the best of our knowledge, our work is the first paper that pointed this out. Third, we solved the Stieltjes transform explicitly and derived explicit formulae for the risk and variance in our setup. In addition, we also provided rigorous treatment to the ridgeless setting and also obtained explicit formulae for it. Fourth, there is another difference between our paper and the papers that the reviewer mentioned. (Wu & Xu, 2020; ichi Amari et al., 2021; Dobriban & Wager, 2018; Richards et al., 2021) assumed a prior on the true linear model and takes expectation over the prior. In our paper, we do not assume a prior on the true linear model and our risk does not take the expectation over a random true linear model.

In the setting of generally anisotropic features, this paper gives an asymptotic characterization of the generalization risk curve with more samples. The asymptotic regime is an approximation for large $n, d$ and can also shed light on practical machine learning problems. We first introduce our problem setup.

## 1.1 PROBLEM SETUP

**Data Distribution** Let $\Sigma \in \mathbb{R}^{d \times d}$ be a positive semi-definite matrix which is termed the *covariance matrix*, and let $\theta^* \in \mathbb{R}^d$. The eigenvalues of $\Sigma$ are $\lambda_1, \ldots, \lambda_m$ with multiplicity $d_1, \ldots, d_m$, respectively. We have $d = \sum_{i=1}^{m} d_i$. Assume that $\lambda_1, \ldots, \lambda_m$ are fixed, distinct, all positive, and do not depend on $d$ (i.e., for all $d$, the eigenvalue of $\Sigma$ are always $\lambda_1, \ldots, \lambda_m$). We assume the following data distribution $\mathcal{D}$ for $(x, y) \in \mathbb{R}^d \times \mathbb{R}$:

$$x \sim \mathcal{N}(0, \Sigma), \quad y = x^\top \theta^* + \epsilon,$$

where $x$ and $\epsilon$ are independent and $\epsilon \sim \mathcal{N}(0, \sigma^2)$. In practice, there are natural random variables $x$ that satisfy our assumption. For example, assume that we want to use machine A to measure the length of several objects and use machine B to measure their temperature. The measured lengths and temperatures follow an i.i.d. Gaussian distribution. However, the variance of measurement of machine A is different from that of machine B. Then we consider the random vector formed by the measurements $x = (l_1, \ldots, l_n, t_1, \ldots, t_n)$, where $l_i$ and $t_i$ are the length and temperature of object $i$, respectively. This results in a block-structured covariance matrix. When we measure more objects, the size of the covariance matrix tends to infinity. Second, the motivation came from (Nakkiran et al., 2021). (Nakkiran et al., 2021) observed empirically in their Figure 2 that when the covariance matrix has a block structure (specifically, there are only two fixed different eigenvalues 10 and 1), the expected excess risk exhibits multiple descent. We quantitatively studied this observation and

obtained the related formulae. The excess risk of an estimator $\theta \in \mathbb{R}^d$ is given by

$$R(\theta) = \mathbb{E}_{x,y \sim \mathcal{D}} \left[ \left( y - x^\top \theta \right)^2 - \left( y - x^\top \theta^* \right)^2 \right] .$$

Assume that the training data $\{(x_i, y_i)\}_{i=1}^n \subseteq \mathbb{R}^d \times \mathbb{R}$ is drawn i.i.d. from $\mathcal{D}$. Write

$$X = \begin{pmatrix} x_1^\top \\ \vdots \\ x_n^\top \end{pmatrix} \in \mathbb{R}^{n \times d}, \quad \mathbf{y} = \begin{pmatrix} y_1 \\ \vdots \\ y_n \end{pmatrix} \in \mathbb{R}^n . \tag{1}$$

We have $\mathbf{y} = X\theta^* + \epsilon$, where $\epsilon \sim \mathcal{N}(0, \sigma^2 I_n)$.

### Ridge Estimator and Minimum-Norm Estimator

**Definition 1** (Ridge estimator)**.** The ridge estimator $\hat{\theta}_{\lambda,n,d} \in \mathbb{R}^d$ ($\lambda > 0$) solves the following minimization problem

$$\min_{\theta \in \mathbb{R}^d} \frac{1}{n} \|X\theta - \mathbf{y}\|_2^2 + \lambda \|\theta\|_2^2 .$$

**Definition 2** (Minimum-norm estimator)**.** The minimum-norm estimator (also known as the ridge-less estimator) $\hat{\theta}_{0,n,d} \in \mathbb{R}^d$ solves the following minimization problem

$$\min_{\theta \in \mathbb{R}^d} \|\theta\|_2 \quad \text{such that} \quad \|X\theta - \mathbf{y}\|_2 = \min_{\theta \in \mathbb{R}^d} \|X\theta - \mathbf{y}\|_2 .$$

We are interested in the expected excess risk of $\hat{\theta}_{\lambda,n,d}$, which is given by

$$R_{\lambda,n,d} = \mathbb{E} \left[ R \left( \hat{\theta}_{\lambda,n,d} \right) \right] .$$

The expectation is taken over the randomness of the training data $\{(x_i, y_i)\}_{i=1}^n$.

**Asymptotic Regime** Let $\Pi_i \in \mathbb{R}^{d \times d}$ be the orthogonal projection to the eigenspace of $\lambda_i$. This paper focuses on the asymptotic behavior of the expected excess risk of $\hat{\theta}_{\lambda,n,d}$ where $n, d_i \to +\infty$, $d_i/n \to z_i$ ($z_i$ is a fixed positive constant), and $\|\Pi_i \theta^*\|_2 \to \eta_i$. In other words, we are interested in

$$\lim_{\substack{n,d_i \to +\infty \\ d_i/n \to z_i \\ \|\Pi_i \theta^*\|_2 \to \eta_i}} R_{\lambda,n,d} .$$

### 1.2 Our Contributions

Our contributions are summarized as follows.

1. We obtain the formulae for the limiting bias and variance, and thereby the limiting risk. We use two methods to obtain these formulae. Specifically, we obtain the limiting bias and variance by solving the Stieltjes transform and computing its derivatives and antiderivatives. We also use convex Gaussian min-max theorem (CGMT) (Thrampoulidis et al., 2015) to compute the limiting variance. The advantage of the CGMT method is that it is more mathematically tractable for the ridgeless estimator. Through the CGMT approach, we obtain a closed-form formula for the variance in the underparameterized regime and simplify the formula for the variance in the overparameterized regime. Moreover, based on the simplified formula, we deduce a closed-form expression for the variance if the covariance matrix of the data distribution has two different eigenvalues.

2. We find and theoretically prove that sample-wise multiple descent happens when the covariance matrix has eigenvalues of very different orders of magnitude (thus the covariance matrix is highly ill-conditioned).

3. We show that if the true linear model $\theta^*$ satisfies $\|\Pi_i \theta^*\|_2 = \sqrt{\frac{d_i}{d}}$, optimal regularization (i.e., pick $\lambda$ that minimizes the generalization risk of $\hat{\theta}_{\lambda,n,d}$) results in a monotone generalization risk curve—in other words, with optimal regularization, more data samples

always improve generalization. Thus there is no sample-wise double or multiple descent. This provides a theoretical proof of a phenomenon observed in (Nakkiran et al., 2021) that optimal regularization can mitigate double descent for anisotropic data. Note that without regularization, there will be a blow-up in expected excess risk when $n = d$ (the linear model exactly interpolates the data) and therefore, there is no samplewise descent across the under- and over-parameterized regimes.

## 2 PRELIMINARIES

**Notation** Write $[m]$ for $\{1, 2, \ldots, m\}$. Let $\mathbf{i}$ denote the imaginary unit. If $x \in \mathbb{R}^n$ and $\Sigma \in \mathbb{R}^{n \times n}$ is a positive semidefinite matrix, write $\|x\|_\Sigma \triangleq \sqrt{x^\top \Sigma x}$. For a vector $x$, let $\|\cdot\|_1$ and $\|\cdot\|_2$ denote the $\ell^1$ and $\ell^2$ norm, respectively. Let $\odot$ denote the Hadamard (entry-wise) product between vectors. Write $\|\cdot\|_2$ and $\|\cdot\|_F$ for the spectral matrix norm and Frobenius matrix norm, respectively. Let $\preccurlyeq$ denotes the Loewner order. For two square matrices $A$ and $B$ of the same size, write $A \preccurlyeq B$ if $B - A$ is positive semidefinite. Define $\mathrm{spec}\,(A)$ as the set of all eigenvalues of $A$. Let $O(d) = \{A \in \mathbb{R}^{d \times d} \mid AA^\top = A^\top A = I_d\}$ denote the set of $d \times d$ orthogonal matrices. Define $\mathbb{S}^{d-1}(r) \triangleq \{x \in \mathbb{R}^d \mid \|x\|_2 = r\}$. Denote almost sure convergence by $\overset{\text{a.s.}}{\to}$, and convergence in probability $\mathrm{plim}$ and $\overset{P}{\to}$.

**Ridge Estimator and Minimum-Norm Estimator** We begin with the equivalent characterizations of the ridge and minimum-norm estimator. An equivalent characterization of the ridge estimator $\hat{\theta}_{\lambda,n,d}$ is

$$\hat{\theta}_{\lambda,n,d} = \left(X^\top X + \lambda n I_d\right)^{-1} X^\top \mathbf{y} = X^\top \left(\lambda n I_n + XX^\top\right)^{-1} \mathbf{y} \,. \tag{2}$$

The second equality in Equation (2) is because of the Sherman–Morrison–Woodbury formula. A proof of Equation (2) can be found in (Tsigler & Bartlett, 2020).

An equivalent definition of the minimum-norm estimator $\hat{\theta}_{0,n,d}$ is that $\hat{\theta}_{0,n,d}$ solves the following minimization problem

$$\min_{\theta \in \mathbb{R}^d} \|\theta\|_2 \quad \text{such that} \quad X^\top X \theta = X^\top \mathbf{y} \,.$$

Thus we have

$$\hat{\theta}_{0,n,d} = \left(X^\top X\right)^+ X^\top \mathbf{y} = X^\top \left(XX^\top\right)^+ \mathbf{y} = X^+ \mathbf{y} \,,$$

where $A^+$ denotes the pseudo-inverse of $A$. The second and third equalities are because of the identity $X^+ = \left(X^\top X\right)^+ X^\top = X^\top \left(XX^\top\right)^+$. The minimum-norm estimator is the limit of the ridge estimator $\hat{\theta}_{\lambda,n,d}$ as $\lambda \to 0^+$:

$$\hat{\theta}_{0,n,d} = \lim_{\lambda \to 0^+} \hat{\theta}_{\lambda,n,d} \,.$$

This is because of the identity $\lim_{\lambda \to 0^+} \left(X^\top X + \lambda n I_d\right)^{-1} X^\top = \lim_{\lambda \to 0^+} X^\top \left(\lambda n I_n + XX^\top\right)^{-1} = X^+$.

**Bias-Variance Decomposition of Expected Excess Risk** We first show that the excess risk of an estimator $\theta$ equals the norm of $\theta - \theta^*$:

$$R(\theta) = \mathbb{E}_{(x,y) \sim \mathcal{D}} \left[\left(y - x^\top \theta\right)^2 - \left(y - x^\top \theta^*\right)\right] = \mathbb{E}_x \left[\left(x^\top \left(\theta^* - \theta\right)\right)^2\right]$$

$$= \mathbb{E}\left[\left(\theta^* - \theta\right)^\top \Sigma \left(\theta^* - \theta\right)\right] = \mathbb{E}\left[\|\theta^* - \theta\|_\Sigma^2\right] \,.$$

For the ridge estimator, the expected excess risk is

$$\begin{aligned}
R_{\lambda,d,n} =& \mathbb{E}\left[\|\theta^* - X^\top(n\lambda I_n + XX^\top)^{-1}(X\theta^* + \boldsymbol{\epsilon})\|_\Sigma^2\right] \\
=& \mathbb{E}\left[\|(I_d - X^\top(n\lambda I_n + XX^\top)^{-1}X)\theta^* - X^\top(n\lambda I_n + XX^\top)^{-1}\boldsymbol{\epsilon}\|_\Sigma^2\right] \\
=& \mathbb{E}\left[\|(I_d - X^\top(n\lambda I_n + XX^\top)^{-1}X)\theta^*\|_\Sigma^2\right] + \mathbb{E}\left[\|X^\top(n\lambda I_n + XX^\top)^{-1}\boldsymbol{\epsilon}\|_\Sigma^2\right] \\
=& \mathbb{E}\left[\|(I_d - X^\top(n\lambda I_n + XX^\top)^{-1}X)\theta^*\|_\Sigma^2\right] + \sigma^2 \mathbb{E}\,\mathrm{tr}\left[X\Sigma X^\top(n\lambda I_n + XX^\top)^{-2}\right] \\
\triangleq& B_{\lambda,d,n} + V_{\lambda,d,n} \,.
\end{aligned} \tag{3}$$

For the minimum-norm estimator, the expected excess risk is

$$
\begin{aligned}
R_{0,d,n} &= \mathbb{E}\left[\|\theta^* - X^+(X\theta^* + \boldsymbol{\epsilon})\|_\Sigma^2\right] \\
&= \mathbb{E}\left[\left\|\left(I_d - X^+X\right)\theta^* - X^+\boldsymbol{\epsilon}\right\|_\Sigma^2\right] \\
&= \mathbb{E}\left[\left\|\left(I_d - X^+X\right)\theta^*\right\|_\Sigma^2\right] + \mathbb{E}\left[\left\|X^+\boldsymbol{\epsilon}\right\|_\Sigma^2\right] \\
&= \mathbb{E}\left[\left\|\left(I_d - X^+X\right)\theta^*\right\|_\Sigma^2\right] + \sigma^2\mathbb{E}\operatorname{tr}\left[\left(X^+\right)^\top \Sigma X^+\right] \\
&\triangleq B_{0,d,n} + V_{0,d,n}\,.
\end{aligned}
\tag{4}
$$

We call $B_{\lambda,d,n}$ and $B_{0,d,n}$ the bias term, and call $V_{\lambda,d,n}$ and $V_{0,d,n}$ the variance term. The bias and variance for the minimum-norm estimator are the limit of their counterpart for the ridge estimator as $\lambda \to 0^+$, i.e., $\lim_{\lambda\to 0^+} B_{\lambda,d,n} = B_{0,d,n}$ and $\lim_{\lambda\to 0^+} V_{\lambda,d,n} = V_{0,d,n}$ (this can be shown by Lebesgue's dominated convergence theorem, see our proof in Lemma 5 and Lemma 6, respectively).

## 3 MAIN RESULTS

### 3.1 LIMITING RISK AND SAMPLE-WISE MULTIPLE DESCENT

We study the limiting bias and variance for a linear regression problem in which the data distribution follows a multivariate normal distribution, the spectrum of the covariance matrix exhibits a block structure and tends to a discrete distribution. Thanks to the random matrix theory, we obtain the formulae (presented in Theorem 1) for the limiting bias and variance, and thereby the total risk.

We use two methods to obtain these formulae. The first method is through the Stieltjes transform of the matrix $\frac{1}{n}XX^\top$. The central quantity for computing the limiting bias and variance through the first method is the solution $\rho^*$ to the optimization problem Equation (5) in Item 1 of Theorem 1. Item 1 guarantees the existence of a solution and determines its optimality condition Equation (6). Item 2 computes the Jacobian matrix of $\rho^*$ with respect to $\lambda_i$ and provides a closed-form formula to compute the Jacobian matrix. Equation (9) and Equation (10) in Item 4 give the formulae for the limiting bias obtained by the first method. Equation (11) and Equation (12) give the limiting variance.

The second method is through the convex Gaussian min-max theorem (CGMT) (Thrampoulidis et al., 2015). The central quantity is the solution $\mathbf{r}^*$ to the minimax optimization problem Equation (8) in Item 3. We use CGMT to obtain the formulae for the variance term. They are presented in Equation (13) and Equation (14) in Item 4.

**Theorem 1.** *The following statements hold:*

1. *There exists a minimizer $\rho \in \mathbb{R}_+^m$ that solves*

$$
\inf_{\rho\in\mathbb{R}_+^m}\left[\log\left(\lambda + \sum_{j=1}^m \lambda_j\rho_j\right) + \sum_{j=1}^m\left(\rho_j - z_j(\log\frac{\rho_j}{z_j} + 1)\right)\right].
\tag{5}
$$

   *The minimizer $\rho^*$ satisfies*

$$
\frac{\lambda_i}{\lambda + \sum_{j=1}^m \lambda_j\rho_j^*} + 1 - \frac{z_i}{\rho_i} = 0\,, \quad \forall i \in [m]\,.
\tag{6}
$$

2. *Let $\rho^* \in \mathbb{R}^m$ be a minimizer of Equation (5) and $J = \frac{\partial\rho^*}{\partial\boldsymbol{\lambda}} \in \mathbb{R}^{m\times m}$ be the Jacobian matrix $J_{ij} = \frac{\partial\rho_i^*}{\partial\lambda_j}$. Then $J$ is given by*

$$
J = \left(\operatorname{diag}\left(\boldsymbol{\lambda}\right) + \left(\lambda + \boldsymbol{\lambda}^\top\rho^*\right)I_m - \left(\mathbf{z} - \rho^*\right)\boldsymbol{\lambda}^\top\right)^{-1}\left(\left(\mathbf{z} - \rho^*\right)\rho^{*\top} - \operatorname{diag}\left(\rho^*\right)\right)
$$

   *and the matrix $\left(\operatorname{diag}\left(\boldsymbol{\lambda}\right) + \left(\lambda + \boldsymbol{\lambda}^\top\rho^*\right)I_m - \left(\mathbf{z} - \rho^*\right)\boldsymbol{\lambda}^\top\right)$ is always invertible.*

3. *Define $\mathbf{r} = (r_1, \ldots, r_m)$, $\boldsymbol{\lambda} = (\lambda_1, \ldots, \lambda_m)$, and*

$$
\vartheta(r_t, \mathbf{r}, \boldsymbol{\lambda}) = 2r_t\sqrt{1 + \sum_{i\in[m]} r_i^2} - 2r_t\sum_{i\in[m]}\sqrt{z_i}r_i + \sum_{i\in[m]}\frac{1}{\lambda_i}r_i^2 - \lambda r_t^2\,.
\tag{7}
$$

*For any $K_t \geq \frac{2}{\lambda}$ and $K_u \geq \frac{2\lambda_+ \left(2+\sqrt{\gamma}\right)}{\lambda}$, we have*

$$\max_{0 \leq r_t \leq K_t} \min_{0 \leq r_i \leq K_u} \vartheta(r_t, \mathbf{r}, \boldsymbol{\lambda}) = \min_{0 \leq r_i \leq K_u} \max_{0 \leq r_t \leq K_t} \vartheta(r_t, \mathbf{r}, \boldsymbol{\lambda}) = \max_{r_t \geq 0} \min_{r_i \geq 0} \vartheta(r_t, \mathbf{r}, \boldsymbol{\lambda}) = \min_{r_i \geq 0} \max_{r_t \geq 0} \vartheta(r_t, \mathbf{r}, \boldsymbol{\lambda}) \tag{8}$$

*and the above optimization problem has a solution.*

4. *Let* $\mathbf{r}^* = (r_1^*, \ldots, r_m^*)$ *solve Equation* (8). *Define* $\mathbf{q} = \left(\eta_1^2/z_1, \ldots, \eta_m^2/z_m\right)^\top$ *and view* $\boldsymbol{\lambda} = (\lambda_1, \ldots, \lambda_m)^\top$ *as a column vector. The limiting bias is given by*

$$\lim_{\substack{n,d_i \to +\infty \\ d_i/n \to z_i \\ \|\Pi_i \theta^*\|_2 \to \eta_i}} B_{\lambda,d,n} = \mathbf{q}^\top \left(\boldsymbol{\lambda} \odot \rho^* + J \boldsymbol{\lambda}^{\odot 2}\right), \tag{9}$$

$$\lim_{\substack{n,d_i \to +\infty \\ d_i/n \to z_i \\ \|\Pi_i \theta^*\|_2 \to \eta_i}} B_{0,d,n} = \lim_{\lambda \to 0^+} \mathbf{q}^\top \left(\boldsymbol{\lambda} \odot \rho^* + J \boldsymbol{\lambda}^{\odot 2}\right). \tag{10}$$

*The limiting variance is given by*

$$\lim_{\substack{n,d_i \to +\infty \\ d_i/n \to z_i}} V_{\lambda,d,n} = \sigma^2 \frac{\boldsymbol{\lambda}^{\odot 2 \top}(\rho^* + J^\top \boldsymbol{\lambda})}{(\lambda + \boldsymbol{\lambda}^\top \rho^*)^2}, \tag{11}$$

$$\lim_{\substack{n,d_i \to +\infty \\ d_i/n \to z_i}} V_{0,d,n} = \sigma^2 \lim_{\lambda \to 0^+} \frac{\boldsymbol{\lambda}^{\odot 2 \top}(\rho^* + J^\top \boldsymbol{\lambda})}{(\lambda + \boldsymbol{\lambda}^\top \rho^*)^2}, \tag{12}$$

$$\lim_{\substack{n,d_i \to +\infty \\ d_i/n \to z_i}} V_{\lambda,d,n} = \sigma^2 \sum_{i=1}^m r_i^{*2}, \tag{13}$$

$$\lim_{\substack{n,d_i \to +\infty \\ d_i/n \to z_i}} V_{0,d,n} = \sigma^2 \lim_{\lambda \to 0^+} \sum_{i=1}^m r_i^{*2}. \tag{14}$$

Figure 1 illustrates the theoretical and numerical values of the bias, variance, and total risk. We observe a triple descent in Figure 1a where the covariance matrix has three blocks, and a quadruple descent in Figure 1b where the covariance has four blocks. In the three-block example, we set $\lambda_3 \gg \lambda_2 \gg \lambda_1$ ($\lambda_1 = 1$, $\lambda_2 = 100$, $\lambda_3 = 1000$). In the four-block example, we set $\lambda_4 \gg \lambda_3 \gg \lambda_2 \gg \lambda_1$ ($\lambda_1 = 1$, $\lambda_2 = 100$, $\lambda_3 = 10^4$, $\lambda_4 = 10^7$). For the values of other parameters, please refer to the caption of Figure 1 Our findings provide an explanation for the occurrence of sample-wise multiple descent: it occurs when the covariance matrix is highly ill-conditioned. Moreover, we find that the generalization risk curve is continuous in ridge regression ($\lambda > 0$) while it blows up at $n = d$ in ridgeless regression ($\lambda = 0$). We can see the singularity (at $n = d = 200$) of the ridgeless generalization risk curve in Figure 2a.

Following Theorem 1, we focus on the variance in the ridgeless case ($\lambda = 0$) and further study the expressions in Equation (13) and Equation (14). We find that the variance exhibits sharply different behaviors in the underparameterized and overparameterized regimes. Recall that we will let $n, d_i \to +\infty$ and keep $d_i/n \to z_i$. Then $d/n \to \sum_{i \in [m]} z_i$. If $\lim d/n = \sum_{i \in [m]} z_i > 1$, we are in the underparameterized regime. In this regime, the bias vanishes and therefore the risk equals the variance. If $\lim d/n < 1$, we are in the overparameterized regime.

**Theorem 2.** *If* $d/n \to \sum_{i \in [m]} z_i > 1$ *and* $\mathbf{r}^* = (r_1^*, \ldots, r_m^*)$ *solves*

$$\min_{r_i \geq 0} \sum_{i \in [m]} \frac{1}{\lambda_i} r_i^2 \quad \text{subject to} \quad \sqrt{\sum_{i \in [m]} r_i^2 + 1} = \sum_{i \in [m]} \sqrt{z_i} r_i,$$

*then we have an optimality condition for* $\mathbf{r}^*$:

$$\frac{r_i^*}{r_j^*} = \frac{\lambda_i}{\lambda_j} \cdot \frac{\sqrt{z_i} A^* - r_i^*}{\sqrt{z_j} A^* - r_j^*}, \quad i, j \in [m], \tag{15}$$

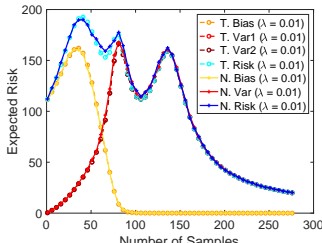 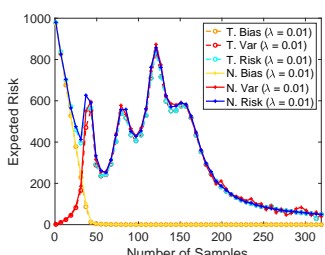

(a) Sample-wise Triple Descent

(b) Sample-wise quadruple descent

Figure 1: Figure 1a and Figure 1b illustrate sample-wise triple and quadruple descent, respectively. We specify the parameters that we used as follows. Figure 1a: There are 3 blocks. We set $d_1 = 60$, $d_2 = d_3 = 40$, $\lambda_1 = 1$, $\lambda_2 = 100$, $\lambda_3 = 1000$, $\|\Pi_1\theta^*\|_2 = \|\Pi_3\theta^*\|_2 = 0.1$ and $\|\Pi_2\theta^*\|_2 = 1$. The three descents occur at $n = 36, 80, 136$, respectively. Figure 1b: There are 4 blocks. We set $d_1 = d_2 = d_3 = d_4 = 40$, $\lambda_1 = 1$, $\lambda_2 = 100$, $\lambda_3 = 10^4$, $\lambda_4 = 10^7$, and $\|\Pi_i\theta^*\|_2 = 0.01(i \in [4])$. The four descents occur at around $n = 1, 37, 80, 120, 150$, respectively. In the legend, the items starting with "T." are theoretical values predicted by Theorem 1. Items starting with "N." are numerical values. We plot two curves for the variance in Figure 1a. "T. Var1" is obtained by Equation (11) of Theorem 1. "T. Var2" is obtained by Equation (13).

where $A^* = \sqrt{\sum_{i\in[m]} r_i^{*2} + 1}$. Moreover, we have $\lim_{\substack{n,d_i\to+\infty \\ d_i/n\to z_i}} V_{0,d,n} = \sigma^2 \lim_{\lambda\to 0^+} \sum_{i=1}^m r_i^{*2}$.

If $d/n \to \sum_{i\in[m]} z_i < 1$, then we have

$$\lim_{\substack{n,d_i\to+\infty \\ d_i/n\to z_i}} V_{0,d,n} = \sigma^2 \frac{\sum_{i\in[m]} z_i}{1 - \sum_{i\in[m]} z_i}.$$

.

**Corollary 1.** *If $m = 1$ and $d/n \to z_1 > 1$, we have $\lim_{\substack{n,d_i\to+\infty \\ d_i/n\to z_i}} V_{0,d,n} = \sigma^2 \frac{1}{z_1-1}$.*

*Proof.* In the case $m = 1$, we have $r_1^*$ solves $\min_{r_1\geq 0} \frac{1}{\lambda_1} r_1^2$ subject to $\sqrt{r_1^2 + 1} = \sqrt{z_1} r_1$. The equality constraint gives $r_1^{*2} = \frac{1}{z_1-1}$. Then by Theorem 2, the limiting variance is $\sigma^2 r_1^{*2} = \sigma^2 \frac{1}{z_1-1}$. □

In Theorem 2, we find that in the underparameterized regime, $\mathbf{r}^*$ solves an equality-constrained minimization problem. In the proof of Theorem 2, we see that the equality constraint is feasible in the underparameterized regime but infeasible in the overparameterized regime. Moreover, we present an optimality condition for $\mathbf{r}^*$, which will be used in Theorem 3 to study the two-block ($m = 2$) case. If the data distribution is isotropic (which means that the covariance matrix is a scalar matrix), Collorary 1 shows that the limiting variance is $\sigma^2 \frac{1}{z_1-1}$, which agrees with (Hastie et al., 2019, Theorem 1).

In the overparameterized regime, however, we find that the limiting variance does not depend on the spectrum $\{\lambda_1, \ldots, \lambda_m\}$ of the covariance matrix and only depends on the noise intensity $\sigma$ and the ratios $z_i = \lim d_i/n$. This agrees with (Hastie et al., 2019, Proposition 2).

In Theorem 3, we study the case $m = 2$ and present a concrete closed-form formula for the limiting variance in the overparameterized regime. Recall that the limiting variance in the underparameterized regime has a closed-form $\sigma^2 \frac{\sum_{i\in[m]} z_i}{1-\sum_{i\in[m]} z_i}$ for general $m$, as shown in Theorem 2.

**Theorem 3.** *If $m = 2$ and $d/n \to z_1 + z_2 > 1$, we have*

$$\lim_{\substack{n,d_i\to+\infty \\ d_i/n\to z_i}} V_{0,d,n} = \sigma^2 \frac{q^2 + 1}{q^2(z_1 - 1) + 2q\sqrt{z_1 z_2} + z_2 - 1}.$$

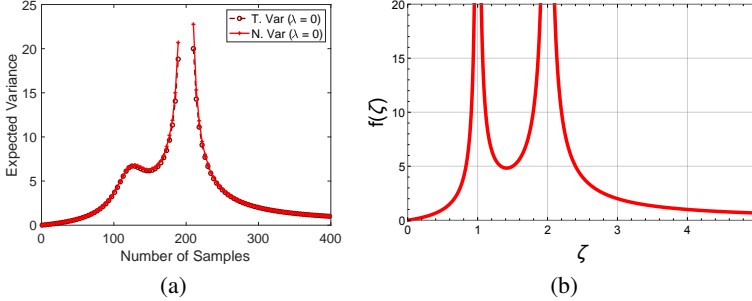

Figure 2: Figure 2a: We illustrate sample-wise triple descent of the variance term in ridgeless regression ($\lambda = 0$). There are 2 blocks. We set $d_1 = 80$, $d_2 = 120$, $\lambda_1 = 1$ and $\lambda_2 = 100$. The two descents occur at around $n = 125, 200$, respectively. In the legend, "T. Var" denotes the theoretical values predicted by Theorem 3. "N. Var" denotes the numerical values. Figure 2b: Function $f(\zeta)$ defined in Equation (17) with $\sigma = 1$.

*where*

$$q = \frac{\lambda_1 (z_1 - 1) + \lambda_2 (1 - z_2) + \sqrt{(\lambda_1 (z_1 - 1) + \lambda_2 (1 - z_2))^2 + 4\lambda_1\lambda_2 z_1 z_2}}{2\lambda_2\sqrt{z_1 z_2}} . \tag{16}$$

We illustrate the theoretical values predicted by Theorem 3 (overparameterized regime) and Theorem 2 (underparameterized regime) in Figure 2a and compare it to the numerical values.

**Corollary 2** (Triple descent in the two-block case). *Assume $m = 2$, $z_1 = z_2$, $d/n \to \zeta = 2z_1$, and $\lambda_2/\lambda_1 = \varrho$. Define $f_\varrho(\zeta) = \lim_{\substack{n,d_i \to +\infty \\ d_i/n \to z_i}} V_{0,d,n}$. We have*

$$f(\zeta) \triangleq \lim_{\varrho \to +\infty} f_\rho(\zeta) = \begin{cases} \sigma^2 \frac{\zeta}{1-\zeta} & \zeta < 1, \\ \sigma^2 \left(\frac{1}{\zeta-1} + \frac{2}{2-\zeta} - 1\right) & 1 < \zeta < 2 \\ \sigma^2 \frac{2}{\zeta-2} & \zeta > 2 \end{cases} . \tag{17}$$

*There exists $\zeta_1, \zeta_2, \zeta_3, \zeta_4$ and $\varrho_0$ such that for all $\varrho > \varrho_0$, we have $f'_\varrho(\zeta_1) < 0$, $f'_\varrho(\zeta_2) > 0$, $f'_\varrho(\zeta_3) < 0$, and $f'_\varrho(\zeta_4) < 0$.*

*Proof.* The case $\zeta < 1$ is already given in Theorem 2. In the sequel, assume $\zeta > 1$. Define $q$ as in Equation (16). We have

$$q = \frac{\zeta + \sqrt{\zeta^2(\varrho+1)^2 - 4\zeta(\varrho-1)^2 + 4(\varrho-1)^2} - (\zeta-2)\varrho - 2}{2\zeta\varrho} .$$

Recall Theorem 3, we get

$$f_\varrho(\zeta) = \lim_{\substack{n,d_i \to +\infty \\ d_i/n \to z_i}} V_{0,d,n} = \frac{2\left(q^2 + 1\right)}{\zeta\left(q + 1\right)^2 - 2\left(q^2 + 1\right)} = \frac{2\sigma^2}{\zeta\frac{(q+1)^2}{q^2+1} - 2} .$$

Direct calculation yields

$$\lim_{\varrho \to +\infty} f_\varrho(\zeta) = \begin{cases} \sigma^2 \frac{\zeta}{1-\zeta} & \zeta < 1, \\ \sigma^2 \left(\frac{1}{\zeta-1} + \frac{2}{2-\zeta} - 1\right) & 1 < \zeta < 2, \\ \sigma^2 \frac{2}{\zeta-2} & \zeta > 2. \end{cases}$$

$$g(\zeta) \triangleq \lim_{\varrho \to +\infty} f'_\varrho(\zeta) = \begin{cases} \sigma^2 \frac{1}{(\zeta-1)^2} & \zeta < 1, \\ \sigma^2 \frac{\zeta^2-2}{(\zeta^2-3\zeta+2)^2} & 1 < \zeta < 2, \\ \sigma^2 \frac{-2}{(\zeta-2)^2} & \zeta > 2. \end{cases}$$

The function $g(\zeta) > 0$ if $\zeta \in \left(\sqrt{2}, 2\right)$ and we have $g(\zeta) < 0$ if $\zeta < \sqrt{2}$ or $\zeta > 2$. Pick $\zeta_1 > 2 > \zeta_2 > \sqrt{2} > \zeta_3 > 1 > \zeta_4$. Then we have $g(\zeta_1) < 0$, $g(\zeta_2) > 0$, $g(\zeta_3) < 0$, and $g(\zeta_4) > 0$. There exists $\varrho_0$ such that for all $\varrho > \varrho_0$, we have $f'_\varrho(\zeta_1) < 0$, $f'_\varrho(\zeta_2) > 0$, $f'_\varrho(\zeta_3) < 0$, and $f'_\varrho(\zeta_4) < 0$. $\qquad\square$

Collorary 2 theoretically proves that there exists triple descent when $m = 2$ and $\lambda_2 \gg \lambda_1$. Note that a larger $\zeta = \lim d/n$ reflects a relatively smaller $n$. If $f'_\varrho(\zeta) < 0$, then $f_\varrho(\zeta)$ decreases on a neighborhood of $\zeta$ and therefore the limiting variance increases with a relatively larger $n$. As $n$ becomes relatively larger, we see an increasing stage, a decreasing stage, and finally an increasing stage in order in the overparameterized regime ($n < d$). When we further increase $n$ and enter the underparameterized regime, we observe a decreasing stage. We illustrate $f(\zeta)$ in Figure 2b. In Figure 2b, we observe two singularities at $\zeta = 1$ and $\zeta = 2$.

## 3.2 Optimal Regularization Monotonizes Generalization Risk Curve

Recall the definition of the ridge estimator in Definition 1. Since this subsection concerns sample-wise monotonicity, we add a subscript $n$ to $X$ and $\mathbf{y}$ (they are defined by Equation (1) in Section 1.1) to emphasize that they consist of $n$ data items. Therefore we write

$$\hat{\theta}_{\lambda,n,d} \triangleq \arg\min_\theta \frac{1}{n} \|\mathbf{y}_n - X_n\theta\|_2^2 + \lambda \|\theta\|_2^2 \ .$$

In this subsection, under an assumption, we show that optimal regularization (i.e., pick $\lambda$ that minimizes the generalization risk of $\hat{\theta}_{\lambda,n,d}$) results in a monotone generalization risk curve—in other words, with optimal regularization, more data always reduces the generalization risk. The assumption is that $\|\Pi_i \theta^*\|_2 = \sqrt{\frac{d_i}{d}}$, i.e., the squared norm of the projection of $\theta^*$ onto each eigenspace of the covariance matrix is proportional to the dimension of that eigenspace. (Nakkiran et al., 2021) showed by numerical results that optimal regularization can mitigate double descent for anisotropic data distribution. We give a partial theoretical proof of their observed phenomenon.

To ease the notation, we use $\gamma_i \triangleq \lim n/d_i$ rather than $z_i \triangleq \lim d_i/n$ in Theorem 4 because a larger $\gamma$ reflects a relatively larger $n$ (in the limit). Theorem 4 shows that with the optimal regularization, the limiting risk is an increasing function of $\gamma_1, \ldots, \gamma_m$.

**Theorem 4** (Optimal regularization). *If*

$$\|\Pi_i \theta^*\|_2 = \sqrt{\frac{d_i}{d}} \ , \tag{18}$$

*then there exists a function $g(\gamma_1, \ldots, \gamma_m)$ such that $g(\gamma_1, \ldots, \gamma_m)$ is increasing in every $\gamma_i$ and*

$$\lim_{\substack{n,d_i \to \infty \\ n/d_i \to \gamma_i}} \inf_{\lambda > 0} \mathbb{E}_{X_n, \mathbf{y}_n} \left\| \hat{\theta}_{\lambda,n} - \theta^* \right\|_\Sigma^2 = g(\gamma_1, \ldots, \gamma_m) \ .$$

## 4 Conclusion

We studied the generalization risk (test error) versus the number of training samples in ridgeless regression. Under the assumption that the data distribution is Gaussian and the spectrum distribution of its covariance matrix converges to a discrete distribution, we obtained the exact formulae for the limiting bias and variance terms using the random matrix theory when the dimension and the number of training samples go to infinity in a proportional manner. Using these formulae, we proved the sample-wise multiple descent phenomenon of the generalization risk curve. Moreover, we theoretically showed that the ridge estimator with optimal regularization can result in a monotone generalization risk curve and thereby eliminate multiple descent under some assumptions.

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

| Notation | | Comments |
|---|---|---|
| $\Sigma = P\Lambda P^\top$ | $\mathbb{R}^{d\times d}$ | Covariance matrix of data |
| $P$ | $O(d)$ | |
| $\Lambda = \operatorname{diag}(\lambda_1 I_{d_1}, \ldots, \lambda_m I_{d_m})$ | $\mathbb{R}^{d\times d}$ | |
| $\theta' \triangleq P^\top \theta^*$ | $\mathbb{R}^d$ | |
| $X = Z^\top \Lambda^{1/2} P^\top = (x_1, \ldots, x_n)^\top$ | $\mathbb{R}^{n\times d}$ | $x_i \sim \mathcal{N}(0, \Sigma)$ |
| $Z = (Z_1, \ldots, Z_m)^\top$ | $\mathbb{R}^{d\times n}$ | $Z_i \in \mathbb{R}^{n\times d_i}$. Each entry follows $\mathcal{N}(0, 1)$. |

Table 1: Notation

## A    EIGENDECOMPOSITION AND MORE NOTATION

Write $\Sigma = P\Lambda P^\top$, where $P$ is an orthogonal matrix and $\Lambda = \operatorname{diag}(\lambda_1 I_{d_1}, \ldots, \lambda_m I_{d_m}) \in \mathbb{R}^{d\times d}$ is a diagonal matrix. Write $\lambda_- = \min_{i\in[m]} \lambda_i$ and $\lambda_+ = \max_{i\in[m]} \lambda_i$. We can generate $x_1, \ldots, x_n$ from standard normal random vector $z_i \sim \mathcal{N}(0, I_d)$ by setting $x_i = P\Lambda^{1/2} z_i$. Therefore, if $Z = \begin{pmatrix} z_1 & \ldots & z_n \end{pmatrix} \in \mathbb{R}^{d\times n}$, we get

$$X^\top = \begin{pmatrix} x_1 & \ldots & x_n \end{pmatrix} = P\Lambda^{1/2} \begin{pmatrix} z_1 & \ldots & z_n \end{pmatrix} = P\Lambda^{1/2} Z .$$

Take the transpose gives $X = Z^\top \Lambda^{1/2} P^\top$. Note that every entry of $Z \in \mathbb{R}^{d\times n}$ follows i.i.d. $\mathcal{N}(0, 1)$. Write $Z$ in a row-partitioned form

$$Z = \begin{pmatrix} Z_1^\top \\ \vdots \\ Z_m^\top \end{pmatrix},$$

where $Z_i \in \mathbb{R}^{n\times d_i}$. Write $P$ in a column-partitioned form

$$P = \begin{pmatrix} P_1 & \ldots & P_m \end{pmatrix},$$

where $P_i \in \mathbb{R}^{d\times d_i}$. Recall that $\Pi_i \in \mathbb{R}^{d\times d}$ denotes the orthogonal projection to the eigenspace of $\lambda_i$. We have $\Pi_i = P_i P_i^\top$. Define $\theta' \triangleq P^\top \theta^*$ and write it in a row-partitioned form

$$\theta' = \begin{pmatrix} P_1^\top \theta^* \\ \vdots \\ P_m^\top \theta^* \end{pmatrix} = \begin{pmatrix} \theta_1' \\ \vdots \\ \theta_m' \end{pmatrix}, \tag{19}$$

where $\theta_i' \in \mathbb{R}^{d_i}$. Then $\|\theta_i'\|_2 = \|P_i^\top \theta^*\|_2 = \|P_i P_i^\top \theta^*\|_2 = \|\Pi_i \theta^*\|_2$. We summarize part of the notation above in Table 1.

## B    BIAS AND VARIANCE UNDER EIGENDECOMPOSITION

Lemma 1 characterizes the smallest and largest eigenvalue of $\frac{Z^\top Z}{d}$ (if $n/d \to \gamma < 1$) and $\frac{ZZ^\top}{n}$ (if $n/d \to \gamma > 1$). Recall that we study the asymptotic regime $d_i/n \to z_i$. Therefore $\gamma = \frac{1}{\sum_{j\in[m]} z_j}$.

**Lemma 1** ((Bai & Yin, 2008, Theorem 2)). *Let $Z \in \mathbb{R}^{d\times n}$ be a random matrix whose entries are i.i.d. $\mathcal{N}(0, 1)$ random variables. As $n, d \to +\infty$, $n/d \to \gamma \in (0, 1)$, we have*

$$\lim \lambda_{\min}\left(\frac{Z^\top Z}{d}\right) = (1 - \sqrt{\gamma})^2 , \quad \lim \lambda_{\max}\left(\frac{Z^\top Z}{d}\right) = (1 + \sqrt{\gamma})^2$$

*almost surely. If $\gamma \in (1, \infty)$, as $n, d \to +\infty$, $n/d \to \gamma$, we have*

$$\lim \lambda_{\min}\left(\frac{ZZ^\top}{n}\right) = \left(1 - \sqrt{1/\gamma}\right)^2 , \quad \lim \lambda_{\max}\left(\frac{ZZ^\top}{n}\right) = \left(1 + \sqrt{1/\gamma}\right)^2$$

*almost surely.*

**Lemma 2** (Corollary 5.35 (Vershynin, 2010)). *Let $A$ be an $N \times n$ matrix whose entries are independent standard normal random variables. Then for every $t \geq 0$, with probability at least $1 - 2\exp\left(-t^2/2\right)$ one has*

$$\sqrt{N} - \sqrt{n} - t \leq s_{\min}(A) \leq s_{\max}(A) \leq \sqrt{N} + \sqrt{n} + t\,,$$

*where $s_{\min}(A)$ and $s_{\max}(A)$ are the smallest and largest singular value of A.*

**Lemma 3.** *Let $Z \in \mathbb{R}^{d \times n}$ be a random matrix whose entries are i.i.d. $\mathcal{N}(0,1)$ random variables, where $d = d(n)$ satisfies $\lim_{n \to \infty} \frac{n}{d(n)} = \gamma$. There exists universal positive constants $C_1, C_2, N$ such that for all $n > N$, we have*

$$0 < C_1 < \frac{1}{n}s_{\min}^2(Z) \leq \frac{1}{n}s_{\max}^2(Z) < C_2\,.$$

*Proof.* Since $d(n) \asymp n$, with loss of generality, we assume $n/d \to \gamma \in (0,1)$. Take $t = c_1\sqrt{n}$ in Lemma 2, where $c_1 = \frac{1}{2}\left(\frac{1}{\sqrt{\gamma}} - 1\right) > 0$. With probability at least $1 - 2e^{-c_1^2 n/2}$, we have

$$\sqrt{d} - \sqrt{n} - c_1\sqrt{n} \leq s_{\min}(Z) \leq s_{\max}(Z) \leq \sqrt{d} + \sqrt{n} + c_1\sqrt{n}\,.$$

Therefore, we deduce

$$\left(\sqrt{\frac{d}{n}} - 1 - c_1\right)^2 \leq \frac{1}{n}s_{\min}^2(Z) \leq \frac{1}{n}s_{\max}^2(Z) \leq \left(\sqrt{\frac{d}{n}} + 1 + c_1\right)^2\,.$$

Define $C_1 = \frac{1}{8}\left(\frac{1}{\sqrt{\gamma}} - 1\right)^2 > 0$ and $C_2 = \left(\frac{3}{\sqrt{\gamma}} + 1\right)^2$. Then there exists a universal constant $N_1$ such that for all $n > N_1$, with probability at least $1 - 2e^{-c_1^2 n/2}$, we have

$$0 < C_1 < \frac{1}{n}s_{\min}^2(Z) \leq \frac{1}{n}s_{\max}^2(Z) < C_2\,.$$

Define event $E_n = \left\{C_1 < \frac{1}{n}s_{\min}^2(Z) \leq \frac{1}{n}s_{\max}^2(Z) < C_2\right\}^c$. Then we have $\Pr\{E_n\} \leq 2e^{-c_1^2 n/2}$. Since $\sum_{n \geq 1} \Pr\{E_n\} \leq \sum_{n \geq 1} 2e^{-c_1^2 n/2} < \infty$, then the probability that infinitely many of $E_n$ occur is 0, i.e.,

$$\Pr\left\{\limsup_n E_n\right\} = 0\,.$$

Therefore, there exists a universal constant $N_2$ such that for all $n > N_2$, $E_n$ does not happen, in other words,

$$0 < C_1 < \frac{1}{n}s_{\min}^2(Z) \leq \frac{1}{n}s_{\max}^2(Z) < C_2$$

holds. $\qquad\square$

**Lemma 4.** *Let $Z \in \mathbb{R}^{d \times n}$ be a random matrix whose entries are i.i.d. $\mathcal{N}(0,1)$ random variables, and let $p$ be a fixed positive integer which is viewed as a constant and hidden in $\lesssim$. If $n \asymp d$, we have $\mathbb{E}\operatorname{tr}\left(ZZ^\top\right) \asymp n^2$, $\mathbb{E}\operatorname{tr}\left(ZZ^\top\right)^2 \asymp n^3$, and $\mathbb{E}\|Z\|_2^p \lesssim n^{p/2}$.*

*Proof.* We have

$$\mathbb{E}\operatorname{tr}\left(ZZ^\top\right) = \mathbb{E}\|Z\|_F^2 = \sum_{i \in [d], j \in [n]} \mathbb{E}z_{ij}^2 = nd \asymp n^2\,.$$

Write $Z = \begin{pmatrix} z_1^\top \\ \vdots \\ z_d^\top \end{pmatrix}$, where $z_i \in \mathbb{R}^n$ and $z_i \sim \mathcal{N}(0, I_n)$. We have $\mathbb{E}\left(z_i^\top z_i\right)^2 = \mathbb{E}\|z_i\|_2^4 = n(n+2)$. For $i \neq j$, we deduce $\mathbb{E}\left(z_i^\top z_j\right)^2 = \mathbb{E}\left(\|z_i\|_2 \|z_j\|_2 u^\top v\right)^2$ where $u, v \sim \operatorname{Unif}\left(\mathbb{S}^{n-1}\right)$ and $\|z_i\|_2, \|z_j\|_2, u, v$ are independent. Then we get

$$\mathbb{E}\left(\|z_i\|\|z_j\| s_i^\top s_j\right)^2 = \mathbb{E}\|z_i\|_2^2 \|z_j\|_2^2 \left(s_i^\top s_j\right)^2 = n^2 \mathbb{E}u_1^2 = n^2 \cdot \frac{1}{n} = n\,.$$

As a result, we have

$$\mathbb{E}\operatorname{tr}\left(ZZ^\top\right)^2 = \mathbb{E}\left\|ZZ^\top\right\|_F^2 = \sum_{i,j\in[d]} \mathbb{E}\left(z_i^\top z_j\right)^2 = dn(n+2) + \left(d^2 - d\right)n \asymp n^3\,.$$

By (Vershynin, 2018), there exists a universal constant $C > 0$ such that for any $t > 0$, $\mathbb{P}\left\{\|Z\|_2 > C\left(\sqrt{n} + \sqrt{d} + t\right)\right\} < 2e^{-t^2}$. Define $K = C\left(\sqrt{n} + \sqrt{d}\right)$. Then we have

$$\mathbb{P}\left\{\|Z\|_2 > K + t\right\} < 2e^{-t^2/C^2}\,. \tag{20}$$

Recall $\Gamma(z) = \int_0^\infty x^{z-1}e^{-x}dx$. Setting $t = C\sqrt{u}$ in the equation below yields

$$\int_0^\infty e^{-t^2/C^2}t^{p-1}dt \lesssim \int_0^\infty e^{-u}u^{\frac{p}{2}-1}du = \Gamma\left(\frac{p}{2}\right) \asymp 1\,.$$

Then we can bound the following integral

$$\int_K^\infty \mathbb{P}\left\{\|Z\|_2 \geq t\right\}pt^{p-1}dt$$

$$= \int_0^\infty \mathbb{P}\left\{\|Z\|_2 \geq K + t\right\}p\left(t + K\right)^{p-1}dt$$

$$\lesssim \int_0^\infty e^{-t^2/C^2}\left(t + K\right)^{p-1}dt$$

$$\lesssim \int_0^\infty e^{-t^2/C^2}\left(t^{p-1} + K^{p-1}\right)dt$$

$$= \int_0^\infty e^{-t^2/C^2}t^{p-1}dt + K^{p-1}\int_0^\infty e^{-t^2/C^2}dt$$

$$\lesssim n^{\frac{p-1}{2}}\,,$$

where the first inequality is because of Equation (20). We are in a position to bound $\mathbb{E}\|Z\|_2^p$:

$$\mathbb{E}\|Z\|_2^p$$

$$= \int_0^\infty \mathbb{P}\left\{\|Z\|_2^p \geq u\right\}du$$

$$= \int_0^\infty \mathbb{P}\left\{\|Z\|_2 \geq t\right\}pt^{p-1}dt$$

$$= \int_0^K \mathbb{P}\left\{\|Z\|_2 \geq t\right\}pt^{p-1}dt + \int_K^\infty \mathbb{P}\left\{\|Z\|_2 \geq t\right\}pt^{p-1}dt$$

$$\lesssim n^{p/2} + n^{(p-1)/2}$$

$$\lesssim n^{p/2}\,,$$

where the first inequality is because

$$\int_0^K \mathbb{P}\left\{\|Z\|_2 \geq t\right\}pt^{p-1}dt \leq \int_0^K pt^{p-1}dt = K^p \lesssim n^{p/2}\,.$$

$\qquad\qquad\qquad\qquad\qquad\qquad\qquad\qquad\qquad\qquad\qquad\qquad\qquad\qquad\qquad\qquad\qquad\square$

**Lemma 5.** *The following equation for the bias term $B_{\lambda,d,n}$ (defined in Equation (3)) holds*

$$B_{\lambda,d,n} = \mathbb{E}\left[\|\Lambda^{1/2}\left(I_d - \Lambda^{1/2}Z\left(n\lambda I_n + Z^\top\Lambda Z\right)^{-1}Z^\top\Lambda^{1/2}\right)\theta'\|_2^2\right] \tag{21}$$

$$= \mathbb{E}\left[\|\Lambda^{1/2}\left(I_d + \frac{1}{n\lambda}\Lambda^{1/2}ZZ^\top\Lambda^{1/2}\right)^{-1}\theta'\|_2^2\right]\,. \tag{22}$$

*Moreover, we have $|B_{\lambda,d,n}| \lesssim \|\theta^*\|_2^2$ and $\lim_{\lambda\to 0^+} B_{\lambda,d,n} = B_{0,d,n}$. For all sufficiently large $n$ and $d$ such that $n/d \to \gamma \in (0,1)$, we have $0 \leq \frac{d}{d\lambda}B_{\lambda,d,n} \lesssim \|\theta^*\|_2^2$. Therefore, $\{B_{\lambda,d,n}\}$ is uniformly bounded and uniformly equicontinuous with respect to $\lambda \in (0,\infty)$.*

*Proof.* Introduce the shorthand notation $M = \Lambda^{1/2} Z Z^\top \Lambda^{1/2} \in \mathbb{R}^{d \times d}$, $A = I_d + \frac{1}{n\lambda} M \in \mathbb{R}^{d \times d}$, $N = n\lambda I_n + Z^\top \Lambda Z \in \mathbb{R}^{n \times n}$, and $Q = I_d - \Lambda^{1/2} Z N^{-1} Z^\top \Lambda^{1/2} \in \mathbb{R}^{d \times d}$. Because

$$X^\top (n\lambda I_n + X X^\top)^{-1} X = P \Lambda^{1/2} Z \left(n\lambda I_n + Z^\top \Lambda Z\right)^{-1} Z^\top \Lambda^{1/2} P^\top,$$

we have

$$
\begin{aligned}
B_{\lambda, d, n} &= \mathbb{E}\left[ \|(I_d - P\Lambda^{1/2} Z \left(n\lambda I_n + Z^\top \Lambda Z\right)^{-1} Z^\top \Lambda^{1/2} P^\top) \theta^*\|_{P\Lambda P^\top}^2 \right] \\
&= \mathbb{E}\left[ \|\Lambda^{1/2} \left(I_d - \Lambda^{1/2} Z \left(n\lambda I_n + Z^\top \Lambda Z\right)^{-1} Z^\top \Lambda^{1/2}\right) \theta'\|_2^2 \right] \\
&= \mathbb{E}\left[ \|\Lambda^{1/2} Q \theta'\|_2^2 \right].
\end{aligned}
$$

Using the Sherman–Morrison-Woodbury formula yields

$$
\begin{aligned}
N^{-1} &= \frac{1}{n\lambda} I_n - \frac{1}{(n\lambda)^2} Z^\top \Lambda^{1/2} \left(I + \frac{1}{n\lambda} \Lambda^{1/2} Z Z^\top \Lambda^{1/2}\right)^{-1} \Lambda^{1/2} Z \\
&= \frac{1}{n\lambda} \left(I_n - Z^\top \Lambda^{1/2} \left(n\lambda I_d + \Lambda^{1/2} Z Z^\top \Lambda^{1/2}\right)^{-1} \Lambda^{1/2} Z\right) \\
&= \frac{1}{n\lambda} \left(I_n - Z^\top \Lambda^{1/2} \left(n\lambda I_d + M\right)^{-1} \Lambda^{1/2} Z\right).
\end{aligned} \tag{23}
$$

It follows that

$$
\begin{aligned}
Q &= I_d - \Lambda^{1/2} Z N^{-1} Z^\top \Lambda^{1/2} \\
&= I_d - \frac{1}{n\lambda} \Lambda^{1/2} Z \left(I_n - Z^\top \Lambda^{1/2} \left(n\lambda I_d + M\right)^{-1} \Lambda^{1/2} Z\right) Z^\top \Lambda^{1/2} \\
&= I_d - \frac{M}{n\lambda} \left(I_d - \left(n\lambda I_d + M\right)^{-1} M\right) \\
&= I_d - \frac{M}{n\lambda} \left(I_d - \left(n\lambda I_d + M\right)^{-1} \left(n\lambda I_d + M - n\lambda I_d\right)\right) \\
&= I_d - M \left(n\lambda I_d + M\right)^{-1} \\
&= \left(I_d + \frac{1}{n\lambda} M\right)^{-1} \\
&= A^{-1}.
\end{aligned}
$$

Therefore, we deduce

$$B_{\lambda, d, n} = \mathbb{E}\left[ \|\Lambda^{1/2} \left(I_d + \frac{1}{n\lambda} M\right)^{-1} \theta'\|_2^2 \right] = \mathbb{E}\left[ \|\Lambda^{1/2} A^{-1} \theta'\|_2^2 \right].$$

Because $\left\|\Lambda^{1/2}\right\|_2 \lesssim 1$ and $\left\|\left(I_d + \frac{1}{n\lambda} \Lambda^{1/2} Z Z^\top \Lambda^{1/2}\right)^{-1}\right\|_2 \leq 1$, we have

$$\|\Lambda^{1/2} A^{-1} \theta'\|_2^2 \lesssim \|\theta'\|_2^2 = \|\theta^*\|_2^2.$$

Therefore $|B_{\lambda, d, n}| \lesssim \|\theta^*\|_2^2$. Moreover, by the dominated convergence theorem,

$$\lim_{\lambda \to 0^+} B_{\lambda, d, n} = B_{0, d, n}.$$

We compute the derivative of $A^{-1}$:

$$\frac{dA^{-1}}{d\lambda} = -A^{-1} \frac{dA}{d\lambda} A^{-1} = \frac{M A^{-2}}{n\lambda^2}.$$

The matrix $\frac{M}{n} = \frac{\Lambda^{1/2} Z Z^\top \Lambda^{1/2}}{n} \in \mathbb{R}^{d \times d}$ is positive semidefinite and its $d - n$ smallest eigenvalues are zeros. Its non-zero eigenvalues are the same as the non-zero eigenvalues of $\frac{Z^\top \Lambda Z}{n}$. Because all eigenvalues of $\frac{Z^\top \Lambda Z}{n}$ are positive almost surely, the spectrum of $\frac{M}{n}$ consists of $d - n$ zeros and the

spectrum of $\frac{Z^\top \Lambda Z}{n}$. We study the range of the spectrum of $\frac{Z^\top \Lambda Z}{n}$. Because $\lambda_- \frac{Z^\top Z}{n} \preccurlyeq \frac{Z^\top \Lambda Z}{n} \preccurlyeq \lambda_+ \frac{Z^\top Z}{n}$, we deduce

$$\lambda_{\min}\left(\frac{Z^\top \Lambda Z}{n}\right) \geq \lambda_- \lambda_{\min}\left(\frac{Z^\top Z}{n}\right) \to \lambda_-\left(1 - \sqrt{1/\gamma}\right)^2 \tag{24}$$

$$\lambda_{\max}\left(\frac{Z^\top \Lambda Z}{n}\right) \leq \lambda_+ \lambda_{\max}\left(\frac{Z^\top Z}{n}\right) \to \lambda_+\left(1 + \sqrt{1/\gamma}\right)^2. \tag{25}$$

Define $L_1 = \lambda_- \lambda_{\min}\left(\frac{Z^\top Z}{n}\right)$ and $L_2 = \lambda_+ \lambda_{\max}\left(\frac{Z^\top Z}{n}\right)$. We get $\lim_{\substack{n,d\to+\infty \\ n/d\to\gamma<1}} L_1 = \lambda_-\left(1 - \sqrt{1/\gamma}\right)^2$, $\lim_{\substack{n,d\to+\infty \\ n/d\to\gamma<1}} L_2 = \lambda_+\left(1 + \sqrt{1/\gamma}\right)^2$ and

$$\mathrm{spec}\left(\frac{Z^\top \Lambda Z}{n}\right) \subseteq [L_1, L_2].$$

We bound $\left\|MA^{-3}\right\|_2$

$$\begin{aligned}\left\|MA^{-3}\right\|_2 &= n\left\|\frac{M}{n}\left(I_d + \frac{M}{n\lambda}\right)^{-3}\right\|_2 \\ &= n \max_{s\in\mathrm{spec}\left(\frac{M}{n}\right)} \frac{s}{(1+s/\lambda)^3} \\ &= n \max_{s\in\{0\}\cup\mathrm{spec}\left(\frac{Z^\top \Lambda Z}{n}.\right)} \frac{s}{(1+s/\lambda)^3} \\ &= n \max_{s\in\mathrm{spec}\left(\frac{Z^\top \Lambda Z}{n}.\right)} \frac{s}{(1+s/\lambda)^3} \\ &\leq n \max_{s\in[L_1,L_2]} \frac{s}{(1+s/\lambda)^3}.\end{aligned}$$

We compute $\frac{d}{d\lambda}\|\Lambda^{1/2}A^{-1}\theta'\|_2^2$:

$$\begin{aligned}&\frac{d}{d\lambda}\|\Lambda^{1/2}A^{-1}\theta'\|_2^2 \\ &= \frac{1}{n\lambda^2}\theta'^\top\left(A^{-1}\Lambda MA^{-2} + MA^{-2}\Lambda A^{-1}\right)\theta' \\ &= \frac{1}{n\lambda^2}\left(A^{-1}\theta'\right)^\top\left(\Lambda MA^{-1} + MA^{-1}\Lambda\right)\left(A^{-1}\theta'\right)\end{aligned}$$

Next, we bound $\left|\frac{d}{d\lambda}\|\Lambda^{1/2}A^{-1}\theta'\|_2^2\right|$:

$$\begin{aligned}&\left|\frac{d}{d\lambda}B_{\lambda,d,n}\right| \\ &\leq \frac{1}{n\lambda^2}\left\|MA^{-2}\Lambda A^{-1} + A^{-1}\Lambda MA^{-2}\right\|_2 \|\theta'\|_2^2 \\ &\leq \frac{2}{n\lambda^2}\left\|MA^{-2}\Lambda A^{-1}\right\|_2 \|\theta'\|_2^2 \\ &= \frac{2}{n\lambda^2}\left\|MA^{-3}A\Lambda A^{-1}\right\|_2 \|\theta'\|_2^2 \\ &\leq \frac{2}{n\lambda^2}\left\|MA^{-3}\right\|_2\left\|A\Lambda A^{-1}\right\|_2 \|\theta'\|_2^2 \\ &\lesssim \frac{1}{\lambda^2}\max_{s\in[L_1,L_2]}\frac{s}{(1+s/\lambda)^3}\|\theta'\|_2^2,\end{aligned}$$

where the last inequality is because $\left\|A\Lambda A^{-1}\right\|_2 = \|\Lambda\|_2 \leq \lambda_+ \lesssim 1$. Define $f(s) = \frac{s}{(1+s/\lambda)^3}$. Because $f'(s) = \frac{\lambda^3(\lambda-2s)}{(\lambda+s)^4}$, the function $f$ is increasing on $[0, \lambda/2]$ and decreasing on $[\lambda/2, +\infty)$.

If $\lambda \leq 2L_1$, we have

$$\max_{s \in [L_1, L_2]} \frac{s}{(1 + s/\lambda)^3} = \frac{L_1}{(1 + L_1/\lambda)^3} \, .$$

It follows that

$$\frac{1}{\lambda^2} \cdot \frac{L_1}{(1 + L_1/\lambda)^3} = \frac{L_1 \lambda}{(\lambda + L_1)^3} \leq \max_{\lambda \in [0, 2L_1]} \frac{L_1 \lambda}{(\lambda + L_1)^3} \lesssim \frac{1}{L_1} \, .$$

If $\lambda \geq 2L_2$, we get

$$\frac{1}{\lambda^2} \max_{s \in [L_1, L_2]} \frac{s}{(1 + s/\lambda)^3} = \frac{1}{\lambda^2} \cdot \frac{L_2}{(1 + L_2/\lambda)^3} \leq \max_{\lambda \in [2L_2, \infty)} \frac{L_2 \lambda}{(\lambda + L_2)^3} \lesssim \frac{1}{L_2} \leq \frac{1}{L_1} \, .$$

If $2L_1 < \lambda < 2L_2$, we obtain

$$\frac{1}{\lambda^2} \max_{s \in [L_1, L_2]} \frac{s}{(1 + s/\lambda)^3} \lesssim \frac{1}{\lambda} \lesssim \frac{1}{L_1} \, .$$

In all three cases, we show that $\frac{1}{\lambda^2} \max_{s \in [L_1, L_2]} \frac{s}{(1 + s/\lambda)^3} \lesssim \frac{1}{L_1}$. It follows that

$$\left| \frac{d}{d\lambda} \| \Lambda^{1/2} A^{-1} \theta' \|_2^2 \right| \lesssim \frac{1}{L_1} \| \theta' \|_2^2 = \frac{\| \theta' \|_2^2}{\lambda_- \lambda_{\min} \left( \frac{Z^\top Z}{n} \right)} \asymp \frac{\| \theta' \|_2^2}{\lambda_{\min} \left( \frac{Z^\top Z}{n} \right)} \, .$$

By Lemma 3, there exists a universal constant $n_0$ such that for all $n > n_0$, one has $\frac{1}{\lambda_{\min} \left( \frac{Z^\top Z}{n} \right)} \lesssim 1$.

Thus we conclude that

$$\left| \frac{d}{d\lambda} \| \Lambda^{1/2} A^{-1} \theta' \|_2^2 \right| \lesssim \| \theta' \|_2^2 \, .$$

We can exchange differentiation and expectation and get

$$\frac{d}{d\lambda} B_{\lambda, d, n} = \mathbb{E} \left[ \frac{d}{d\lambda} \| \Lambda^{1/2} A^{-1} \theta' \|_2^2 \right]$$

and

$$\left| \frac{d}{d\lambda} B_{\lambda, d, n} \right| = \mathbb{E} \left[ \left| \frac{d}{d\lambda} \| \Lambda^{1/2} A^{-1} \theta' \|_2^2 \right| \right] \lesssim \| \theta' \|_2^2 \, .$$

$\square$

**Lemma 6.** *The following equation for the variance term holds*

$$V_{\lambda, d, n} = \sigma^2 \mathbb{E} \| \Lambda^{1/2} \left( \lambda n I_d + \Lambda^{1/2} Z Z^\top \Lambda^{1/2} \right)^{-1} \Lambda^{1/2} Z \|_2^2$$

$$= \sigma^2 \mathbb{E} \| \Lambda Z \left( \lambda n I_n + Z^\top \Lambda Z \right)^{-1} \|_2^2 \, .$$

*Moreover, for all sufficiently large $n$ and $d$ such that $n/d \to \gamma \neq 1$, we have $\lim_{\lambda \to 0^+} V_{\lambda, d, n} = V_{0, d, n}$, $|V_{\lambda, d, n}| \lesssim 1$ and $\left| \frac{d}{d\lambda} V_{\lambda, d, n} \right| \lesssim 1$. Therefore, $\{V_{\lambda, d, n}\}$ is uniformly bounded and uniformly equicontinuous with respect to $\lambda \in (0, \infty)$.*

*Proof.* As in the proof of Lemma 5, define $M = \Lambda^{1/2} Z Z^\top \Lambda^{1/2} \in \mathbb{R}^{d \times d}$ and $N = n\lambda I_n + Z^\top \Lambda Z \in \mathbb{R}^{n \times n}$. Recalling $\Sigma = P \Lambda P^\top$ and $X = Z^\top \Lambda^{1/2} P^\top$, we have

$$V_{\lambda, d, n} = \sigma^2 \mathbb{E} \operatorname{tr} \left[ X \Sigma X^\top (n\lambda I_n + X X^\top)^{-2} \right]$$

$$= \sigma^2 \mathbb{E} \operatorname{tr} \left[ Z^\top \Lambda^2 Z N^{-2} \right]$$

$$= \sigma^2 \mathbb{E} \operatorname{tr} \left[ N^{-1} Z^\top \Lambda^2 Z N^{-1} \right]$$

$$= \sigma^2 \mathbb{E} \left\| \Lambda Z N^{-1} \right\|_F^2 \, .$$

Recalling Equation ([23]) yields

$$
\begin{aligned}
\Lambda Z N^{-1} &= \frac{1}{n\lambda} \Lambda Z \left( I_n - Z^\top \Lambda^{1/2} \left( n\lambda I_d + M \right)^{-1} \Lambda^{1/2} Z \right) \\
&= \frac{1}{n\lambda} \Lambda^{1/2} \left( I_n - M \left( n\lambda I_d + M \right)^{-1} \right) \Lambda^{1/2} Z \\
&= \Lambda^{1/2} \left( n\lambda I_d + M \right)^{-1} \Lambda^{1/2} Z \,.
\end{aligned}
$$

Define $R = \Lambda^{1/2} Z \in \mathbb{R}^{d \times n}$. We get

$$
\left( n\lambda I_d + M \right)^{-1} \Lambda^{1/2} Z = \left( n\lambda I_d + RR^\top \right)^{-1} R \,.
$$

Notice that if $0 < a < b$, then $a I_d + RR^\top \preccurlyeq b I_d + RR^\top$. We deduce

$$
\left( b I_d + RR^\top \right)^2 - \left( a I_d + RR^\top \right)^2 = \left( b^2 - a^2 \right) I_d + 2 \left( b - a \right) RR^\top \succcurlyeq 0 \,.
$$

Thus $\left( b I_d + RR^\top \right)^2 \succcurlyeq \left( a I_d + RR^\top \right)^2$, which implies $\left( b I_d + RR^\top \right)^{-2} \preccurlyeq \left( a I_d + RR^\top \right)^{-2}$. We get

$$
\begin{aligned}
R^\top \left( b I_d + RR^\top \right)^{-2} R &\preccurlyeq R^\top \left( a I_d + RR^\top \right)^{-2} R \,, \\
\mathrm{tr} \left( R^\top \left( b I_d + RR^\top \right)^{-2} R \right) &\leq \mathrm{tr} \left( R^\top \left( a I_d + RR^\top \right)^{-2} R \right)
\end{aligned}
$$

Let $\lambda_0 \left( \cdot \right)$ denote the smallest non-zero eigenvalue of a positive semidefinite matrix. We bound the Frobenius norm

$$
\begin{aligned}
&\left\| \left( n\lambda I_d + M \right)^{-1} \Lambda^{1/2} Z \right\|_F^2 \\
&= \mathrm{tr} \left( R^\top \left( n\lambda I_d + RR^\top \right)^{-1} R \right) \\
&\leq \mathrm{tr} \left( \lim_{\lambda \to 0^+} R^\top \left( n\lambda I_d + RR^\top \right)^{-2} R \right) \\
&= \mathrm{tr} \left( R^\top R \right)^+ \\
&= \mathrm{tr} \left( Z^\top \Lambda Z \right)^+ \\
&\lesssim \mathrm{tr} \left( Z^\top Z \right)^+ \,.
\end{aligned}
$$

It follows that

$$
\left\| \Lambda Z N^{-1} \right\|_F^2 = \left\| \Lambda^{1/2} \left( n\lambda I_d + M \right)^{-1} \Lambda^{1/2} Z \right\|_F^2 \lesssim \left\| \left( n\lambda I_d + M \right)^{-1} \Lambda^{1/2} Z \right\|_F^2 \lesssim \mathrm{tr} \left( Z^\top Z \right)^+ = \mathrm{tr} \left( ZZ^\top \right)^+ \,.
$$

If $n/d \to \gamma < 1$, the matrix $Z^\top Z$ is full-rank almost surely. Then, using the formula for the mean of inverse Wishart distribution, we have $\mathbb{E} \, \mathrm{tr} \left( Z^\top Z \right)^+ = \mathrm{tr} \, \mathbb{E} \left( Z^\top Z \right)^{-1} = \mathrm{tr} \left( \frac{I_n}{d - n - 1} \right) \asymp 1$. If $n/d \to \gamma > 1$, the matrix $ZZ^\top$ is full-rank almost surely. Similarly, we have $\mathbb{E} \, \mathrm{tr} \left( ZZ^\top \right)^+ = \mathrm{tr} \, \mathbb{E} \left( ZZ^\top \right)^{-1} \asymp 1$. By the dominated convergence theorem, we have $\lim_{\lambda \to 0^+} V_{\lambda, d, n} = V_{0, d, n}$. Moreover, $V_{\lambda, d, n} \lesssim \mathbb{E} \left\| \Lambda Z N^{-1} \right\|_F^2 \lesssim 1$.

Next we bound $\frac{d}{d\lambda} V(\hat{\theta})$. Because $\frac{dN^{-1}}{d\lambda} = -N^{-1} \frac{dN}{d\lambda} N^{-1} = -n N^{-2}$, we deduce

$$
\frac{d}{d\lambda} \left\| \Lambda Z N^{-1} \right\|_2^2 = \frac{d}{d\lambda} \, \mathrm{tr} \left( N^{-1} Z^\top \Lambda^2 Z N^{-1} \right) = -2n \, \mathrm{tr} \left( Z^\top \Lambda^2 Z N^{-3} \right) \leq 0 \,.
$$

On the other hand, we have

$$
\begin{aligned}
&\mathrm{tr} \left( Z^\top \Lambda^2 Z N^{-3} \right) \\
&= \mathrm{tr} \left( N^{-3/2} Z^\top \Lambda^2 Z N^{-3/2} \right) \\
&\lesssim \mathrm{tr} \left( N^{-3/2} Z^\top \Lambda Z N^{-3/2} \right) \\
&= \mathrm{tr} \left( Z^\top \Lambda Z N^{-3} \right) \\
&= \sum_{s \in \mathrm{spec}(Z^\top \Lambda Z)} \frac{s}{\left( \lambda n + s \right)^3} \,.
\end{aligned}
$$

Because the number of non-zero eigenvalues of $Z^\top \Lambda Z$ equals $\mathrm{rank}\left(Z^\top \Lambda Z\right) = n \wedge d \asymp n$, we get

$$\left|\frac{d}{d\lambda}\left\|\Lambda Z N^{-1}\right\|_2^2\right| \asymp n\,\mathrm{tr}\left(Z^\top \Lambda^2 Z N^{-3}\right) \lesssim n^2 \max_{s \in \mathrm{spec}(Z^\top \Lambda Z)} \frac{s}{(\lambda n + s)^3} = \max_{s \in \mathrm{spec}\left(\frac{Z^\top \Lambda Z}{n}\right) \setminus \{0\}} \frac{s}{(\lambda + s)^3}\,.$$

If $\gamma < 1$, the matrix $\frac{Z^\top \Lambda Z}{n}$ is full-rank almost surely. By Equation (24) and Equation (25) in the proof of Lemma 5, there exists universal positive constants $C_1$ and $C_2$ such that $\mathrm{spec}\left(\frac{Z^\top \Lambda Z}{n}\right) \subseteq [C_1, C_2]$ for all sufficiently large $n$ and $d$ such that $n/d \to \gamma < 1$. If $\gamma > 1$, the non-zero eigenvalues of $\frac{Z^\top \Lambda Z}{n}$ and $\frac{M}{n}$ are the same. The matrix $\frac{M}{n}$ is full-rank almost surely. Thus $\mathrm{spec}\left(\frac{Z^\top \Lambda Z}{n}\right) \setminus \{0\} = \mathrm{spec}\left(\frac{M}{n}\right)$. Because $y^\top \Lambda^{-1} y \lesssim y^\top y$,

$$\lambda_{\min}\left(\frac{M}{n}\right) = \min_{x \neq 0} \frac{x^\top \frac{\Lambda^{1/2} Z Z^\top \Lambda^{1/2}}{n} x}{x^\top x} = \min_{y \neq 0} \frac{y^\top \frac{ZZ^\top}{n} y}{y^\top \Lambda^{-1} y} \gtrsim \min_{y \neq 0} \frac{y^\top \frac{ZZ^\top}{n} y}{y^\top y} = \lambda_{\min}\left(\frac{ZZ^\top}{n}\right)\,.$$

Similarly, we get

$$\lambda_{\max}\left(\frac{M}{n}\right) \lesssim \lambda_{\max}\left(\frac{ZZ^\top}{n}\right)\,.$$

Therefore, there exists universal positive constants $C_1$ and $C_2$ such that

$$\mathrm{spec}\left(\frac{Z^\top \Lambda Z}{n}\right) \setminus \{0\} \subseteq \left[C_1 \lambda_{\min}\left(\frac{Z^\top Z}{n}\right), C_2 \lambda_{\max}\left(\frac{Z^\top Z}{n}\right)\right]\,.$$

Thus in both cases, we have shown that there exists universal positive constants $C_1$ and $C_2$ such that

$$\mathrm{spec}\left(\frac{Z^\top \Lambda Z}{n}\right) \setminus \{0\} \subseteq \left[C_1 \lambda_{\min}\left(\frac{Z^\top Z}{n}\right), C_2 \lambda_{\max}\left(\frac{Z^\top Z}{n}\right)\right]\,.$$

Define $L_1 = C_1 \lambda_{\min}\left(\frac{Z^\top Z}{n}\right)$ and $L_2 = C_2 \lambda_{\max}\left(\frac{Z^\top Z}{n}\right)$. As a result, we get

$$\left|\frac{d}{d\lambda}\left\|\Lambda Z N^{-1}\right\|_2^2\right| \lesssim \max_{s \in [L_1, L_2]} \frac{s}{(\lambda + s)^3}\,.$$

Define $f(s) = \frac{s}{(\lambda + s)^3}$. Because $f'(s) = \frac{\lambda - 2s}{(\lambda + s)^4}$, the function $f$ is increasing on $[0, \lambda/2]$ and decreasing on $[\lambda/2, +\infty)$. If $\lambda \geq 2L_2$ or $\lambda \leq 2L_1$, we get

$$\max_{s \in [L_1, L_2]} \frac{s}{(\lambda + s)^3} \leq \frac{L_1}{(\lambda + L_1)^3} \vee \frac{L_2}{(\lambda + L_2)^3} \leq \frac{1}{L_1^2}\,.$$

If $2L_1 < \lambda < 2L_2$, we get

$$\max_{s \in [L_1, L_2]} \frac{s}{(\lambda + s)^3} \lesssim \frac{1}{\lambda^2} \lesssim \frac{1}{L_1^2}\,.$$

As a result, for all sufficiently large $n$, we have

$$\left|\frac{d}{d\lambda}\left\|\Lambda Z N^{-1}\right\|_2^2\right| = \max_{s \in [L_1, L_2]} \frac{s}{(\lambda + s)^3} \lesssim \frac{1}{L_1^2} \lesssim \frac{1}{\lambda_{\min}^2\left(\frac{Z^\top Z}{n}\right)} \lesssim 1\,,$$

where the final inequality is because of Lemma 3. We can exchange the expectation and differentiation and obtain

$$\frac{d}{d\lambda} V_{\lambda,d,n} = \sigma^2 \mathbb{E}\frac{d}{d\lambda}\left\|\Lambda Z N^{-1}\right\|_2^2$$

and

$$\left|\frac{d}{d\lambda} V_{\lambda,d,n}\right| \leq \sigma^2 \mathbb{E}\left|\frac{d}{d\lambda}\left\|\Lambda Z N^{-1}\right\|_2^2\right| \lesssim 1\,.$$

$\square$

## C  LEMMAS ON STIELTJES TRANSFORM

**Definition 3** (Stieltjes transform). The Stieltjes transform of a distribution with cumulative distribution function $F$ is defined by

$$s_F(z) = \int \frac{1}{\lambda - z} dF(\lambda) \quad (z \in \mathcal{H} \triangleq \{z \in \mathbb{C} \mid \Im z > 0\}).$$

**Lemma 7** (Theorem 4.3 (Bai & Silverstein, 2010))**.** *Suppose that the entries of $X_n \in \mathbb{C}^{n \times p}$ are complex random variables that are independent for each $n$ and identically distributed for all $n$ and satisfy $\mathbb{E}\left[|x_{11} - \mathbb{E}x_{11}|^2\right] = 1$. Also, assume that $T_n = \operatorname{diag}(\tau_1, \ldots, \tau_p)$, $\tau_i$ is real, and the empirical distribution function of $\{\tau_1, \ldots, \tau_p\}$ converges almost surely to a probability distribution function $H$ as $n \to \infty$. The entries of both $X_n$ and $T_n$ may depend on $n$, which is suppressed for brevity. Set $B_n = A_n + \frac{1}{n}X_n T_n X_n^*$, where $X_n^*$ is the conjugate transpose of $X_n$, $A_n$ is Hermitian, $n \times n$ satisfying $F^{A_n} \to F^A$ almost surely, where $F^A$ is a distribution function (possibly defective) on the real line. Assume also that $X_n$, $T_n$, and $A_n$ are independent. When $p = p(n)$ with $p/n \to y > 0$ as $n \to \infty$, then, almost surely, $F^{B_n}$, the empirical spectral distribution of the eigenvalues of $B_n$, converges vaguely, as $n \to \infty$, to a (nonrandom) distribution function $F$, where for any $z \in \mathbb{C}^+ = \{z \in \mathbb{C} \mid \Im z > 0\}$, its Stieltjes transform $s = s(z)$ is the unique solution in $\mathbb{C}^+$ to the equation*

$$s = s_A\left(z - y \int \frac{\tau dH(\tau)}{1 + \tau s}\right),$$

*where $s_A$ is the Stieltjes transform of $F^A$.*

**Lemma 8.** *If the functions $f_\alpha, g_\alpha : I \to \mathbb{R}$ satisfy $f_\alpha(x) - g_\alpha(x) \to 0$ uniformly as $\alpha \to +\infty$, then $\lim_{\alpha \to +\infty}\left(\inf_{x \in I} f(x) - \inf_{x \in I} g(x)\right) = 0$.*

*Proof.* Because $f_\alpha(x) - g_\alpha(x) \to 0$ uniformly as $\alpha \to +\infty$, we have for $\forall \epsilon > 0$, there exists $N(\epsilon)$ such that for $\forall \alpha > N(\epsilon)$ and $\forall x \in I$, it holds that $|f_\alpha(x) - g_\alpha(x)| < \epsilon$. Therefore, we get

$$g_\alpha(x) - \epsilon < f_\alpha(x) < g_\alpha(x) + \epsilon.$$

Thus we obtain

$$\inf_{x \in I} f_\alpha(x) \leq f_\alpha(x) < g_\alpha(x) + \epsilon$$
$$\inf_{x \in I} g_\alpha(x) - \epsilon \leq g_\alpha(x) - \epsilon < f_\alpha(x),$$

which in turn implies

$$\inf_{x \in I} f_\alpha(x) \leq \inf_{x \in I} g_\alpha(x) + \epsilon$$
$$\inf_{x \in I} g_\alpha(x) - \epsilon \leq \inf_{x \in I} f_\alpha(x).$$

It follows that $|\inf_{x \in I} f_\alpha(x) - \inf_{x \in I} g_\alpha(x)| \leq \epsilon$. In other words, we proved

$$\lim_{\alpha \to +\infty}\left(\inf_{x \in I} f(x) - \inf_{x \in I} g(x)\right) = 0.$$

$\square$

**Lemma 9.** *Define $N = \lambda n I_n + Z^\top \Lambda Z$. Then we have*

$$\lim_{\substack{n, d_i \to +\infty \\ d_i/n \to z_i}} \operatorname{tr}\left(N^{-1}\right) = \frac{d}{d\lambda} \inf_{\rho \in \mathbb{R}_+^m}\left[\log\left(\lambda + \sum_{i \in [m]} \lambda_i \rho_i\right) + \sum_{i \in [m]}\left(\rho_i - z_i\left(\log\frac{\rho_i}{z_i} + 1\right)\right)\right],$$
(26)

$$\lim_{\substack{n, d_i \to +\infty \\ d_i/n \to z_i}} \frac{1}{n}\log\det\frac{N}{n} = \inf_{\rho \in \mathbb{R}_+^m}\left[\log\left(\lambda + \sum_{i \in [m]} \lambda_i \rho_i\right) + \sum_{i \in [m]}\left(\rho_i - z_i\left(\log\frac{\rho_i}{z_i} + 1\right)\right)\right].$$
(27)

*Proof.* **Proof of Equation (26).** We apply Lemma 7 with $A_n = \mathbf{0}_{n \times n}$, $X_n = Z^\top \in \mathbb{R}^{n \times d}$, $T_n = \Lambda$, and $B_n = \frac{1}{n} Z^\top \Lambda Z$. The distribution function of $\mathbf{0}_{n \times n}$ converges to $\mathbf{1}_{t \leq 0}$ and its Stieltjes transform is $s_A(z) = \int \frac{1}{\lambda - z} d\mathbf{1}_{\lambda \leq 0} = -\frac{1}{z}$. The empirical distribution function of $\{\underbrace{\lambda_1, \ldots, \lambda_1}_{d_1}, \ldots, \underbrace{\lambda_m \ldots, \lambda_m}_{d_m}\}$ is $H_{n, d_i}(t) = \sum_{i \in [m]} \frac{d_i}{d} \mathbf{1}_{t \leq \lambda_i}$. Recall $d_i/n \to z_i$. Thus $d_i/d \to z_i/K$, where $d/n \to y = \sum_{j \in [m]} z_j$. The empirical distribution function converges to $H(t) = \sum_{i \in [m]} \frac{z_i}{y} \mathbf{1}_{t \leq \lambda_i}$. Then the empirical spectral distribution of the eigenvalues of $\frac{1}{n} Z^\top \Lambda Z$ converges vaguely to a nonrandom distribution function $F$ and its Stieltjes transform is

$$s = s(z) = \lim_{\substack{n, d_i \to +\infty \\ d_i/n = z_i}} \frac{1}{n} \operatorname{tr} \left( \frac{1}{n} Z^\top \Lambda Z - z I_n \right)^{-1} = \lim_{\substack{n, d_i \to +\infty \\ d_i/n = z_i}} \operatorname{tr} \left( Z^\top \Lambda Z - z n I_n \right)^{-1}$$

(this is because of (Bai & Silverstein, 2010, Theorem B.9)). By Lemma 7, $s(z)$ is the unique solution in $\mathbb{C}^+$ to the equation

$$s(z) = s_A \left( z - y \int \frac{\tau dH(\tau)}{1 + \tau s} \right) = -\frac{1}{z - \sum_{i \in [m]} \frac{\lambda_i z_i}{1 + \lambda_i s(z)}},$$

which gives

$$s(z) \left( z - \sum_{i \in [m]} \frac{\lambda_i z_i}{1 + \lambda_i s(z)} \right) = -1.$$

We want to prove Equation (26) first. The lefthand side of Equation (26) equals

$$\lim_{\substack{n, d_i \to +\infty \\ d_i/n = z_i}} \operatorname{tr} \left( \lambda n I_n + Z^\top \Lambda Z \right)^{-1} = s(-\lambda).$$

Because the matrix $\frac{1}{n} Z^\top \Lambda Z$ is positive semidefinite and thereby all of its eigenvalues are non-negative, its limiting spectral distribution is supported on $[0, \infty)$. The Stieltjes transform $s(z)$ of the limiting spectral distribution can be continuously extended to $(-\infty, 0)$. Therefore, for $\forall \lambda > 0$, $s(-\lambda)$ is the unique solution to the following equation

$$s(-\lambda) \left( \lambda + \sum_{i=1}^m \frac{\lambda_i z_i}{1 + \lambda_i s(-\lambda)} \right) = 1. \tag{28}$$

We will verify that

$$\frac{d}{d\lambda} \inf_{\rho \in \mathbb{R}_+^m} \left[ \log \left( \lambda + \sum_{j=1}^m \lambda_j \rho_j \right) + \sum_{j=1}^m \left( \rho_j - z_j (\log \frac{\rho_j}{z_j} + 1) \right) \right]$$

satisfies Equation (28). Take a minimizer $\rho^*$ of Equation (5). Using the envelope theorem yields

$$\frac{d}{d\lambda} \inf_{\rho \in \mathbb{R}_+^m} \left[ \log \left( \lambda + \sum_{j=1}^m \lambda_j \rho_j \right) + \sum_{j=1}^m \left( \rho_j - z_j (\log \frac{\rho_j}{z_j} + 1) \right) \right] = \frac{1}{\lambda + \sum_{j=1}^m \lambda_j \rho_j^*}. \tag{29}$$

Plugging the righthand side of Equation (29) into Equation (28), we get

$$\frac{1}{\lambda + \sum_{j=1}^m \lambda_j \rho_j^*} \left( \lambda + \sum_{i=1}^m \frac{\lambda_i z_i}{1 + \lambda_i \cdot \frac{1}{\lambda + \sum_{j=1}^m \lambda_j \rho_j^*}} \right) = 1.$$

Rewriting the above equation yields

$$\sum_{i=1}^m \frac{\lambda_i z_i}{1 + \lambda_i \cdot \frac{1}{\lambda + \sum_{j=1}^m \lambda_j \rho_j^*}} = \sum_{i=1}^m \lambda_i \rho_i^*.$$

It suffices to show that each summand on the lefthand side equals its counterpart on the righthand side

$$\frac{\lambda_i z_i}{1 + \lambda_i \cdot \frac{1}{\lambda + \sum_{j=1}^m \lambda_j \rho_j^*}} = \lambda_i \rho_i^*.$$

We need to show

$$\frac{z_i}{\rho_i^*} = 1 + \lambda_i \cdot \frac{1}{\lambda + \sum_{j=1}^m \lambda_j \rho_j^*} \,,$$

which is equivalent to Equation (6) and therefore holds. Hence we have proved Equation (26).

**Proof of Equation (27).** We use $\alpha$ to denote the indices $n, d_i$. Define

$$h(\lambda) = \inf_{\rho \in \mathbb{R}_+^m} \left[ \log \left( \lambda + \sum_{j=1}^m \lambda_j \rho_j \right) + \sum_{j=1}^m \left( \rho_j - z_j (\log \frac{\rho_j}{z_j} + 1) \right) \right] .$$

First, we want to show that $\lim_{\lambda_0 \to +\infty} (h(\lambda_0) - \log \lambda_0) = 0$. Define

$$l_{\lambda_0}(\rho) = \log \left( 1 + \frac{1}{\lambda_0} \sum_{j=1}^m \lambda_j \rho_j \right) + \sum_{j=1}^m \left( \rho_j - z_j \left( \log \frac{\rho_j}{z_j} + 1 \right) \right) ,$$

$$q(\rho) = \sum_{j=1}^m \left( \rho_j - z_j \left( \log \frac{\rho_j}{z_j} + 1 \right) \right) .$$

The Hessian matrix of $q(p)$ is $\mathrm{diag} \left( \frac{z_1}{\rho_1^2}, \dots, \frac{z_m}{\rho_m^2} \right)$, which is positive definite since $z_i, \rho_i > 0$. Therefore, $q(p)$ is convex and the minimum of $q(\rho)$ on $\mathbb{R}_+^m$ is attained at $\rho = \mathbf{z}$, where $\mathbf{z} = (z_1, \dots, z_m)^\top$. The minimum is $\inf_{\rho \in \mathbb{R}_+^m} q(\rho) = q(\mathbf{z}) = 0$. Because $\lim_{\|\rho\|_2 \to +\infty} l_{\lambda_0}(\rho) = +\infty$, there exists a universal constant $K_1 > \|\mathbf{z}\|_2 > 0$ such that $l_{\lambda_0}(\rho) > l_{\lambda_0}(\mathbf{z})$ for all $\|\rho\|_2 > K_1$. Define $E = \left\{ \rho \in \mathbb{R}_+^m \mid \|\rho\|_2 \le K_1 \right\}$. We have $\mathbf{z} \in E$, $\inf_{\rho \in E} l_{\lambda_0}(\rho) = \inf_{\rho \in \mathbb{R}_+^m} l_{\lambda_0}(\rho)$, and $\inf_{\rho \in E} q(\rho) = \inf_{\rho \in \mathbb{R}_+^m} q(\rho) = 0$. Therefore, we get

$$h(\lambda_0) - \log \lambda_0 = \inf_{\rho \in \mathbb{R}_+^m} l_{\lambda_0}(\rho) = \inf_{\rho \in E} l_{\lambda_0}(\rho) - \inf_{\rho \in E} q(\rho) \,. \tag{30}$$

On $E$, there exists a universal constant $K_2 > 0$ such that $\sum_{j \in [m]} \lambda_j \rho_j < K_2$. Thus on $E$, we deduce

$$0 < l_{\lambda_0}(\rho) - q(\rho) = \log \left( 1 + \frac{1}{\lambda_0} \sum_{j=1}^m \lambda_j \rho_j \right) < \log \left( 1 + \frac{K_2}{\lambda_0} \right) .$$

The right-hand side $\log \left( 1 + \frac{K_2}{\lambda_0} \right) \to 0$ as $\lambda_0 \to +\infty$. Thus $\lim_{\lambda_0 \to +\infty} (l_{\lambda_0}(\rho) - q(\rho)) = 0$ uniformly for $\rho \in E$. By Lemma 8, we get

$$\lim_{\lambda_0 \to +\infty} \left( \inf_{\rho \in E} l_{\lambda_0}(\rho) - \inf_{\rho \in E} q(\rho) \right) = 0 \,.$$

Recalling Equation (30) yields

$$\lim_{\lambda_0 \to +\infty} (h(\lambda_0) - \log \lambda_0) = 0 \,. \tag{31}$$

Define $f_\alpha(\lambda) = \frac{1}{n} \log \det \frac{N}{n}$. Second, we want to show $\lim_\alpha f_\alpha(\lambda) = h(\lambda)$, where $\lim_\alpha$ means $\lim_{\substack{n, d_i \to +\infty \\ d_i/n = z_i}}$. We have $f_\alpha(\lambda) - f_\alpha(\lambda_0) = \int_{\lambda_0}^\lambda f'_\alpha(x) dx$ for $\forall \lambda, \lambda_0 > 0$. It follows that

$$
\begin{aligned}
|f_\alpha(\lambda) - h(\lambda)| &\le |f_\alpha(\lambda) - h(\lambda) + h(\lambda_0) - f_\alpha(\lambda_0) + f_\alpha(\lambda_0) - \log \lambda_0 + \log \lambda_0 - h(\lambda_0)| \\
&\le |f_\alpha(\lambda) - h(\lambda) + h(\lambda_0) - f_\alpha(\lambda_0)| + |f_\alpha(\lambda_0) - \log \lambda_0| + |\log \lambda_0 - h(\lambda_0)| \\
&= \left| \int_{\lambda_0}^\lambda f'_\alpha(x) dx - (h(\lambda) - h(\lambda_0)) \right| + |f_\alpha(\lambda_0) - \log \lambda_0| + |\log \lambda_0 - h(\lambda_0)| \,.
\end{aligned}
$$

Taking $\limsup_\alpha$ on both sides gives

$$\limsup_\alpha |f_\alpha(\lambda) - h(\lambda)| \le \limsup_\alpha \left| \int_{\lambda_0}^\lambda f'_\alpha(x) dx - (h(\lambda) - h(\lambda_0)) \right| + \limsup_\alpha |f_\alpha(\lambda_0) - \log \lambda_0| + |\log \lambda_0 - h(\lambda_0)| \,. \tag{32}$$

Recall $f'_\alpha(\lambda) = \operatorname{tr} N^{-1}$ and $\lim_\alpha f'_\alpha(\lambda) = h'(\lambda)$ (this is exactly Equation (26)). Because $\left|\operatorname{tr} N^{-1}\right| = \operatorname{tr} N^{-1} \le \frac{1}{\lambda}$ and $\int_{\lambda_0}^{\lambda} \frac{1}{x} dx < +\infty$, by the dominated convergence theorem, we have

$$\lim_\alpha \int_{\lambda_0}^{\lambda} f'_\alpha(x)dx = \int_{\lambda_0}^{\lambda} h'(x)dx = h(\lambda) - h(\lambda_0).$$

It follows that

$$\limsup_\alpha \left| \int_{\lambda_0}^{\lambda} f'_\alpha(x)dx - (h(\lambda) - h(\lambda_0)) \right| = \lim_\alpha \left| \int_{\lambda_0}^{\lambda} f'_\alpha(x)dx - (h(\lambda) - h(\lambda_0)) \right| = 0. \quad (33)$$

Since

$$f_\alpha(\lambda_0) - \log \lambda_0 = \frac{1}{n} \log \det \left( \lambda_0 I_n + \frac{1}{n} Z^\top \Lambda Z \right) - \frac{1}{n} \log \det \left( \lambda_0 I_n \right) = \frac{1}{n} \log \det \left( I_n + \frac{1}{n\lambda_0} Z^\top \Lambda Z \right)$$

and the matrix $\frac{1}{n\lambda_0} Z^\top \Lambda Z$ is positive semidefinite, we have

$$f_\alpha(\lambda_0) - \log \lambda_0 \ge 0.$$

We have

$$f_\alpha(\lambda_0) - \log \lambda_0$$
$$= \frac{1}{n} \log \det \left( I_n + \frac{1}{n\lambda_0} Z^\top \Lambda Z \right)$$
$$\le \frac{1}{n} \log \det \left( I_n + \frac{\lambda_+}{n\lambda_0} Z^\top Z \right)$$
$$\le \log \left( 1 + \frac{\lambda_+}{\lambda_0} \lambda_{\max} \left( \frac{Z^\top Z}{n} \right) \right)$$
$$\le \frac{\lambda_+}{\lambda_0} \lambda_{\max} \left( \frac{Z^\top Z}{n} \right).$$

Then taking $\limsup_\alpha$, we get

$$\limsup_\alpha |f_\alpha(\lambda_0) - \log \lambda_0| = \limsup_\alpha (f_\alpha(\lambda_0) - \log \lambda_0) \le \frac{\lambda_+}{\lambda_0} \limsup_\alpha \lambda_{\max} \left( \frac{Z^\top Z}{n} \right) \lesssim \frac{1}{\lambda_0}, \quad (34)$$

where the last inequality is because $\limsup_\alpha \lambda_{\max} \left( \frac{Z^\top Z}{n} \right) = \left( 1 + \sqrt{\gamma \vee \frac{1}{\gamma}} \right)^2 \asymp 1$ by Lemma 1. Using Equation (32), Equation (33) and Equation (34) gives

$$\limsup_\alpha |f_\alpha(\lambda) - h(\lambda)| \lesssim \frac{1}{\lambda_0} + |\log \lambda_0 - h(\lambda_0)|.$$

Then taking $\lim_{\lambda_0 \to +\infty}$ and recalling Equation (31) yields

$$\lim_\alpha |f_\alpha(\lambda) - h(\lambda)| = \limsup_\alpha |f_\alpha(\lambda) - h(\lambda)| = 0.$$

Therefore, we conclude $\lim_\alpha f_\alpha(\lambda) = h(\lambda)$.

$\square$

**Lemma 10.** *Define* $N = \lambda n I_n + Z^\top \Lambda Z = \lambda n I_n + \sum_{i \in [m]} \lambda_i Z_i Z_i^\top$. *The following equation holds*

$$\lim_{\substack{n, d_i \to +\infty \\ d_i/n \to z_i}} \mathbb{E} \left[ \frac{\partial}{\partial \lambda_i} \frac{1}{n} \log \det \frac{N}{n} \right] = \frac{\partial}{\partial \lambda_i} \inf_{\rho \in \mathbb{R}_+^m} \left[ \log \left( \lambda + \sum_{i \in [m]} \lambda_i \rho_i \right) + \sum_{i \in [m]} \left( \rho_i - z_i \left( \log \frac{\rho_i}{z_i} + 1 \right) \right) \right], \quad (35)$$

$$\lim_{\substack{n, d_i \to +\infty \\ d_i/n \to z_i}} \mathbb{E} \left[ \frac{\partial^2}{\partial \lambda_j \partial \lambda_i} \frac{1}{n} \log \det \frac{N}{n} \right] = \frac{\partial^2}{\partial \lambda_j \partial \lambda_i} \inf_{\rho \in \mathbb{R}_+^m} \left[ \log \left( \lambda + \sum_{i \in [m]} \lambda_i \rho_i \right) + \sum_{i \in [m]} \left( \rho_i - z_i \left( \log \frac{\rho_i}{z_i} + 1 \right) \right) \right]. \quad (36)$$

*Proof.* **Proof of Equation (35).** We use $\alpha$ to denote the indices $n, d_i$ and use $\lim_\alpha$ to denote $\lim_{\substack{n,d_i \to +\infty \\ d_i/n = z_i}}$. Define $f_\alpha(\lambda_i) = \mathbb{E}\left[\frac{1}{n} \log \det \frac{N}{n}\right]$, $f'_\alpha(\lambda_i) = \frac{\partial}{\partial \lambda_i} \mathbb{E}\left[\frac{1}{n} \log \det \frac{N}{n}\right]$, and

$$h(\lambda_i) = \inf_{\rho \in \mathbb{R}_+^m} \left[ \log\left( \lambda + \sum_{i \in [m]} \lambda_i \rho_i \right) + \sum_{i \in [m]} \left( \rho_i - z_i \left( \log \frac{\rho_i}{z_i} + 1 \right) \right) \right].$$

We have

$$\left| \frac{1}{n} \log \det \frac{N}{n} \right| \le \frac{1}{n} \log \det \left( \lambda I_n + \lambda_+ \frac{Z^\top Z}{n} \right) = \log \lambda + \frac{1}{n} \log \det \left( I_n + \frac{\lambda_+}{n\lambda} Z^\top Z \right).$$

By Lemma 3, there exists a universal constant $C > 0$ such that for all sufficiently large $n$,

$$\frac{1}{n} \log \det \left( I_n + \frac{\lambda_+}{n\lambda} Z^\top Z \right) \le \log \left( 1 + \frac{C}{\lambda} \right).$$

Therefore, we get

$$\left| \frac{1}{n} \log \det \frac{N}{n} \right| \le \log \left( \lambda + C \right).$$

By the dominated convergence theorem and Lemma 9 (specifically, Equation (27)), we obtain

$$\lim_\alpha f_\alpha(\lambda_i) = h(\lambda_i). \tag{37}$$

Because

$$\left| \frac{\partial}{\partial \lambda_i} \frac{1}{n} \log \det \frac{N}{n} \right| = \frac{1}{n} \operatorname{tr}\left( Z_i^\top N^{-1} Z_i \right) \le \frac{1}{\lambda n^2} \operatorname{tr}\left( Z_i^\top Z_i \right)$$

and $\mathbb{E}\left[ \frac{1}{\lambda n^2} \operatorname{tr}\left( Z_i^\top Z_i \right) \right] < +\infty$, we can interchange the differentiation and the expectation and get

$$f'_\alpha(\lambda_i) = \frac{\partial}{\partial \lambda_i} \mathbb{E}\left[ \frac{1}{n} \log \det \frac{N}{n} \right] = \mathbb{E}\left[ \frac{\partial}{\partial \lambda_i} \frac{1}{n} \log \det \frac{N}{n} \right]. \tag{38}$$

Thus we deduce

$$\left| \frac{\partial}{\partial \lambda_i} \mathbb{E}\left[ \frac{1}{n} \log \det \frac{N}{n} \right] \right| = \left| \mathbb{E}\left[ \frac{\partial}{\partial \lambda_i} \frac{1}{n} \log \det \frac{N}{n} \right] \right| \le \mathbb{E}\left| \frac{\partial}{\partial \lambda_i} \frac{1}{n} \log \det \frac{N}{n} \right| \le \mathbb{E}\left[ \frac{1}{\lambda n^2} \operatorname{tr}\left( Z_i^\top Z_i \right) \right].$$

By Lemma 4, $\mathbb{E} \operatorname{tr}\left( Z_i^\top Z_i \right) \asymp n^2$ and therefore $\mathbb{E}\left[ \frac{1}{\lambda n^2} \operatorname{tr}\left( Z_i^\top Z_i \right) \right] \lesssim \frac{1}{\lambda}$. The function sequence $\{f'_\alpha\}$ is uniformly bounded.

Then we want to show that $\{f'_\alpha\}$ is uniformly equicontinuous by showing that $\{f''_\alpha\}$ is uniformly bounded. Because

$$\left| \frac{\partial^2}{\partial \lambda_i^2} \frac{1}{n} \log \det \frac{N}{n} \right| = \frac{1}{n} \operatorname{tr}\left( Z_i^\top N^{-1} Z_i \right)^2 \le \frac{1}{n\lambda^2} \operatorname{tr}\left( \frac{Z_i^\top Z_i}{n} \right)^2$$

and $\mathbb{E}\left[ \frac{1}{n\lambda^2} \operatorname{tr}\left( \frac{Z_i^\top Z_i}{n} \right)^2 \right] < +\infty$, we can interchange the differentiation and the expectation and get

$$\frac{\partial^2}{\partial \lambda_i^2} \mathbb{E}\left[ \frac{1}{n} \log \det \frac{N}{n} \right] = \frac{\partial}{\partial \lambda_i} \mathbb{E}\left[ \frac{\partial}{\partial \lambda_i} \frac{1}{n} \log \det \frac{N}{n} \right] = \mathbb{E}\left[ \frac{\partial^2}{\partial \lambda_i^2} \frac{1}{n} \log \det \frac{N}{n} \right].$$

Therefore, we deduce

$$\left| \frac{\partial^2}{\partial \lambda_i^2} \mathbb{E}\left[ \frac{1}{n} \log \det \frac{N}{n} \right] \right| \le \mathbb{E}\left| \frac{\partial^2}{\partial \lambda_i^2} \frac{1}{n} \log \det \frac{N}{n} \right| \le \frac{1}{n\lambda^2} \mathbb{E} \operatorname{tr}\left( \frac{Z_i^\top Z_i}{n} \right)^2.$$

Again, by Lemma 4, $\operatorname{tr}\left( \frac{Z_i^\top Z_i}{n} \right)^2 \asymp n$. It follows that $\frac{1}{n\lambda^2} \mathbb{E} \operatorname{tr}\left( \frac{Z_i^\top Z_i}{n} \right)^2 \lesssim \frac{1}{\lambda^2}$. Therefore $\{f'_\alpha\}$ is uniformly equicontinuous.

We want to show $\lim_\alpha f'_\alpha(\lambda_i) = h'(\lambda_i)$ by contradiction. If it is not true, there exists $\epsilon > 0$ and a subsequence $\{f'_{\alpha_k}\}$ such that $\left| f'_{\alpha_k}(\lambda_i) - h'(\lambda_i) \right| \ge \epsilon$. Let $E = [a, b] \ni \lambda_i$ ($b > a > 0$) be

a closed interval that contains $\lambda_i$. The subsequence $\{f'_{\alpha_k}\}$ is uniformly bounded and uniformly equicontinuous. By the Arzela-Ascoli theorem, there exists a subsequence $\left\{f'_{\alpha_{k_j}}\right\}$ that converges uniformly on $\lambda_i \in E$. Recall $\lim_\alpha f_\alpha(\lambda_i) = h(\lambda_i)$ (Equation (37)). Thus $\lim_j f_{\alpha_{k_j}}(\lambda_i) = h(\lambda_i)$. By (Rudin, 1976, Theorem 7.17), for $\lambda_i \in E$, we have

$$\lim_j f'_{\alpha_{k_j}}(\lambda_i) = h'(\lambda_i).$$

This is a contradiction. Hence, we have shown that $\lim_\alpha f'_\alpha(\lambda_i) = h'(\lambda_i)$, which is exactly Equation (35) (recall $f'_\alpha(\lambda_i) = \frac{\partial}{\partial \lambda_i} \mathbb{E}\left[\frac{1}{n}\log\det\frac{N}{n}\right] = \mathbb{E}\left[\frac{\partial}{\partial \lambda_i}\frac{1}{n}\log\det\frac{N}{n}\right]$ in Equation (38)).

**Proof of Equation (36).** Define $g_\alpha(\lambda_j) = \frac{\partial}{\partial \lambda_i}\mathbb{E}\left[\frac{1}{n}\log\det\frac{N}{n}\right] = \mathbb{E}\left[\frac{\partial}{\partial \lambda_i}\frac{1}{n}\log\det\frac{N}{n}\right]$. Then $g'_\alpha(\lambda_j) = \frac{\partial^2}{\partial\lambda_j\partial\lambda_i}\mathbb{E}\left[\frac{1}{n}\log\det\frac{N}{n}\right] = \frac{\partial}{\partial\lambda_j}\mathbb{E}\left[\frac{\partial}{\partial\lambda_i}\frac{1}{n}\log\det\frac{N}{n}\right]$. We have

$$\left|\frac{\partial^2}{\partial\lambda_j\partial\lambda_i}\frac{1}{n}\log\det\frac{N}{n}\right|$$
$$= \frac{1}{n}\operatorname{tr}\left(Z_i Z_i^\top N^{-1} Z_j Z_j^\top N^{-1}\right)$$
$$= \frac{1}{n}\operatorname{tr}\left(Z_i^\top N^{-1} Z_j Z_j^\top N^{-1} Z_i\right)$$
$$= \frac{1}{n}\left\|Z_j^\top N^{-1} Z_i\right\|_F^2$$
$$\leq \frac{1}{n}\left\|Z_j\right\|_2^2\left\|Z_i\right\|_2^2\left\|N^{-1}\right\|_F^2$$
$$\leq \frac{1}{\lambda^2 n^2}\left\|Z_j\right\|_2^2\left\|Z_i\right\|_2^2.$$

where the last inequality is because $\left\|N^{-1}\right\|_F^2 \leq \left\|\frac{1}{\lambda n}I_n\right\|_F^2 = \frac{1}{\lambda^2 n}$. If $i \neq j$, by Lemma 4, we have

$$\frac{1}{\lambda^2 n^2}\mathbb{E}\left\|Z_j\right\|_2^2\left\|Z_i\right\|_2^2 = \frac{1}{\lambda^2 n^2}\mathbb{E}\left\|Z_j\right\|_2^2 \cdot \mathbb{E}\left\|Z_i\right\|_2^2 \lesssim \frac{1}{\lambda^2}.$$

If $i = j$, by Lemma 4, we have

$$\frac{1}{\lambda^2 n^2}\mathbb{E}\left\|Z_i\right\|_2^4 \lesssim \frac{1}{\lambda^2 n^2}\cdot n^2 = \frac{1}{\lambda^2}.$$

As a result, we get

$$\left|\frac{\partial^2}{\partial\lambda_j\partial\lambda_i}\frac{1}{n}\log\det\frac{N}{n}\right| \lesssim \frac{1}{n}\cdot n^2 \cdot \frac{1}{\lambda^2 n} = \frac{1}{\lambda^2}.$$

Thus we can interexchange $\frac{\partial}{\partial\lambda_j}$ and expectation, and get $g'_\alpha(\lambda_j) = \mathbb{E}\left[\frac{\partial^2}{\partial\lambda_j\partial\lambda_i}\frac{1}{n}\log\det\frac{N}{n}\right]$. Because $|g'_\alpha(\lambda_j)| \leq \mathbb{E}\left|\frac{\partial^2}{\partial\lambda_j\partial\lambda_i}\frac{1}{n}\log\det\frac{N}{n}\right| \lesssim \frac{1}{\lambda^2}$, the function sequence $\{g'_\alpha\}$ is uniformly bounded for $\lambda_j$.

Define $L = Z_j^\top N^{-1} Z_i$ and $W = Z_j^\top N^{-1} Z_j$. We have

$$\left| \frac{\partial^3}{\partial \lambda_j^2 \partial \lambda_i} \frac{1}{n} \log \det \frac{N}{n} \right|$$

$$= \frac{2}{n} \operatorname{tr} \left( L^\top W L \right)$$

$$\lesssim \frac{1}{\lambda n^2} \operatorname{tr} \left( L^\top Z_j^\top Z_j L \right)$$

$$= \frac{1}{\lambda n^2} \operatorname{tr} \left( Z_i^\top N^{-1} \left( Z_j Z_j^\top \right)^2 N^{-1} Z_i \right).$$

$$= \frac{1}{\lambda n^2} \left\| Z_j Z_j^\top N^{-1} Z_i \right\|_F^2$$

$$\leq \frac{1}{\lambda n^2} \left\| N^{-1} \right\|_F^2 \left\| Z_j Z_j^\top \right\|_2^2 \left\| Z_i \right\|_2^2$$

$$\leq \frac{1}{\lambda^3 n^3} \left\| Z_j Z_j^\top \right\|_2^2 \left\| Z_i \right\|_2^2$$

$$= \frac{1}{\lambda^3 n^3} \left\| Z_j \right\|_2^4 \left\| Z_i \right\|_2^2,$$

where the first inequality is because $W \preccurlyeq \frac{1}{\lambda n} Z_j^\top Z_j$ and the third inequality is because $N^{-1} \preccurlyeq \frac{1}{\lambda n} I_n$ and then $\left\| N^{-1} \right\|_F^2 \leq \left\| \frac{1}{\lambda n} I_n \right\|_F^2 \leq \frac{1}{\lambda^2 n}$. By Lemma 4, we have $\mathbb{E} \left\| Z_j \right\|_2^4 \lesssim n^2$ and $\mathbb{E} \left\| Z_i \right\|_2^2 \lesssim n$. If $i \neq j$, then $Z_j$ and $Z_i$ are independent, and we deduce

$$\frac{1}{\lambda^3 n^3} \mathbb{E} \left\| Z_j \right\|_2^4 \left\| Z_i \right\|_2^2 \lesssim \frac{1}{\lambda^3}.$$

If $i = j$, we have

$$\frac{1}{\lambda^3 n^3} \mathbb{E} \left\| Z_i \right\|_2^4 \left\| Z_i \right\|_2^2 = \frac{1}{\lambda^3 n^3} \mathbb{E} \left\| Z_i \right\|_2^6 \lesssim \frac{1}{\lambda^3}.$$

As a result, we deduce $\mathbb{E} \left[ \frac{\partial^3}{\partial \lambda_j^2 \partial \lambda_i} \frac{1}{n} \log \det \frac{N}{n} \right] = \frac{\partial}{\partial \lambda_j} \mathbb{E} \left[ \frac{\partial^2}{\partial \lambda_j \partial \lambda_i} \frac{1}{n} \log \det \frac{N}{n} \right] = g_\alpha''(\lambda_j)$. Moreover, we have

$$\left| g_\alpha''(\lambda_j) \right| \leq \mathbb{E} \left| \frac{\partial^3}{\partial \lambda_j^2 \partial \lambda_i} \frac{1}{n} \log \det \frac{N}{n} \right| \lesssim \frac{1}{\lambda^3}.$$

Therefore $\{ g_\alpha' \}$ is uniformly equicontinuous.

Define

$$w(\lambda_j) = \frac{\partial}{\partial \lambda_i} \inf_{\rho \in \mathbb{R}_+^m} \left[ \log \left( \lambda + \sum_{i \in [m]} \lambda_i \rho_i \right) + \sum_{i \in [m]} \left( \rho_i - z_i \left( \log \frac{\rho_i}{z_i} + 1 \right) \right) \right].$$

We want to show by contradiction that $\lim_\alpha g_\alpha'(\lambda_j) = w'(\lambda_j)$. Assume that it is not true. Then there exists $\epsilon > 0$ and a subsequence $\{ g_{\alpha_k}' \}$ such that $\left| g_{\alpha_k}'(\lambda_j) - w'(\lambda_j) \right| > \epsilon$. Since $\{ g_{\alpha_k}' \}$ is uniformly bounded and uniformly equicontinuous, by the Arzela-Ascoli theorem, there is a subsequence $\left\{ g_{\alpha_{k_r}}' \right\}$ that converges uniformly on a closed interval $E$ containing $\lambda_j$. Equation (35) shows that $\lim_\alpha g_\alpha(\lambda_j) = w(\lambda_j)$. It follows that $\lim_r g_{\alpha_{k_r}}(\lambda_j) = w(\lambda_j)$. By (Rudin, 1976, Theorem 7.17), for $\lambda_i \in E$, we have

$$\lim_r g_{\alpha_{k_r}}'(\lambda_j) = w'(\lambda_j),$$

which is a contradiction. Therefore, we have shown that $\lim_\alpha g_\alpha'(\lambda_j) = w'(\lambda_j)$, which is exactly Equation (36).

$\square$

# D    PROOF OF THEOREM 1

## D.1    PROOF OF ITEM 1

Define $g(\rho) = \log\left(\lambda + \sum_{j=1}^{m} \lambda_j \rho_j\right) + \sum_{j=1}^{m}\left(\rho_j - z_j(\log\frac{\rho_j}{z_j} + 1)\right)$. The function $g(\rho)$ is continuously differentiable on $\mathbb{R}_+^m$. The boundary of $\mathbb{R}_m^+$ is $\partial\mathbb{R}_+^m = \{\rho \in \mathbb{R}^m \mid (\forall i \in [m], \rho_i \geq 0) \wedge (\exists i \in [m], \rho_i = 0)\}$. Because $\lim_{\mathbb{R}_+^m \ni \rho \to \rho_0 \in \partial\mathbb{R}_+^m} g(\rho) = \lim_{\mathbb{R}_+^m \ni \rho \to \infty} g(\rho) = +\infty$, there exists a minimizer $\rho^* \in \mathbb{R}_+^m$ of $g(\rho)$.

Taking the derivative with respect to $\rho_i$ gives

$$\frac{\partial}{\partial \rho_i}\left[\log\left(\lambda + \sum_{j=1}^{m} \lambda_j \rho_j\right) + \sum_{j=1}^{m}\left(\rho_j - z_j(\log\frac{\rho_j}{z_j} + 1)\right)\right] = \frac{\lambda_i}{\lambda + \sum_{j=1}^{m}\lambda_j\rho_j} + 1 - \frac{z_i}{\rho_i}.$$

Setting it to zero gives Equation (6).

## D.2    PROOF OF ITEM 2

Recall Equation (6)

$$\frac{\lambda_i}{\lambda + \sum_{j=1}^{m}\lambda_j \rho_j^*} + 1 - \frac{z_i}{\rho_i^*} = 0, \quad \forall i \in [m].$$

Rewriting the above equation gives

$$(z_i - \rho_i^*)\left(\lambda + \sum_{k=1}^{m}\lambda_k\rho_k^*\right) = \lambda_i\rho_i^*, \quad \forall i \in [m].$$

Rewriting it in the linear algebraic form yields

$$(\mathbf{z} - \rho^*)\left(\lambda + \boldsymbol{\lambda}^\top\rho^*\right) = \boldsymbol{\lambda} \odot \rho^*.$$

Applying $\frac{\partial}{\partial \boldsymbol{\lambda}}$ to both sides and using the implicit function theorem, we get

$$(\mathbf{z} - \rho^*)\left(\rho^{*\top} + \boldsymbol{\lambda}^\top J\right) - J\left(\lambda + \boldsymbol{\lambda}^\top\rho^*\right) = \mathrm{diag}\,(\boldsymbol{\lambda})\,J + \mathrm{diag}\,(\rho^*)\,.$$

Arranging the above equation yields

$$\left(\mathrm{diag}\,(\boldsymbol{\lambda}) + \left(\lambda + \boldsymbol{\lambda}^\top\rho^*\right)I_m - (\mathbf{z} - \rho^*)\boldsymbol{\lambda}^\top\right)J = (\mathbf{z} - \rho^*)\rho^{*\top} - \mathrm{diag}\,(\rho^*)\,.$$

Define $a = \lambda + \boldsymbol{\lambda}^\top\rho^*$, $A = \mathrm{diag}\,(\boldsymbol{\lambda}) + \left(\lambda + \boldsymbol{\lambda}^\top\rho^*\right)I_m = \mathrm{diag}\,(\boldsymbol{\lambda}) + aI_m$ and $B = \mathrm{diag}\,(\boldsymbol{\lambda}) + \left(\lambda + \boldsymbol{\lambda}^\top\rho^*\right)I_m - (\mathbf{z} - \rho^*)\boldsymbol{\lambda}^\top = A - (\mathbf{z} - \rho^*)\boldsymbol{\lambda}^\top$. The matrix determinant lemma gives

$$\det(B) = \left(1 - \boldsymbol{\lambda}^\top A^{-1}(\mathbf{z} - \rho^*)\right)\det(A)\,.$$

Recall Equation (6) again and we have

$$\lambda_i + a = \frac{z_i a}{\rho_i^*}\,.$$

We have

$$a - \sum_{i \in [m]}\lambda_i\rho_i^*\left(1 - \rho_i^*/z_i\right)$$

$$= \left(\lambda + \sum_{i\in[m]}\lambda_i\rho_i^*\right) - \sum_{i\in[m]}\lambda_i\rho_i^*\left(1 - \rho_i^*/z_i\right)$$

$$= \lambda + \sum_{i\in[m]}\lambda_i\frac{(\rho_i^*)^2}{z_i} > 0\,.$$

It follows that

$$\frac{\sum_{i\in[m]}\lambda_i\rho_i^*\left(1 - \rho_i^*/z_i\right)}{a} < 1\,.$$

Then we compute $\boldsymbol{\lambda}^\top A^{-1} (\mathbf{z} - \rho^*)$:

$$\boldsymbol{\lambda}^\top A^{-1} (\mathbf{z} - \rho^*)$$
$$= \sum_{i \in [m]} \frac{\lambda_i (z_i - \rho_i^*)}{\lambda_i + a}$$
$$= \sum_{i \in [m]} \frac{\lambda_i (z_i - \rho_i^*)}{\frac{z_i a}{\rho_i^*}}$$
$$= \frac{\sum_{i \in [m]} \lambda_i \rho_i^* (1 - \rho_i^*/z_i)}{a} < 1 \, .$$

Thus we get $1 - \boldsymbol{\lambda}^\top A^{-1} (\mathbf{z} - \rho^*) > 0$. Therefore, $\det B \neq 0$ and the matrix $B$ is invertible.

### D.3 PROOF OF ITEM 3

**Lemma 11.** *Define* $N = \lambda n I_n + Z^\top \Lambda Z$, $\gamma = \sum_{i \in [m]} z_i$, $\mathbf{r} = (r_1, \dots, r_m)$, $\boldsymbol{\lambda} = (\lambda_1, \dots, \lambda_m)$, *and*

$$\vartheta(r_t, \mathbf{r}, \boldsymbol{\lambda}) = 2r_t \sqrt{1 + \sum_{i \in [m]} r_i^2} - 2r_t \sum_{i \in [m]} \sqrt{z_i} r_i + \sum_{i \in [m]} \frac{1}{\lambda_i} r_i^2 - \lambda r_t^2 \, .$$

*For any* $K_t \geq \frac{2}{\lambda}$ *and* $K_u \geq \frac{2\lambda_+ (2 + \sqrt{\gamma})}{\lambda}$, *we have*

$$\lim_{\substack{n, d_i \to +\infty \\ d_i/n \to z_i}} \operatorname{tr} N^{-1} = \lim_{\substack{n, d_i \to +\infty \\ d_i/n \to z_i}} \mathbb{E} \operatorname{tr} N^{-1}$$
$$= \max_{0 \leq r_t \leq K_t} \min_{0 \leq r_i \leq K_u} \vartheta(r_t, \mathbf{r}, \boldsymbol{\lambda}) = \min_{0 \leq r_i \leq K_u} \max_{0 \leq r_t \leq K_t} \vartheta(r_t, \mathbf{r}, \boldsymbol{\lambda}) = \max_{r_t \geq 0} \min_{r_i \geq 0} \vartheta(r_t, \mathbf{r}, \boldsymbol{\lambda}) = \min_{r_i \geq 0} \max_{r_t \geq 0} \vartheta(r_t, \mathbf{r}, \boldsymbol{\lambda}) \, . \tag{39}$$

*If* $r^*$ *is a solution to the optimization problem in Equation* (39)*, then*

$$1 + \sum_{j=1}^m r_j^{*2} = \left( \sum_{j=1}^m r_j^* \sqrt{z_j} + \lambda r_t^* \right)^2 , \tag{40}$$

$$r_t^* \frac{r_i^*}{\sqrt{1 + \sum_{j=1}^m r_j^{*2}}} = r_t^* \sqrt{z_i} - \frac{r_i^*}{\lambda_i} \, . \tag{41}$$

*Moreover, we have*

$$\frac{\partial}{\partial \lambda_i} \max_{r_t \geq 0} \min_{r_i \geq 0} \vartheta(r_t, \mathbf{r}, \boldsymbol{\lambda}) = -\frac{r_i^{*2}}{\lambda_i^2} \, .$$

*Proof.* Let $g \sim \mathcal{N}(0, I_n)$ be a multivariate standard normal random vector. We have

$$\operatorname{tr} N^{-1}$$
$$= \mathbb{E}_g g^\top N^{-1} g$$
$$= \mathbb{E}_g \sup_{t \in \mathbb{R}^n} \left( 2g^\top t - t^\top N t \right)$$
$$= \mathbb{E}_g \sup_{t \in \mathbb{R}^n} \left( 2g^\top t - t^\top Z^\top \Lambda Z t - n\lambda \|t\|_2^2 \right)$$
$$= \mathbb{E}_g \sup_{t \in \mathbb{R}^n} \inf_{u \in \mathbb{R}^d} \left( 2g^\top t - 2u^\top \Lambda Z t + u^\top \Lambda u - n\lambda \|t\|_2^2 \right)$$
$$= -2 \mathbb{E}_g \inf_{t \in \mathbb{R}^n} \sup_{u \in \mathbb{R}^d} \left( u^\top \Lambda Z t - g^\top t - \frac{1}{2} u^\top \Lambda u + \frac{1}{2} n\lambda \|t\|_2^2 \right)$$
$$= -2 \mathbb{E}_g \inf_{t \in \mathbb{R}^n} \sup_{u \in \mathbb{R}^d} \left( u^\top Z t - g^\top t - \frac{1}{2} u^\top \Lambda^{-1} u + \frac{1}{2} n\lambda \|t\|_2^2 \right)$$

We view

$$\inf_{t \in \mathbb{R}^n} \sup_{u \in \mathbb{R}^d} \left( u^\top Z t - g^\top t - \frac{1}{2} u^\top \Lambda^{-1} u + \frac{1}{2} n \lambda \|t\|_2^2 \right) \tag{42}$$

as the primal optimization (PO) problem in the convex Gaussian min-max theorem (CGMT) (Thrampoulidis et al., 2015).

The KKT conditions for Equation (42) give

$$Z^\top u - g + n\lambda t = 0 \,,$$
$$Z t - \Lambda^{-1} u = 0 \,.$$

Solving the above equations gives

$$t = N^{-1} g \,, \quad u = \Lambda Z N^{-1} g \,.$$

With probability at least $1 - 4 \exp(-cn)$ ($c > 0$ is a universal constant), we have $\|g\|_2 \leq 2\sqrt{n}$ and $\|Z\| \leq \sqrt{d} + 2\sqrt{n} \leq \left(2 + \sqrt{\gamma}\right)\sqrt{n}$. Therefore, we get

$$\|t\|_2 \leq \|N^{-1}\| \|g\|_2 \leq \frac{1}{\lambda n} \cdot 2\sqrt{n} = \frac{2}{\lambda \sqrt{n}} \,,$$

$$\|u\|_2 \leq \lambda_+ \|Z\| \|t\|_2 \leq \lambda_+ \left(2 + \sqrt{\gamma}\right) \sqrt{n} \cdot \frac{2}{\lambda \sqrt{n}} = \frac{2\lambda_+ \left(2 + \sqrt{\gamma}\right)}{\lambda} \,.$$

Write $u = \begin{pmatrix} u_1 \\ \vdots \\ u_m \end{pmatrix}$, where $u_i \in \mathbb{R}^{d_i}$. For all $K_t \geq \frac{2}{\lambda}$, $K_u \geq \frac{2\lambda_+ \left(2 + \sqrt{\gamma}\right)}{\lambda}$, the optimal solutions $t^*$ and $u^*$ to Equation (42) satisfy $\sqrt{n} \|t^*\|_2 \leq K_t$ and $\|u_i\|_2 \leq K_u$ for all $i \in [m]$ with probability at least $1 - 4 \exp(-cn)$. Define $S_t = \{t \in \mathbb{R}^n \mid \sqrt{n} \|t\|_2 \leq K_t\}$ and $S_u = \left\{u \in \mathbb{R}^d \mid \|u_i\| \leq K_u, \forall i \in [m]\right\}$. We use $\alpha$ to denote the indices $n, d_i$ and use $\lim_\alpha$ to denote $\lim_{\substack{n, d_i \to +\infty \\ d_i/n = z_i}}$. Define event

$$E_\alpha = \left\{ \inf_{t \in \mathbb{R}^n} \sup_{u \in \mathbb{R}^d} \left( u^\top Z t - g^\top t - \frac{1}{2} u^\top \Lambda^{-1} u + \frac{1}{2} n \lambda \|t\|_2^2 \right) = \inf_{t \in S_t} \sup_{u \in S_u} \left( u^\top Z t - g^\top t - \frac{1}{2} u^\top \Lambda^{-1} u + \frac{1}{2} n \lambda \|t\|_2^2 \right) \right\} \,.$$

Then with probability at least $1 - 4 \exp(-cn)$, we have $t^* \in S_t$ and $u^* \in S_u$. Therefore the event $E_\alpha$ occurs with probability at least $1 - 4 \exp(-cn)$, which yields

$$\mathbb{P}\{E_\alpha^c\} \leq 4 \exp(-cn) \,.$$

Since $\sum_{n \geq 1} 4 \exp(-cn) < +\infty$, by Borel-Cantelli lemma, we have

$$\mathbb{P}\left\{ \limsup_\alpha E_\alpha^c \right\} = \mathbb{P}\left\{ \left( \liminf_\alpha E_\alpha \right)^c \right\} = 0 \,.$$

Then with probability 1, all but finitely many $E_\alpha$ occur. Then almost surely there exists $n_0$ such that for all $n > n_0$, $E_\alpha$ occurs.

The auxiliary optimization (AO) problem is

$$\inf_{t \in S_t} \sup_{u \in S_u} \left( \|t\|_2 \, g_1^\top u + \|u\|_2 \, g_2^\top t - g^\top t - \frac{1}{2} u^\top \Lambda^{-1} u + \frac{1}{2} n \lambda \|t\|_2^2 \right)$$

$$= \inf_{0 \leq r_t \leq K_t} \sup_{0 \leq r_i \leq K_u} \left( -\frac{\left\| g - \sqrt{\sum_{i \in [m]} r_i^2} g_2 \right\|_2}{\sqrt{n}} r_t + r_t \sum_{i \in [m]} \frac{\|g_{1,i}\|_2}{\sqrt{n}} r_i - \frac{1}{2} \sum_{i \in [m]} \frac{1}{\lambda_i} r_i^2 + \frac{1}{2} \lambda r_t^2 \right)$$

$$= \inf_{0 \leq r_t \leq K_t} \sup_{0 \leq r_i \leq K_u} \left( -\sqrt{1 + \sum_{i \in [m]} r_i^2} \frac{\|g_3\|_2}{\sqrt{n}} r_t + r_t \sum_{i \in [m]} \frac{\|g_{1,i}\|_2}{\sqrt{n}} r_i - \frac{1}{2} \sum_{i \in [m]} \frac{1}{\lambda_i} r_i^2 + \frac{1}{2} \lambda r_t^2 \right) \,,$$

where $g_1 \sim \mathcal{N}(0, I_d)$, $g_2 \sim \mathcal{N}(0, I_n)$, and $g_3 \sim \mathcal{N}(0, I_n)$.

Taking $n, d_i \to +\infty$ with $d_i/n \to z_i$ constant, the strong law of large numbers gives

$$\sqrt{1 + \sum_{i \in [m]} r_i^2} \frac{\|g_3\|_2}{\sqrt{n}} \overset{\text{a.s.}}{\to} \sqrt{1 + \sum_{i \in [m]} r_i^2},$$

$$\frac{\|g_{1,j}\|_2}{\sqrt{n}} = \sqrt{\frac{d_j}{n}} \frac{\|g_{1,j}\|_2}{\sqrt{d_j}} \overset{\text{a.s.}}{\to} \sqrt{z_j}.$$

Define

$$X_\alpha(r_t, \mathbf{r}) = -\sqrt{1 + \sum_{i \in [m]} r_i^2} \frac{\|g_3\|_2}{\sqrt{n}} r_t + r_t \sum_{i \in [m]} \frac{\|g_{1,i}\|_2}{\sqrt{n}} r_i - \frac{1}{2} \sum_{i \in [m]} \frac{1}{\lambda_i} r_i^2 + \frac{1}{2} \lambda r_t^2.$$

It is a stochastic process on $(r_t, \mathbf{r}) \in [0, K_t] \times [0, K_u]^m$. We have

$$\lim_\alpha X_\alpha(r_t, \mathbf{r}) = X(r_t, \mathbf{r}) := -r_t \sqrt{1 + \sum_{i \in [m]} r_i^2} + r_t \sum_{i \in [m]} \sqrt{z_i} r_i - \frac{1}{2} \sum_{i \in [m]} \frac{1}{\lambda_i} r_i^2 + \frac{1}{2} \lambda r_t^2$$

almost surely. Since $\sqrt{1 + x^2}$ is convex and increasing and the function $\|\mathbf{r}\|_2$ is convex, thus $\sqrt{1 + \|\mathbf{r}\|_2^2}$ is convex in $\mathbf{r}$ and then $-\sqrt{1 + \sum_{i \in [m]} r_i^2} \frac{\|g_3\|_2}{\sqrt{n}} r_t = -\sqrt{1 + \|\mathbf{r}\|_2^2} \frac{\|g_3\|_2}{\sqrt{n}} r_t$ is concave in $\mathbf{r}$. Because $-\frac{1}{2} \sum_{i \in [m]} \frac{1}{\lambda_i} r_i^2$ is concave in $\mathbf{r}$ and $r_t \sum_{i \in [m]} \frac{\|g_{1,i}\|_2}{\sqrt{n}} r_i$ is linear in $\mathbf{r}$, we deduce that $X_\alpha(r_t, \mathbf{r})$ is concave in $\mathbf{r}$. By (Liese & Miescke, 2008, Lemma 7.75), $\sup_{\mathbf{r} \in [0, K_u]^m} |X_\alpha(r_t, \mathbf{r}) - X(r_t, \mathbf{r})| \to 0$ almost surely. Then for $\forall \epsilon > 0$, there exists $n_0(\epsilon), d_{0,i}(\epsilon), \delta_{0,i}(\epsilon)$ such that for all $n > n_0(\epsilon)$, $d_i > d_{0,i}(\epsilon)$, $|d_i/n - z_i| < \delta_{0,i}(\epsilon)$ and for all $\mathbf{r} \in [0, K_u]^m$, we have

$$X(r_t, \mathbf{r}) - \epsilon < X_\alpha(r_t, \mathbf{r}) < X(r_t, \mathbf{r}) + \epsilon.$$

Thus we obtain

$$X(r_t, \mathbf{r}) - \epsilon < X_\alpha(r_t, \mathbf{r}) \le \sup_{\mathbf{r} \in [0, K_u]^m} X_\alpha(r_t, \mathbf{r})$$

$$X_\alpha(r_t, \mathbf{r}) < X(r_t, \mathbf{r}) + \epsilon \le \sup_{\mathbf{r} \in [0, K_u]^m} X(r_t, \mathbf{r}) + \epsilon,$$

which in turn implies

$$\sup_{\mathbf{r} \in [0, K_u]^m} X(r_t, \mathbf{r}) - \epsilon \le \sup_{\mathbf{r} \in [0, K_u]^m} X_\alpha(r_t, \mathbf{r})$$

$$\sup_{\mathbf{r} \in [0, K_u]^m} X_\alpha(r_t, \mathbf{r}) \le \sup_{\mathbf{r} \in [0, K_u]^m} X(r_t, \mathbf{r}) + \epsilon.$$

It follows that $\left| \sup_{\mathbf{r} \in [0, K_u]^m} X_\alpha(r_t, \mathbf{r}) - \sup_{\mathbf{r} \in [0, K_u]^m} X(r_t, \mathbf{r}) \right| \le \epsilon$. In other words, we showed

$$|Y_\alpha(r_t) - Y(r_t)| \to 0$$

almost surely, where $Y(r_t) := \sup_{\mathbf{r} \in [0, K_u]^m} X_\alpha(r_t, \mathbf{r})$ and $Y(r_t) := \sup_{\mathbf{r} \in [0, K_u]^m} X(r_t, \mathbf{r})$.

Because $X_\alpha(r_t, \mathbf{r})$ is convex in $r_t$, then $Y(r_t) = \sup_{\mathbf{r} \in [0, K_u]^m} X_\alpha(r_t, \mathbf{r})$ is convex in $r_t$. By (Liese & Miescke, 2008, Lemma 7.75) again, $\sup_{r_t \in [0, K_t]} |Y_\alpha(r_t) - Y(r_t)| \to 0$ almost surely. A similar argument shows that

$$\left| \inf_{r_t \in [0, K_t]} Y_\alpha(r_t) - \inf_{r_t \in [0, K_t]} Y(r_t) \right| = \left| \inf_{r_t \in [0, K_t]} \sup_{\mathbf{r} \in [0, K_u]^m} X_\alpha(r_t, \mathbf{r}) - \inf_{r_t \in [0, K_t]} \sup_{\mathbf{r} \in [0, K_u]^m} X(r_t, \mathbf{r}) \right| \to 0$$

almost surely.

Therefore, we obtain

$$\inf_{t \in S_t} \sup_{u \in S_u} \left( \|t\|_2 \, g_1^\top u + \|u\|_2 \, g_2^\top t - g^\top t - \frac{1}{2} u^\top \Lambda^{-1} u + \frac{1}{2} n \lambda \|t\|_2^2 \right)$$

$$= \inf_{0 \le r_t \le K_t} \sup_{0 \le r_i \le K_u} \left( -\sqrt{1 + \sum_{i \in [m]} r_i^2} \frac{\|g_3\|_2}{\sqrt{n}} r_t + r_t \sum_{i \in [m]} \frac{\|g_{1,i}\|_2}{\sqrt{n}} r_i - \frac{1}{2} \sum_{i \in [m]} \frac{1}{\lambda_i} r_i^2 + \frac{1}{2} \lambda r_t^2 \right)$$

$$\xrightarrow{\text{a.s.}} \inf_{0 \le r_t \le K_t} \sup_{0 \le r_i \le K_u} \left( -r_t \sqrt{1 + \sum_{i \in [m]} r_i^2} + r_t \sum_{i \in [m]} \sqrt{z_i} r_i - \frac{1}{2} \sum_{i \in [m]} \frac{1}{\lambda_i} r_i^2 + \frac{1}{2} \lambda r_t^2 \right) \tag{43}$$

$$=: \mu \,.$$

Define event

$$A_\alpha = \left\{ \left| \inf_{t \in \mathbb{R}^n} \sup_{u \in \mathbb{R}^d} \left( u^\top Z t - g^\top t - \frac{1}{2} u^\top \Lambda^{-1} u + \frac{1}{2} n \lambda \|t\|_2^2 \right) - \mu \right| > \tau \right\},$$

$$B_\alpha = \left\{ \left| \inf_{t \in S_t} \sup_{u \in S_u} \left( u^\top Z t - g^\top t - \frac{1}{2} u^\top \Lambda^{-1} u + \frac{1}{2} n \lambda \|t\|_2^2 \right) - \mu \right| > \tau \right\},$$

$$C_\alpha = \left\{ \left| \inf_{t \in S_t} \sup_{u \in S_u} \left( \|t\|_2 \, g_1^\top u + \|u\|_2 \, g_2^\top t - g^\top t - \frac{1}{2} u^\top \Lambda^{-1} u + \frac{1}{2} n \lambda \|t\|_2^2 \right) - \mu \right| > \tau \right\}.$$

Recall

$$E_\alpha = \left\{ \inf_{t \in \mathbb{R}^n} \sup_{u \in \mathbb{R}^d} \left( u^\top Z t - g^\top t - \frac{1}{2} u^\top \Lambda^{-1} u + \frac{1}{2} n \lambda \|t\|_2^2 \right) = \inf_{t \in S_t} \sup_{u \in S_u} \left( u^\top Z t - g^\top t - \frac{1}{2} u^\top \Lambda^{-1} u + \frac{1}{2} n \lambda \|t\|_2^2 \right) \right\}.$$

We have $A_\alpha \cap E_\alpha \subseteq B_\alpha$. Equation (43) gives $\lim_\alpha \mathbb{P}\{C_\alpha\} = 0$ for any $\tau > 0$ because almost sure convergence implies convergence in probability. By the convex Gaussian min-max theorem (Thrampoulidis et al., 2015), we have

$$\mathbb{P}\{B_\alpha\} \le 2\mathbb{P}\{C_\alpha\} \,.$$

It follows that

$$\mathbb{P}\{A_\alpha\} \le \mathbb{P}\{A_\alpha \cap E_\alpha\} + \mathbb{P}\{E_\alpha^c\} \le \mathbb{P}\{B_\alpha\} + \mathbb{P}\{E_\alpha^c\} \le 2\mathbb{P}\{C_\alpha\} + \mathbb{P}\{E_\alpha^c\} \,.$$

Taking $\limsup_\alpha$ on both sides, because $\limsup_\alpha \mathbb{P}\{B_\alpha\} \le 2\limsup_\alpha \mathbb{P}\{C_\alpha\} = 0$, we get

$$\limsup_\alpha \mathbb{P}\{A_\alpha\} \le \limsup_\alpha \mathbb{P}\{E_\alpha^c\} \le \mathbb{P}\left\{ \limsup_\alpha E_\alpha^c \right\} = 0 \,,$$

where the second inequality is because of the reverse Fatou's lemma. Thus

$$\inf_{t \in \mathbb{R}^n} \sup_{u \in \mathbb{R}^d} \left( u^\top Z t - g^\top t - \frac{1}{2} u^\top \Lambda^{-1} u + \frac{1}{2} n \lambda \|t\|_2^2 \right) \xrightarrow{P} \mu \,.$$

Therefore, we deduce

$$g^\top N^{-1} g \xrightarrow{P} -2 \inf_{0 \le r_t \le K_t} \sup_{0 \le r_i \le K_u} \left( -r_t \sqrt{1 + \sum_{i \in [m]} r_i^2} + r_t \sum_{i \in [m]} \sqrt{z_i} r_i - \frac{1}{2} \sum_{i \in [m]} \frac{1}{\lambda_i} r_i^2 + \frac{1}{2} \lambda r_t^2 \right)$$

$$= \sup_{0 \le r_t \le K_t} \inf_{0 \le r_i \le K_u} \left( 2 r_t \sqrt{1 + \sum_{i \in [m]} r_i^2} - 2 r_t \sum_{i \in [m]} \sqrt{z_i} r_i + \sum_{i \in [m]} \frac{1}{\lambda_i} r_i^2 - \lambda r_t^2 \right)$$

$$= \sup_{0 \le r_t \le K_t} \inf_{0 \le r_i \le K_u} \vartheta(r_t, \mathbf{r}, \boldsymbol{\lambda}) \,. \tag{44}$$

Because $\left| g^\top N^{-1} g \right| \le \frac{1}{\lambda n} \|g\|_2^2$ and $\mathbb{E}\frac{1}{\lambda n} \|g\|_2^2 = \frac{1}{\lambda} < \infty$, by the dominated convergence theorem for convergence in probability (Cohn, 2013, Proposition 3.1.6), we get

$$\lim_\alpha \operatorname{tr} N^{-1} = \lim_\alpha \mathbb{E}_g \left[ g^\top N^{-1} g \right] = \max_{0 \le r_t \le K_t} \min_{0 \le r_i \le K_u} \vartheta(r_t, \mathbf{r}, \boldsymbol{\lambda}) \,. \tag{45}$$

Note that $2r_t\sqrt{1 + \sum_{i\in[m]} r_i^2}$ is convex in $\mathbf{r}$, $-2r_t\sum_{i\in[m]}\sqrt{z_i}r_i$ is linear in $\mathbf{r}$, and $\sum_{i\in[m]}\frac{1}{\lambda_i}r_i^2$ is strongly convex in $\mathbf{r}$. Thus $\vartheta$ is strongly convex in $\mathbf{r}$. Note that $2r_t\sqrt{1+\sum_{i\in[m]} r_i^2} - 2r_t\sum_{i\in[m]}\sqrt{z_i}r_i$ is linear in $r_t$ and that $-\lambda r_t^2$ is strongly concave in $r_t$. Thus $\vartheta$ is strongly concave in $r_t$. Then $\vartheta$ has a unique saddle point $(r_t^*, \mathbf{r}^*)$ on $[0, K_t] \times [0, K_u]^m$ that satisfies

$$\max_{r_t\in[0,K_t]} \min_{\mathbf{r}\in[0,K_u]^m} \vartheta(r_t, \mathbf{r}) = \min_{\mathbf{r}\in[0,K_u]^m} \max_{r_t\in[0,K_t]} \vartheta(r_t, \mathbf{r}) = \vartheta(r_t^*, \mathbf{r}^*), \tag{46}$$

where the first equality is due to Sion's minimax theorem.

Since $\left|\operatorname{tr} N^{-1}\right| \leq \frac{1}{\lambda}$, using the dominated convergence theorem and combining Equation (45) and Equation (46) yields

$$\lim_\alpha \mathbb{E}\operatorname{tr} N^{-1} = \max_{0\leq r_t\leq K_t} \min_{0\leq r_i\leq K_u} \vartheta(r_t, \mathbf{r}, \boldsymbol{\lambda}) = \min_{0\leq r_i\leq K_u} \max_{0\leq r_t\leq K_t} \vartheta(r_t, \mathbf{r}, \boldsymbol{\lambda}).$$

By the uniqueness of the limit, the right-hand side $\max_{0\leq r_t\leq K_t} \min_{0\leq r_i\leq K_u} \vartheta(r_t, \mathbf{r}, \boldsymbol{\lambda})$ and $\min_{0\leq r_i\leq K_u} \max_{0\leq r_t\leq K_t} \vartheta(r_t, \mathbf{r}, \boldsymbol{\lambda})$ do not depend on $K_t$ and $K_u$ as long as $K_t \geq \frac{2}{\lambda} and K_u \geq \frac{2\lambda_+(2+\sqrt{\gamma})}{\lambda}$. Thus we have

$$\lim_\alpha \mathbb{E}\operatorname{tr} N^{-1} = \max_{r_t\geq 0} \min_{r_i\geq 0} \vartheta(r_t, \mathbf{r}, \boldsymbol{\lambda}) = \min_{r_i\geq 0} \max_{r_t\geq 0} \vartheta(r_t, \mathbf{r}, \boldsymbol{\lambda}).$$

If $r_t^* = 0$, then $\vartheta(0, \mathbf{r}^*) = \min_{\mathbf{r}\in[0,K_u]^m}\sum_{i\in[m]}\frac{1}{\lambda_i}r_i^2 = 0$. Thus $\mathbf{r}^*$ must be zero. However, $\vartheta\left(\frac{1}{2\lambda}, \mathbf{0}\right) = \frac{3}{4\lambda} > \vartheta(0, \mathbf{r}^*)$. Therefore $r_t^* > 0$. We compute the partial derivative

$$\frac{\partial\vartheta}{\partial r_i} = 2r_t\frac{r_i}{\sqrt{1+\sum_{i\in[m]} r_i^2}} - 2r_t\sqrt{z_i} + 2\frac{r_i}{\lambda_i}.$$

If $r_i^* = 0$, we have

$$\left.\frac{\partial\vartheta}{\partial r_i}\right|_{r_i=0, r_t=r_t^*} = -2r_t^*\sqrt{z_i} < 0.$$

Therefore, one can increase $r_i^*$ and make $\max_{r_t\in[0,K_t]} \min_{\mathbf{r}\in[0,K_u]^m} \vartheta(r_t, \mathbf{r})$ smaller, which results in a contradiction. Thus $r_i^* > 0$. Thus the minimax value is attained when $r_t, r_i > 0$ for all $i \in [m]$.

To obtain the optimality condition, we compute the partial derivatives

$$\frac{\partial\vartheta}{\partial r_t} = 2\sqrt{1 + \sum_{i\in[m]} r_i^2} - 2\left(\sum_{i\in[m]} r_i\sqrt{z_i}\right) - 2\lambda r_t,$$

$$\frac{\partial\vartheta}{\partial r_i} = 2r_t\frac{r_i}{\sqrt{1+\sum_{i\in[m]} r_i^2}} - 2r_t\sqrt{z_i} + 2\frac{r_i}{\lambda_i}.$$

Setting them to zero gives the optimality condition for $r_t^*, r_1^*, \ldots, r_m^*$ and yields Equation (40) and Equation (41).

Using the envelope theorem, we get

$$\frac{\partial}{\partial\lambda_i} \max_{r_t\in[0,K_t]} \min_{\mathbf{r}\in[0,K_u]^m} \vartheta(r_t, \mathbf{r}, \boldsymbol{\lambda})$$

$$= \frac{\partial\vartheta(r_t^*, \mathbf{r}^*, \lambda_1, \ldots, \lambda_m)}{\partial\lambda_i}$$

$$= -\frac{r_i^{*2}}{\lambda_i^2}.$$

$\square$

**Lemma 12.** *Define $N = \lambda n I_n + Z^\top \Lambda Z$. The following equation holds*

$$\lim_{\substack{n,d_i\to+\infty \\ d_i/n\to z_i}} \mathbb{E}\left[\frac{\partial}{\partial\lambda_i}\operatorname{tr}\left(N^{-1}\right)\right] = \frac{\partial^2}{\partial\lambda_i\partial\lambda}\inf_{\rho\in\mathbb{R}_+^m}\left[\log\left(\lambda + \sum_{j=1}^m \lambda_j\rho_j\right) + \sum_{j=1}^m\left(\rho_j - z_j(\log\frac{\rho_j}{z_j} + 1)\right)\right] = -\frac{r_i^{*2}}{\lambda_i^2},$$

where $r^*$ is a solution to $\sup_{r_t>0} \inf_{r_1,\ldots,r_m>0} \vartheta(r_t, r_1, \ldots, r_m, \lambda_1, \ldots, \lambda_m)$ and

$$\vartheta(r_t, r_1, \ldots, r_m, \lambda_1, \ldots, \lambda_m) = 2r_t \sqrt{1 + \sum_{i \in [m]} r_i^2} - 2r_t \sum_{i \in [m]} \sqrt{z_i} r_i + \sum_{i \in [m]} \frac{1}{\lambda_i} r_i^2 - \lambda r_t^2.$$

*Proof.* Since

$$\left| \frac{\partial}{\partial \lambda_i} \operatorname{tr}\left(N^{-1}\right) \right| = \operatorname{tr}\left(Z_i^\top N^{-2} Z_i\right) \leq \frac{1}{(\lambda n)^2} \operatorname{tr}\left(Z_i^\top Z_i\right) \tag{47}$$

and $\mathbb{E} \frac{1}{(\lambda n)^2} \operatorname{tr}\left(Z_i^\top Z_i\right) \asymp \frac{1}{\lambda^2}$ (by Lemma 4), using the dominated convergence theorem gives

$$\mathbb{E}\left[ \frac{\partial}{\partial \lambda_i} \operatorname{tr}\left(N^{-1}\right) \right] = \frac{\partial}{\partial \lambda_i} \mathbb{E} \operatorname{tr}\left(N^{-1}\right).$$

We use $\alpha$ to denote the indices $n, d_i$ and use $\lim_\alpha$ to denote $\lim_{\substack{n, d_i \to +\infty \\ d_i/n = z_i}}$. Define $f_\alpha(\lambda_i) = \mathbb{E} \operatorname{tr}\left(N^{-1}\right)$, $g(\lambda_i) = \frac{\partial}{\partial \lambda} \inf_{\rho \in \mathbb{R}_+^m} \left[\log\left(\lambda + \sum_{i \in [m]} \lambda_i \rho_i\right) + \sum_{i \in [m]} \left(\rho_i - z_i \left(\log \frac{\rho_i}{z_i} + 1\right)\right)\right]$, and $h(\lambda_i) = \sup_{r_t>0} \inf_{r_1,\ldots,r_m>0} \vartheta(r_t, r_1, \ldots, r_m, \lambda_1, \ldots, \lambda_m)$. Because $\left|\operatorname{tr}\left(N^{-1}\right)\right| \leq \operatorname{tr}\left(\frac{1}{\lambda n} I_n\right) \leq \frac{1}{\lambda}$ and $\lim_\alpha \operatorname{tr}\left(N^{-1}\right) = g(\lambda_i)$ (by Lemma 9), we have Lemma 9 $\lim_\alpha f_\alpha(\lambda_i) = \lim_\alpha \mathbb{E} \operatorname{tr}\left(N^{-1}\right) = g(\lambda_i)$. Lemma 11 shows $\lim_\alpha f_\alpha(\lambda_i) = h(\lambda_i)$. Therefore $\lim_\alpha f_\alpha(\lambda_i) = g(\lambda_i) = h(\lambda_i)$.

Because of Equation (47), we have $f'_\alpha(\lambda_i) = \frac{\partial}{\partial \lambda_i} \mathbb{E} \operatorname{tr}\left(N^{-1}\right)$ and

$$|f'_\alpha(\lambda_i)| = \left| \frac{\partial}{\partial \lambda_i} \mathbb{E} \operatorname{tr}\left(N^{-1}\right) \right| = \left| \mathbb{E}\left[ \frac{\partial}{\partial \lambda_i} \operatorname{tr}\left(N^{-1}\right) \right] \right| \leq \mathbb{E} \left| \frac{\partial}{\partial \lambda_i} \operatorname{tr}\left(N^{-1}\right) \right| \lesssim \frac{1}{\lambda^2}$$

and therefore $\{f'_\alpha\}$ is uniformly bounded for $\lambda_i$. Because

$$\left| \frac{\partial^2}{\partial \lambda_i^2} \operatorname{tr}\left(N^{-1}\right) \right|$$
$$= 2 \operatorname{tr}\left(N^{-1} Z_i Z_i^\top N^{-1} Z_i Z_i^\top N^{-1}\right)$$
$$\lesssim \frac{1}{\lambda n} \operatorname{tr}\left(N^{-1} \left(Z_i Z_i^\top\right)^2 N^{-1}\right)$$
$$= \frac{1}{\lambda n} \operatorname{tr}\left(Z_i Z_i^\top N^{-2} Z_i Z_i^\top\right)$$
$$\leq \frac{1}{(\lambda n)^3} \operatorname{tr}\left(Z_i Z_i^\top\right)^2,$$

and $\mathbb{E} \frac{1}{(\lambda n)^3} \operatorname{tr}\left(Z_i Z_i^\top\right)^2 \asymp \frac{1}{\lambda^3}$ (by Lemma 4), using the dominated convergence theorem yields

$$\mathbb{E}\left[ \frac{\partial^2}{\partial \lambda_i^2} \operatorname{tr}\left(N^{-1}\right) \right] = \frac{\partial}{\partial \lambda_i} \mathbb{E}\left[ \frac{\partial}{\partial \lambda_i} \operatorname{tr}\left(N^{-1}\right) \right] = \frac{\partial^2}{\partial \lambda_i^2} \mathbb{E}\left[\operatorname{tr}\left(N^{-1}\right)\right] = f''_\alpha(\lambda_i).$$

Moreover, we have

$$|f''_\alpha(\lambda_i)| \leq \mathbb{E} \left| \frac{\partial^2}{\partial \lambda_i^2} \operatorname{tr}\left(N^{-1}\right) \right| \lesssim \frac{1}{\lambda^3}.$$

Thus $\{f'_\alpha\}$ is uniformly equicontinuous for $\lambda_i$. We want to show that $\lim_\alpha f'_\alpha(\lambda_i) = g'(\lambda_i)$ by contradiction. Assume that it is not true. Then there exists $\epsilon > 0$ and a subsequence $\{f'_{\alpha_k}\}$ such that $\left|f'_{\alpha_k}(\lambda_i) - g'(\lambda_i)\right| > \epsilon$. Since $\{f'_{\alpha_k}\}$ is uniformly bounded and uniformly equicontinuous for $\lambda_i \in E$ ($E$ is any closed finite interval containing $\lambda_i$), by the Arzela-Ascoli theorem, there exists a subsequence $\{f'_{\alpha_{k_r}}\}$ that converges uniformly on $E$. Since $\lim_\alpha f_{\alpha_{k_r}}(\lambda_i) = g(\lambda_i)$, by (Rudin, 1976, Thoerem 7.17), we have

$$\lim_r f'_{\alpha_{k_r}}(\lambda_i) = g'(\lambda_i),$$

which yields a contradiction. Therefore, we have $\lim_\alpha f'_\alpha(\lambda_i) = g'(\lambda_i)$. Recall $g(\lambda_i) = h(\lambda_i)$ for any $\lambda_i > 0$. Then by the final part of Lemma 11, we have $\lim_\alpha f'_\alpha(\lambda_i) = g'(\lambda_i) = h'(\lambda_i) = -\frac{r_i^{*2}}{\lambda_i^2}$. □

### D.4 BIAS

**Lemma 13.** *Suppose that* $U \sim \bigoplus_{i \in [m]} \mathrm{Unif}\,(O\,(d_i))$ *and* $V$ *are two independent* $d \times d$ *random matrices such that* $V \stackrel{\mathrm{d}}{=} UVU^\top$, *where* $d = \sum_{i=1}^m d_i$. *Let* $\theta \in \mathbb{R}^d$ *be a fixed vector. Write*
$$\theta = \begin{bmatrix} \theta_1 \\ \vdots \\ \theta_m \end{bmatrix}, \text{ where } \theta_i \in \mathbb{R}^{d_i}. \text{ Let}$$

$$\phi \sim \bigoplus_{i \in [m]} \mathrm{Unif}\left(\mathbb{S}^{d_i - 1}\left(\|\theta_i\|_2\right)\right)$$

*be a random vector independent of* $V$ *and let* $\Lambda = \mathrm{diag}\,(\lambda_1 I_{d_1}, \ldots, \lambda_m I_{d_m}) \in \mathbb{R}^{d \times d}$. *Then we have*
$$\mathbb{E}\left[\left\|\Lambda^{1/2} V \theta\right\|_2^2\right] = \mathbb{E}\left[\left\|\Lambda^{1/2} V \phi\right\|_2^2\right].$$

*Proof.* Recall $U\Lambda U^\top = \Lambda$ and noticing $U^\top \theta \stackrel{\mathrm{d}}{=} \phi$, we get
$$\mathbb{E}\left[\left\|\Lambda^{1/2} V \theta\right\|_2^2\right]$$
$$= \mathbb{E}\left[\left\|\Lambda^{1/2} U V U^\top \theta\right\|_2^2\right]$$
$$= \mathbb{E}\left[\theta^\top U V^\top U^\top \Lambda U V U^\top \theta\right]$$
$$= \mathbb{E}\left[\theta^\top U V^\top \Lambda V U^\top \theta\right]$$
$$= \mathbb{E}\left[\left\|\Lambda^{1/2} V U^\top \theta\right\|_2^2\right]$$
$$= \mathbb{E}\left[\left\|\Lambda^{1/2} V \phi\right\|_2^2\right].$$

$\square$

**Lemma 14.** *Define* $\tilde{\Theta} = \mathrm{diag}\left(\|\theta_1'\|_2^2/d_1 I_{d_1}, \ldots, \|\theta_m'\|_2^2/d_m I_{d_m}\right)$ *and* $S = \Lambda^{1/2} Z \left(n\lambda I_n + Z^\top \Lambda Z\right)^{-1} Z^\top \Lambda^{1/2}$. *Then we have*
$$B_{\lambda,d,n} = \|\theta^*\|_\Sigma^2 - 2\mathbb{E}\,\mathrm{tr}\left(\Lambda S \tilde{\Theta}\right) + \mathbb{E}\,\mathrm{tr}\left(S\Lambda S\tilde{\Theta}\right).$$

*Proof.* Recall Equation (22) in Lemma 5
$$B_{\lambda,d,n} = \mathbb{E}\left[\|\Lambda^{1/2}\left(I_d + \frac{1}{n\lambda}\Lambda^{1/2} ZZ^\top \Lambda^{1/2}\right)^{-1}\theta'\|_2^2\right].$$

Let $U \sim \bigoplus_{i \in [m]} \mathrm{Unif}\,(O(d_i))$ be a random matrix independent of $Z$. Because $UZ \stackrel{\mathrm{d}}{=} Z$, we have
$$I_d + \frac{1}{n\lambda}\Lambda^{1/2} ZZ^\top \Lambda^{1/2} \stackrel{\mathrm{d}}{=} I_d + \frac{1}{n\lambda}\Lambda^{1/2} U ZZ^\top U^\top \Lambda^{1/2} = U\left(I_d + \frac{1}{n\lambda}\Lambda^{1/2} ZZ^\top \Lambda^{1/2}\right)U^\top.$$

Define $\tilde{\theta} \sim \bigoplus_{i \in [m]} \mathrm{Unif}(\mathbb{S}^{d_i - 1}(\|\theta_i'\|_2))$. Lemma 13 gives
$$B_{\lambda,d,n} = \mathbb{E}\left[\|\Lambda^{1/2}\left(I_d + \frac{1}{n\lambda}\Lambda^{1/2} ZZ^\top \Lambda^{1/2}\right)^{-1}\tilde{\theta}\|_2^2\right]$$
$$= \mathbb{E}\left[\|\Lambda^{1/2}\left(I_d - S\right)\tilde{\theta}\|_2^2\right]$$
$$= \mathbb{E}\left[\|\left(I_d - S\right)\tilde{\theta}\|_\Lambda^2\right]$$
$$= \mathbb{E}\left\|\tilde{\theta}\right\|_\Lambda^2 - \mathbb{E}\left[\tilde{\theta}^\top \Lambda S \tilde{\theta}\right] - \mathbb{E}\left[\tilde{\theta}^\top S\Lambda \tilde{\theta}\right] + \mathbb{E}\left[\tilde{\theta}^\top S\Lambda S\tilde{\theta}\right].$$

Notice that $\left\|\tilde{\theta}\right\|_\Lambda^2 = \|\theta'\|_\Lambda^2$ and $\tilde{\Theta} = \mathbb{E}\left[\tilde{\theta}\tilde{\theta}^\top\right]$. Because $\tilde{\Theta}$ commutes with $\Lambda$, we have $\operatorname{tr}\left(S\Lambda\tilde{\Theta}\right) = \operatorname{tr}\left(S\tilde{\Theta}\Lambda\right) = \operatorname{tr}\left(\Lambda S\tilde{\Theta}\right)$. In light of these, we deduce

$$B_{\lambda,d,n} = \|\theta'\|_\Lambda^2 - \mathbb{E}\operatorname{tr}\left(\Lambda S\tilde{\Theta}\right) - \mathbb{E}\operatorname{tr}\left(S\Lambda\tilde{\Theta}\right) + \mathbb{E}\operatorname{tr}\left(S\Lambda S\tilde{\Theta}\right)$$
$$= \|\theta'\|_\Lambda^2 - 2\mathbb{E}\operatorname{tr}\left(\Lambda S\tilde{\Theta}\right) + \mathbb{E}\operatorname{tr}\left(S\Lambda S\tilde{\Theta}\right) .$$

$\square$

Lemma 14 expresses the bias $B_{\lambda,d,n}$ as the sum of three terms.

**Computing $\|\theta'\|_\Lambda^2$** Note that $\|\theta'\|_\Lambda^2 = \theta'^\top \Lambda \theta' = \sum_{i\in[m]} \lambda_i \|\theta'_i\|_2^2$. Therefore,

$$\lim_{\substack{n,d_i\to+\infty \\ d_i/n\to z_i \\ \|\Pi_i\theta^*\|_2\to\eta_i}} \|\theta'\|_\Lambda^2 = \mathbf{q}^\top\left(\boldsymbol{\lambda}\odot\mathbf{z}\right) .$$

**Computing $\mathbb{E}\operatorname{tr}\left(\Lambda S\tilde{\Theta}\right)$** Define $N = \lambda n I_n + Z^\top \Lambda Z = \lambda n I_n + \sum_{i\in[m]} \lambda_i Z_i Z_i^\top$. We have

$$\mathbb{E}\operatorname{tr}\left(\Lambda S\tilde{\Theta}\right)$$
$$=\mathbb{E}\operatorname{tr}\left(Z^\top \Lambda^{1/2}\tilde{\Theta}\Lambda^{3/2}Z\left(n\lambda I_n + Z^\top\Lambda Z\right)^{-1}\right)$$
$$=\mathbb{E}\operatorname{tr}\left(Z^\top\Lambda^2\tilde{\Theta}ZN^{-1}\right)$$
$$=\sum_{i\in[m]}\lambda_i^2\frac{\|\theta'_i\|_2^2}{d_i}\mathbb{E}\operatorname{tr}\left(Z_iZ_i^\top N^{-1}\right)$$
$$=\sum_{i\in[m]}\lambda_i^2\|\theta'_i\|_2^2\frac{n}{d_i}\mathbb{E}\left[\frac{\partial}{\partial\lambda_i}\frac{1}{n}\log\det\frac{N}{n}\right]$$
$$=\sum_{i\in[m]}\lambda_i^2\|\theta'_i\|_2^2\frac{n}{d_i}\frac{\partial}{\partial\lambda_i}\mathbb{E}\left[\frac{1}{n}\log\det\frac{N}{n}\right] ,$$

where the second inequality is because $\tilde{\Theta}$ commutes with $\Lambda^{3/2}$ and the final equality is because of Equation (38). Taking $\lim_{\substack{n,d_i\to+\infty \\ d_i/n\to z_i \\ \|\Pi_i\theta^*\|_2\to\eta_i}}$ and using Lemma 10 gives

$$\lim_{\substack{n,d_i\to+\infty \\ d_i/n\to z_i \\ \|\Pi_i\theta^*\|_2\to\eta_i}}\mathbb{E}\operatorname{tr}\left(\Lambda S\tilde{\Theta}\right) = \sum_{i\in[m]}\frac{\lambda_i^2\eta_i^2}{z_i}\frac{\partial}{\partial\lambda_i}\inf_{\rho\in\mathbb{R}_+^m}\left[\log\left(\lambda+\sum_{i\in[m]}\lambda_i\rho_i\right)+\sum_{i\in[m]}\left(\rho_i - z_i\left(\log\frac{\rho_i}{z_i}+1\right)\right)\right] .$$

Using the envelope theorem yields

$$\frac{\partial}{\partial\lambda_i}\inf_{\rho\in\mathbb{R}_+^m}\left[\log\left(\lambda+\sum_{i\in[m]}\lambda_i\rho_i\right)+\sum_{i\in[m]}\left(\rho_i - z_i\left(\log\frac{\rho_i}{z_i}+1\right)\right)\right] = \frac{\rho_i^*}{\lambda+\sum_{i\in[m]}\lambda_i\rho_i^*} = \frac{z_i - \rho_i^*}{\lambda_i} ,$$

where the final equality is because of Equation (6) in Item 1. Therefore, we deduce

$$\lim_{\substack{n,d_i\to+\infty \\ d_i/n\to z_i \\ \|\Pi_i\theta^*\|_2\to\eta_i}}\mathbb{E}\operatorname{tr}\left(\Lambda S\tilde{\Theta}\right) = \sum_{i\in[m]}\frac{\lambda_i^2\eta_i^2}{z_i}\frac{z_i - \rho_i^*}{\lambda_i} = \sum_{i\in[m]}\lambda_i\eta_i^2\left(1 - \frac{\rho_i^*}{z_i}\right) = \mathbf{q}^\top\left(\boldsymbol{\lambda}\odot\left(\mathbf{z}-\boldsymbol{\rho}^*\right)\right) .$$

**Computing** $\mathbb{E}\,\mathrm{tr}\left(S\Lambda S\tilde{\Theta}\right)$    We have

$$
\mathbb{E}\,\mathrm{tr}\left(S\Lambda S\tilde{\Theta}\right)
$$

$$
=\mathbb{E}\,\mathrm{tr}\left[\Lambda^{1/2}ZN^{-1}Z^\top\Lambda^2 ZN^{-1}Z^\top\Lambda^{1/2}\tilde{\Theta}\right]
$$

$$
=\mathbb{E}\,\mathrm{tr}\left[Z^\top\Lambda^{1/2}\tilde{\Theta}\Lambda^{1/2}ZN^{-1}Z^\top\Lambda^2 ZN^{-1}\right]
$$

$$
=\mathbb{E}\,\mathrm{tr}\left[Z^\top\Lambda\tilde{\Theta}ZN^{-1}Z^\top\Lambda^2 ZN^{-1}\right]
$$

$$
=\sum_{i\in[m]}\frac{\lambda_i\|\theta'_i\|_2^2}{d_i}\sum_{j\in[m]}\lambda_j^2\mathbb{E}\,\mathrm{tr}\left[Z_iZ_i^\top N^{-1}Z_jZ_j^\top N^{-1}\right]
$$

$$
=-\sum_{i\in[m]}\frac{\lambda_i\|\theta'_i\|_2^2 n}{d_i}\sum_{j\in[m]}\lambda_j^2\mathbb{E}\left[\frac{\partial^2}{\partial\lambda_j\partial\lambda_i}\frac{1}{n}\log\det\frac{N}{n}\right]\,,
$$

where the third equality is because $\tilde{\Theta}$ commutes with $\Lambda^{1/2}$. Taking $\lim_{\substack{n,d_i\to+\infty\\ d_i/n\to z_i\\ \|\Pi_i\theta^*\|_2\to\eta_i}}$ and using Lemma 10 gives

$$
\lim_{\substack{n,d_i\to+\infty\\ d_i/n\to z_i\\ \|\Pi_i\theta^*\|_2\to\eta_i}}\mathbb{E}\,\mathrm{tr}\left(S\Lambda S\tilde{\Theta}\right)=-\sum_{i\in[m]}\frac{\lambda_i\eta_i^2}{z_i}\sum_{j\in[m]}\lambda_j^2\frac{\partial^2}{\partial\lambda_j\partial\lambda_i}\inf_{\rho\in\mathbb{R}_+^m}\left[\log\left(\lambda+\sum_{l\in[m]}\lambda_l\rho_l\right)+\sum_{l\in[m]}\left(\rho_l-z_l\left(\log\frac{\rho_l}{z_l}+1\right)\right)\right]\,.
$$

Write $\boldsymbol{\lambda}=(\lambda_1,\ldots,\lambda_m)^\top$ and $\mathbf{z}=(z_1,\ldots,z_m)^\top$. Let $\rho^*\in\mathbb{R}^m$ be a minimizer of Equation (5) and $J=\frac{\partial\rho^*}{\partial\boldsymbol{\lambda}}\in\mathbb{R}^{m\times m}$ be the Jacobian matrix $J_{ij}=\frac{\partial\rho_i^*}{\partial\lambda_j}$. Recall Item 2

$$
\left(\mathrm{diag}\left(\boldsymbol{\lambda}\right)+\left(\lambda+\boldsymbol{\lambda}^\top\rho^*\right)I_m-\left(\mathbf{z}-\rho^*\right)\boldsymbol{\lambda}^\top\right)J=\left(\mathbf{z}-\rho^*\right)\rho^{*\top}-\mathrm{diag}\left(\rho^*\right)\,.
$$

Using the envelope theorem, we have

$$
\frac{\partial}{\partial\lambda_i}\inf_{\rho\in\mathbb{R}_+^m}\left[\log\left(\lambda+\sum_{l\in[m]}\lambda_l\rho_l\right)+\sum_{l\in[m]}\left(\rho_l-z_l\left(\log\frac{\rho_l}{z_l}+1\right)\right)\right]=\frac{\rho_i^*}{\lambda+\sum_{l\in[m]}\lambda_l\rho_l^*}=\frac{\rho_i^*}{\lambda+\boldsymbol{\lambda}^\top\rho^*}\,.
$$

Recall Equation (6) yields

$$
\frac{\rho_i^*}{\lambda+\boldsymbol{\lambda}^\top\rho^*}=\frac{z_i-\rho_i^*}{\lambda_i}\,.
$$

Differentiating the above equation with respect to $\lambda_j$ gives

$$
\frac{\partial^2}{\partial\lambda_j\partial\lambda_i}\inf_{\rho\in\mathbb{R}_+^m}\left[\log\left(\lambda+\sum_{l\in[m]}\lambda_l\rho_l\right)+\sum_{l\in[m]}\left(\rho_l-z_l\left(\log\frac{\rho_l}{z_l}+1\right)\right)\right]
$$

$$
=\frac{\partial}{\partial\lambda_j}\frac{z_i-\rho_i^*}{\lambda_i}
$$

$$
=\frac{-\lambda_i J_{ij}-(z_i-\rho_i^*)\delta_{ij}}{\lambda_i^2}\,.
$$

It follows that

$$
\lim_{\substack{n,d_i\to+\infty\\ d_i/n\to z_i\\ \|\Pi_i\theta^*\|_2\to\eta_i}}\mathbb{E}\,\mathrm{tr}\left(S\Lambda S\tilde{\Theta}\right)
$$

$$
=\sum_{i\in[m]}\frac{\lambda_i\eta_i^2}{z_i}\sum_{j\in[m]}\lambda_j^2\frac{\lambda_i J_{ij}+(z_i-\rho_i^*)\delta_{ij}}{\lambda_i^2}
$$

$$
=\sum_{i,j\in[m]}\mathbf{q}_i\lambda_j^2\left(J_{ij}+\frac{(z_i-\rho_i^*)\delta_{ij}}{\lambda_i}\right)
$$

$$
=\mathbf{q}^\top\left(\boldsymbol{\lambda}\odot(\mathbf{z}-\rho^*)+J\boldsymbol{\lambda}^{\odot 2}\right)
$$

Putting all three terms together, we have

$$\lim_{\substack{n,d_i\to+\infty \\ d_i/n\to z_i \\ \|\Pi_i\theta^*\|_2\to\eta_i}} B_{\lambda,d,n} = \mathbf{q}^\top\left(\boldsymbol{\lambda}\odot\mathbf{z}\right)-2\mathbf{q}^\top\left(\boldsymbol{\lambda}\odot(\mathbf{z}-\rho^*)\right)+\mathbf{q}^\top\left(\boldsymbol{\lambda}\odot(\mathbf{z}-\rho^*)+J\boldsymbol{\lambda}^{\odot2}\right) = \mathbf{q}^\top\left(\boldsymbol{\lambda}\odot\rho^*+J\boldsymbol{\lambda}^{\odot2}\right).$$

Since $\{B_{\lambda,d,n}\}$ is uniformly bounded and uniformly equicontinuous for $\lambda \in (0,1]$ by Lemma 5, $\{B_{\lambda,d,n}\}$ can be extended continuously to $[0,1]$ and the family of extended functions is still uniformly bounded and uniformly equicontinuous for $\lambda \in [0,1]$. By the Arzela-Ascoli theorem, $\{B_{\lambda,d,n}\}$ converges uniformly to the limit. By the Moore-Osgood theorem, we can exchange the two limits $\lim_{\substack{n,d_i\to+\infty \\ d_i/n\to z_i \\ \|\Pi_i\theta^*\|_2\to\eta_i}}$ and $\lim_{\lambda\to0+}$ and get

$$\lim_{\substack{n,d_i\to+\infty \\ d_i/n\to z_i \\ \|\Pi_i\theta^*\|_2\to\eta_i}} B_{0,d,n} = \lim_{\substack{n,d_i\to+\infty \\ d_i/n\to z_i \\ \|\Pi_i\theta^*\|_2\to\eta_i}}\lim_{\lambda\to0+} B_{\lambda,d,n} = \lim_{\lambda\to0+}\lim_{\substack{n,d_i\to+\infty \\ d_i/n\to z_i \\ \|\Pi_i\theta^*\|_2\to\eta_i}} B_{\lambda,d,n} = \mathbf{q}^\top\left(\boldsymbol{\lambda}\odot\rho^*+J\boldsymbol{\lambda}^{\odot2}\right)|_{\lambda=0}.$$

### D.5 VARIANCE

Define $N = n\lambda I_n + Z^\top\Lambda Z$. Recalling Lemma 6 gives

$$V_{\lambda,d,n}$$
$$=\sigma^2\mathbb{E}\|\Lambda Z N^{-1}\|_2^2$$
$$=\sigma^2\sum_{i=1}^m\lambda_i^2\mathbb{E}\operatorname{tr}\left(Z_iZ_i^\top N^{-2}\right)$$
$$=-\sigma^2\sum_{i=1}^m\lambda_i^2\mathbb{E}\left[\frac{\partial}{\partial\lambda_i}\operatorname{tr}\left(N^{-1}\right)\right].$$

Using Lemma 12, we get

$$\lim_{\substack{n,d_i\to\infty \\ d_i/n\to z_i}} V_{\lambda,d,n}$$
$$=-\sigma^2\sum_{i=1}^m\lambda_i^2\lim_{\substack{n,d_i\to\infty \\ d_i/n\to z_i}}\mathbb{E}\left[\frac{\partial}{\partial\lambda_i}\operatorname{tr}\left(N^{-1}\right)\right]$$
$$=-\sigma^2\sum_{i=1}^m\lambda_i^2\lim_{\substack{n,d_i\to\infty \\ d_i/n\to z_i}}\mathbb{E}\left[\frac{\partial}{\partial\lambda_i}\operatorname{tr}\left(N^{-1}\right)\right]$$
$$=-\sigma^2\sum_{i=1}^m\lambda_i^2\frac{\partial^2}{\partial\lambda_i\partial\lambda}\inf_{\rho\in\mathbb{R}_+^m}\left[\log\left(\lambda+\sum_{j=1}^m\lambda_j\rho_j\right)+\sum_{j=1}^m\left(\rho_j-z_j(\log\frac{\rho_j}{z_j}+1)\right)\right].$$

Using the envelope theorem, we deduce

$$\frac{\partial}{\partial\lambda}\inf_{\rho\in\mathbb{R}_+^m}\left[\log\left(\lambda+\sum_{j=1}^m\lambda_j\rho_j\right)+\sum_{j=1}^m\left(\rho_j-z_j\left(\log\frac{\rho_j}{z_j}+1\right)\right)\right]=\frac{1}{\lambda+\sum_{j=1}^m\lambda_j\rho_j^*}.$$

Then we take $\frac{\partial}{\partial\lambda_i}$ and obtain

$$\frac{\partial^2}{\partial\lambda_i\partial\lambda}\inf_{\rho\in\mathbb{R}_+^m}\left[\log\left(\lambda+\sum_{j=1}^m\lambda_j\rho_j\right)+\sum_{j=1}^m\left(\rho_j-z_j(\log\frac{\rho_j}{z_j}+1)\right)\right]$$
$$=\frac{\partial}{\partial\lambda_i}\frac{1}{\lambda+\sum_{j=1}^m\lambda_j\rho_j^*}$$
$$=-\frac{\rho_i^*+\sum_{j\in[m]}\lambda_jJ_{ji}}{\left(\lambda+\sum_{j\in[m]}\lambda_j\rho_j^*\right)^2}.$$

As a result,

$$\lim_{\substack{n,d_i \to \infty \\ d_i/n \to z_i}} V_{\lambda,d,n} = \sigma^2 \sum_{i=1}^m \lambda_i^2 \frac{\rho_i^* + \sum_{j\in[m]} \lambda_j J_{ji}}{\left(\lambda + \sum_{j\in[m]} \lambda_j \rho_j^*\right)^2} = \sigma^2 \frac{\left(\boldsymbol{\lambda}^{\odot 2}\right)^\top \left(\rho^* + J^\top \boldsymbol{\lambda}\right)}{\left(\lambda + \boldsymbol{\lambda}^\top \rho^*\right)^2}.$$

By Lemma 12, the variance is given by

$$\lim_{\substack{n,d_i \to \infty \\ d_i/n \to z_i}} V_{\lambda,d,n} = \lim_{\substack{n,d_i \to \infty \\ d_i/n \to z_i}} -\sigma^2 \sum_{i=1}^m \lambda_i^2 \mathbb{E}\left[\frac{\partial}{\partial \lambda_i} \operatorname{tr}\left(N^{-1}\right)\right] = \sigma^2 \sum_{i=1}^m r_i^{*2},$$

where $r^*$ solves

$$\sup_{r_t>0} \inf_{r_1,\ldots,r_m>0} \left(2r_t\sqrt{1 + \sum_{i\in[m]} r_i^2} - 2r_t \sum_{i\in[m]} \sqrt{z_i} r_i + \sum_{i\in[m]} \frac{1}{\lambda_i} r_i^2 - \lambda r_t^2\right).$$

Since $\{V_{\lambda,d,n}\}$ is uniformly bounded and uniformly equicontinuous with respect to $\lambda \in (0,1]$ by Lemma 5, $\{V_{\lambda,d,n}\}$ can be extended continuously to $[0,1]$ and the family of extended functions is still uniformly bounded and uniformly equicontinuous. By the Arzela-Ascoli theorem, $\{V_{\lambda,d,n}\}$ converges uniformly to the limit. By the Moore-Osgood theorem, we can exchange the two limits $\lim_{\substack{n,d_i \to \infty \\ d_i/n \to z_i}}$ and $\lim_{\lambda\to0^+}$ and get

$$\lim_{\substack{n,d_i \to \infty \\ d_i/n \to z_i}} \lim_{\lambda\to0^+} V_{\lambda,d,n} = \lim_{\lambda\to0^+} \lim_{\substack{n,d_i \to \infty \\ d_i/n \to z_i}} V_{\lambda,d,n} = \sigma^2 \sum_{i=1}^m r_i^{*2} \mid_{\lambda=0}.$$

# E   PROOF OF THEOREM 2

We use Theorem 1 to prove Theorem 2. As in Theorem 1, let $\mathbf{r}^*$ solve $\min_{r_i\geq0} \max_{r_t\geq0} \vartheta(r_t, \mathbf{r}, \boldsymbol{\lambda})$, where $\vartheta$ is defined in Equation (7). Note that $\vartheta$ is a quadratic function of $r_t$. Define $A = \sqrt{\sum_{i\in[m]} r_i^2 + 1}$, $B = \sum_{i\in[m]} \sqrt{z_i} r_i$, $A^* = \sqrt{\sum_{i\in[m]} r_i^{*2} + 1}$, and $B^* = \sum_{i\in[m]} \sqrt{z_i} r_i^*$. Then $r_t^* = \frac{A-B}{\lambda}$ and we get

$$\min_{r_i\geq0} \max_{r_t\geq0} \vartheta(r_t, \mathbf{r}, \boldsymbol{\lambda}) = \min_{r_i\geq0} \left(\frac{(A-B)^2}{\lambda} + \sum_{i\in[m]} \frac{1}{\lambda_i} r_i^2\right) = \min_{r_i\geq0} \left(\frac{(A-B)^2}{\lambda} + \sum_{i\in[m]} \frac{1}{\lambda_i} r_i^2\right).$$

Taking the partial derivative with respect to $r_i$ gives

$$\frac{\partial}{\partial r_i}\left(\frac{(A-B)^2}{\lambda} + \sum_{i\in[m]} \frac{1}{\lambda_i} r_i^2\right) = 2 \cdot \frac{A-B}{\lambda}\left(\frac{r_i}{A} - \sqrt{z_i}\right) + 2 \cdot \frac{r_i}{\lambda_i}.$$

Setting it to zero gives the optimality condition for $r_i^*$:

$$\frac{A^* - B^*}{\lambda}\left(\frac{r_i^*}{A^*} - \sqrt{z_i}\right) = -\frac{r_i^*}{\lambda_i}, \quad i \in [m]. \tag{48}$$

It follows that

$$\frac{\frac{r_i^*}{A^*} - \sqrt{z_i}}{\frac{r_j^*}{A^*} - \sqrt{z_j}} = \frac{r_i^*/\lambda_i}{r_j^*/\lambda_j}, \quad i,j \in [m].$$

Some algebraic manipulation in the above equation yields

$$\frac{r_i^*}{r_j^*} = \frac{\lambda_i}{\lambda_j} \cdot \frac{\sqrt{z_i} A^* - r_i^*}{\sqrt{z_j} A^* - r_j^*}, \quad i,j \in [m].$$

Define $\mathbf{z} = (z_1, \ldots, z_m)$. Then $\|\mathbf{z}\|_1 = \sum_{i \in [m]} z_i$. By Cauchy–Schwarz inequality, if $d/n \to \sum_{i \in [m]} z_i < 1$

$$B \leq \sqrt{\sum_{i \in [m]} z_i} \, \|\mathbf{r}\|_2 < \|\mathbf{r}\|_2 < \sqrt{\|\mathbf{r}\|_2^2 + 1} = A \,.$$

Thus there does not exist $\mathbf{r}$ such that $A = B$. If $d/n \to \sum_{i \in [m]} z_i > 1$, then $A = B$ is feasible for $\mathbf{r}$. For example, set

$$\mathbf{r} = \frac{1}{\sqrt{(\|\mathbf{z}\|_1 - 1) \|\mathbf{z}\|_1}} \sqrt{\mathbf{z}} \,.$$

We have

$$B = \langle \mathbf{r}, \sqrt{\mathbf{z}} \rangle = \sqrt{\frac{\|\mathbf{z}\|_1}{\|\mathbf{z}\|_1 - 1}}$$

$$A = \sqrt{1 + \|\mathbf{r}\|_2^2} = \sqrt{1 + \frac{1}{\|\mathbf{z}\|_1 - 1}} = B \,.$$

If $\|\mathbf{z}\|_1 > 1$, since $A = B$ is feasible, then

$$\lim_{\lambda \to 0^+} \min_{r_i \geq 0} \left( \frac{(A - B)^2}{\lambda} + \sum_{i \in [m]} \frac{1}{\lambda_i} r_i^2 \right) = \min_{\substack{r_i \geq 0 \\ A = B}} \sum_{i \in [m]} \frac{1}{\lambda_i} r_i^2 \,.$$

If $\|\mathbf{z}\|_1 < 1$, then $A - B$ always holds. To be precise, we have

$$A - B \geq \left( \sqrt{\|\mathbf{r}\|_2^2 + 1} - \|\mathbf{r}\|_2 \right) \vee \left( \left( 1 - \sqrt{\|\mathbf{z}\|_1} \right) \|\mathbf{r}\|_2 \right) \,.$$

If $\|\mathbf{r}\|_2 > 1$, then $\left( 1 - \sqrt{\|\mathbf{z}\|_1} \right) \|\mathbf{r}\|_2 > 1 - \sqrt{\|\mathbf{z}\|_1}$. If $\|\mathbf{r}\|_2 \leq 1$, then $\sqrt{\|\mathbf{r}\|_2^2 + 1} - \|\mathbf{r}\|_2 \geq \sqrt{2} - 1$. Thus there exists a universal constant $C_0 = \left( 1 - \sqrt{\|\mathbf{z}\|_1} \right) \vee \left( \sqrt{2} - 1 \right) > 0$ such that

$$A - B \geq C_0 \,.$$

Recall Equation (48). We have

$$(A^* - B^*) \left( \frac{r_i^*}{A^*} - \sqrt{z_i} \right) = -\frac{\lambda r_i^*}{\lambda_i} \,, \quad i \in [m] \,.$$

Taking $\lim_{\lambda \to 0^+}$, since $A^* - B^* \geq C_0$ does not go to zero, we have

$$\frac{r_i^*}{A^*} - \sqrt{z_i} = 0 \,, \quad i \in [m] \,.$$

Then we get

$$\frac{r_i^{*2}}{1 + \sum_{j \in [m]} r_j^{*2}} = z_i \,, \quad i \in [m] \,.$$

Summing all $i \in [m]$ yields

$$\|\mathbf{z}\|_1 = \frac{\sum_{i \in [m]} r_i^{*2}}{1 + \sum_{i \in [m]} r_i^{*2}} \,.$$

Therefore, we have

$$\lim_{\substack{n, d_i \to +\infty \\ d_i/n \to z_i}} V_{0,d,n} = \sigma^2 \lim_{\lambda \to 0^+} \sum_{i=1}^{m} r_i^{*2} = \sigma^2 \frac{\sum_{i \in [m]} z_i}{1 - \sum_{i \in [m]} z_i} \,.$$

## F   PROOF OF THEOREM 3

Define $A^* = \sqrt{\sum_{i \in [m]} r_i^{*2} + 1}$ and $B^* = \sum_{i \in [m]} \sqrt{z_i} r_i^*$. Equation (15) in Theorem 2 yields

$$\frac{r_1^*}{r_2^*} = \frac{\lambda_1}{\lambda_2} \cdot \frac{\sqrt{z_1} A^* - r_1^*}{\sqrt{z_2} A^* - r_2^*}.$$

Using the constraint $A^* = B^*$, we get

$$\frac{r_1^*}{r_2^*} = \frac{\lambda_1 \left( \sqrt{z_1} B^* - r_1^* \right)}{\lambda_2 \left( \sqrt{z_2} B^* - r_2^* \right)}.$$

Define $q = \frac{r_1^*}{r_2^*}$. We have the following equation

$$q = \frac{\lambda_1 \left( q(z_1 - 1) + \sqrt{z_1 z_2} \right)}{\lambda_2 \left( q\sqrt{z_1 z_2} + z_2 - 1 \right)}.$$

Solving the above equation yields

$$q = \frac{\lambda_1 (z_1 - 1) + \lambda_2 (1 - z_2) + \sqrt{(\lambda_1 (z_1 - 1) + \lambda_2 (1 - z_2))^2 + 4\lambda_1 \lambda_2 z_1 z_2}}{2\lambda_2 \sqrt{z_1 z_2}}. \qquad (49)$$

Here we discard the negative root. Let $x = r_1^{*2} + r_2^{*2} = r_2^{*2} \left( 1 + q^2 \right)$. **??** yields

$$1 + x = r_2^{*2} \left( q\sqrt{z_1} + \sqrt{z_2} \right)^2 = \frac{x}{1 + q^2} \left( q\sqrt{z_1} + \sqrt{z_2} \right)^2.$$

Solving $x$ from the above equation gives

$$x = \frac{q^2 + 1}{q^2(z_1 - 1) + 2q\sqrt{z_1 z_2} + z_2 - 1}.$$

Therefore,

$$\lim_{\substack{n, d_i \to +\infty \\ d_i/n \to z_i}} V_{0,d,n} = \frac{q^2 + 1}{q^2(z_1 - 1) + 2q\sqrt{z_1 z_2} + z_2 - 1}.$$

## G   PROOF OF THEOREM 4

Instead of considering the $\theta^*$ specified in Equation (18), we first consider a Bayesian setting where $\theta^* \sim \mathcal{N} \left( 0, \frac{1}{d} I_d \right)$. Later, we will show that the setup in Equation (18) is asymptotically (as $d_i \to \infty$) equivalent to this Bayesian setting. The precise meaning of equivalence will also be presented later. Our strategy can be divided into two steps. The first step is to show that the Bayes risk of the Bayes estimator is monotonically decreasing in the sample size $n$. The second step is to translate the sample-wise monotonicity of the Bayes estimator to the excess risk of the optimally regularized estimator $\hat{\theta}_{\lambda,d,n}$ in the setup of Equation (18).

Recall that since we are interested in sample-wise monotonicity, we add a subscript $n$ to $X$ and $\mathbf{y}$ (they are defined by Equation (1) in Section 1.1) to emphasize that they consist of $n$ data items. In this Bayesian setting, the likelihood function of $\theta^*$ is

$$L \left( \theta^* \mid X_n, \mathbf{y}_n \right) = \prod_{i \in [n]} L \left( \theta^* \mid x_i, y_i \right) \propto \exp \left( -\frac{\sum_{i \in [n]} (y_i - \langle \theta^*, x_i \rangle)^2}{2\sigma^2} \right) = \exp \left( -\frac{\|X_n \theta^* - \mathbf{y}_n\|_2^2}{2\sigma^2} \right).$$

The density of the prior of $\theta^*$ is proportional to $\exp \left( -\frac{d}{2} \|\theta^*\|_2^2 \right)$. Therefore, the posterior density of $\theta^*$ is given by

$$p \left( \theta^* \mid X_n, \mathbf{y}_n \right) \propto \exp \left( -\frac{d \|\theta^*\|_2^2}{2} - \frac{\|X_n \theta^* - \mathbf{y}_n\|_2^2}{2\sigma^2} \right).$$

As a result, the posterior distribution of $\theta^*$ is Gaussian. The Bayes estimator is

$$\hat{\theta}_{\text{Bayes}}\left(X_n, \mathbf{y}_n\right) = \arg\min_{\theta} \mathbb{E}_{\theta^* \sim p(\theta^* | X_n, \mathbf{y}_n)} \|\theta - \theta^*\|_{\Sigma}^2 .$$

Taking the derivative with respect to $\theta$ gives

$$\frac{\partial}{\partial \theta} \mathbb{E}_{\theta^* \sim p(\theta^* | X_n, \mathbf{y}_n)} \|\theta - \theta^*\|_{\Sigma}^2 = 2\Sigma\left(\theta - \theta^*\right) .$$

Setting the above equation to zero yields $\Sigma\left(\hat{\theta}_{\text{Bayes}}\left(X_n, \mathbf{y}_n\right) - \mathbb{E}_{\theta^* \sim p(\theta^* | X_n, \mathbf{y}_n)}\theta^*\right) = 0$ and therefore

$$\hat{\theta}_{\text{Bayes}}\left(X_n, \mathbf{y}_n\right) = \mathbb{E}_{\theta^* \sim p(\theta^* | X_n, \mathbf{y}_n)}\theta^* = \mathbb{E}\left[\theta^* \mid X_n, \mathbf{y}_n\right] = \arg\min_{\theta}\left(d\|\theta^*\|_2^2 + \frac{\|X_n\theta^* - \mathbf{y}_n\|_2^2}{\sigma^2}\right) .$$

The final equality is because the posterior mean of a Gaussian distribution equals its mode.

Define the Bayes risk

$$R_n \triangleq \mathbb{E}_{\theta^* \sim \mathcal{N}(0, \frac{1}{d}I_d), X_n, \mathbf{y}_n}\left[\left\|\hat{\theta}_{\text{Bayes}}\left(X_n, \mathbf{y}_n\right) - \theta^*\right\|_{\Sigma}^2\right] .$$

Write $\mathcal{X} = \mathbb{R}^d$ and $\mathcal{Y} = \mathbb{R}$. Define

$$R_n' \triangleq \inf_{\hat{\theta}: \mathcal{X}^n \times \mathcal{Y}^n \to \mathbb{R}} \mathbb{E}_{\theta^* \sim \mathcal{N}(0, \frac{1}{d}I_d), X_n, \mathbf{y}_n}\left\|\hat{\theta}\left(X_n, \mathbf{y}_n\right) - \theta^*\right\|_{\Sigma}^2 .$$

We have

$$
\begin{aligned}
R_n' &= \inf_{\hat{\theta}: \mathcal{X}^n \times \mathcal{Y}^n \to \mathbb{R}^d} \mathbb{E}_{\theta^* \sim \mathcal{N}(0, \frac{1}{d}I_d), X_n, \mathbf{y}_n}\left\|\Sigma^{1/2}\hat{\theta}\left(X_n, \mathbf{y}_n\right) - \Sigma^{1/2}\theta^*\right\|_2^2 \\
&= \inf_{\hat{\theta}: \mathcal{X}^n \times \mathcal{Y}^n \to \mathbb{R}^d} \mathbb{E}_{\theta^* \sim \mathcal{N}(0, \frac{1}{d}I_d), X_n, \mathbf{y}_n}\left\|\hat{\theta}\left(X_n, \mathbf{y}_n\right) - \Sigma^{1/2}\theta^*\right\|_2^2 \\
&= \mathbb{E}_{\theta^* \sim \mathcal{N}(0, \frac{1}{d}I_d), X_n, \mathbf{y}_n}\left\|\mathbb{E}\left[\Sigma^{1/2}\theta^* \mid X_n, \mathbf{y}_n\right] - \Sigma^{1/2}\theta^*\right\|_2^2 \\
&= \mathbb{E}_{\theta^* \sim \mathcal{N}(0, \frac{1}{d}I_d), X_n, \mathbf{y}_n}\left\|\Sigma^{1/2}\hat{\theta}_{\text{Bayes}}\left(X_n, \mathbf{y}_n\right) - \Sigma^{1/2}\theta^*\right\|_2^2 \\
&= R_n .
\end{aligned}
$$

where the third equality is because the conditional expectation minimizes the $\ell^2$ loss. Next, we want to show that $R_{n+1} \le R_n$, i.e., the Bayes risk of the Bayes estimator is monotonically decreasing in the sample size $n$.

$$
\begin{aligned}
R_{n+1} &= \inf_{\hat{\theta}: \mathcal{X}^{n+1} \times \mathcal{Y}^{n+1} \to \mathbb{R}^d} \mathbb{E}_{\theta^* \sim \mathcal{N}(0, \frac{1}{d}I_d), X_{n+1}, \mathbf{y}_{n+1}}\left[\left\|\hat{\theta}\left(X_{n+1}, \mathbf{y}_{n+1}\right) - \theta^*\right\|_{\Sigma}^2\right] \\
&\le \inf_{\hat{\theta}: \mathcal{X}^n \times \mathcal{Y}^n \to \mathbb{R}} \mathbb{E}_{\theta^* \sim \mathcal{N}(0, \frac{1}{d}I_d), X_{n+1}, \mathbf{y}_{n+1}}\left[\left\|\hat{\theta}\left(X_n, \mathbf{y}_n\right) - \theta^*\right\|_{\Sigma}^2\right] \\
&= R_n .
\end{aligned}
$$

Then we want to show that $R_n$ equals the Bayes risk of the optimally regularized estimator $\hat{\theta}_{\lambda, n, d}$:

$$R_n = \inf_{\lambda \ge 0} \mathbb{E}_{\theta^* \sim \mathcal{N}(0, \frac{1}{d}I_d), X_n, \mathbf{y}_n}\left\|\hat{\theta}_{\lambda, n, d} - \theta^*\right\|_{\Sigma}^2 .$$

Since $R_n = \inf_{\hat{\theta}: \mathcal{X}^n \times \mathcal{Y}^n \to \mathbb{R}^d} \mathbb{E}_{\theta^*, X_n, \mathbf{y}_n}\left\|\hat{\theta}\left(X_n, \mathbf{y}_n\right) - \theta^*\right\|_{\Sigma}^2$, we get

$$R_n \le \inf_{\lambda \ge 0} \mathbb{E}_{\theta^* \sim \mathcal{N}(0, \frac{1}{d}I_d), X_n, \mathbf{y}_n}\left\|\hat{\theta}_{\lambda, n, d} - \theta^*\right\|_{\Sigma}^2 .$$

On the other hand, recalling $\hat{\theta}_{\text{Bayes}}(X_n, \mathbf{y}_n) = \arg\min_\theta \left( d \|\theta\|_2^2 + \frac{\|X_n\theta - \mathbf{y}_n\|_2^2}{\sigma^2} \right) = \hat{\theta}_{\frac{\sigma^2 d}{n}, n, d}$ and $R_n = \mathbb{E}_{\theta^* \sim \mathcal{N}(0, \frac{1}{d} I_d), X_n, \mathbf{y}_n} \left[ \left\| \hat{\theta}_{\text{Bayes}}(X_n, \mathbf{y}_n) - \theta^* \right\|_\Sigma^2 \right]$, we deduce

$$R_n \geq \inf_{\lambda \geq 0} \mathbb{E}_{\theta^* \sim \mathcal{N}(0, \frac{1}{d} I_d), X_n, \mathbf{y}_n} \left\| \hat{\theta}_{\lambda, n, d} - \theta^* \right\|_\Sigma^2 .$$

Therefore we deduce $R_n = \inf_{\lambda \geq 0} \mathbb{E}_{\theta^* \sim \mathcal{N}(0, \frac{1}{d} I_d), X_n, \mathbf{y}_n} \left\| \hat{\theta}_{\lambda, n, d} - \theta^* \right\|_\Sigma^2$. As a result, we establish sample-wise monotonicity of the Bayes risk of optimal regularized $\hat{\theta}_{\lambda, n, d}$:

$$R_{n+1} = \inf_{\lambda \geq 0} \mathbb{E}_{\theta^*, X_{n+1}, \mathbf{y}_{n+1}} \left\| \hat{\theta}_{\lambda, n+1, d} - \theta^* \right\|_\Sigma^2 \leq \inf_{\lambda \geq 0} \mathbb{E}_{\theta^*, X_n, \mathbf{y}_n} \left\| \hat{\theta}_{\lambda, n, d} - \theta^* \right\|_\Sigma^2 = R_n . \quad (50)$$

In what follows, we show that if $\theta^*$ is given by Equation (18), the excess risk of $\hat{\theta}_{\lambda, n, d}$ is asymptotically equal to its Bayes risk when $\theta^* \sim \mathcal{N}(0, \frac{1}{d} I_d)$:

$$\lim_{d_i \to \infty} \left| \mathbb{E}_{X_n, \mathbf{y}_n} \left\| \hat{\theta}_{\lambda, n, d} - \theta^* \right\|_\Sigma^2 - \mathbb{E}_{\theta^* \sim \mathcal{N}(0, \frac{1}{d} I_d)} \mathbb{E}_{X_n, \mathbf{y}_n} \left\| \hat{\theta}_{\lambda, n, d} - \theta^* \right\|_\Sigma^2 \right| = 0 .$$

We abuse the notation in the above equation. The $\theta^*$ in $\mathbb{E}_{X_n, \mathbf{y}_n} \left\| \hat{\theta}_{\lambda, n} - \theta^* \right\|_\Sigma^2$ satisfies Equation (18), while the $\theta^*$ in $\mathbb{E}_{\theta^* \sim \mathcal{N}(0, \frac{1}{d} I_d)} \mathbb{E}_{X_n, \mathbf{y}_n} \left\| \hat{\theta}_{\lambda, n, d} - \theta^* \right\|_\Sigma^2$ follows a normal distribution $\mathcal{N}(0, \frac{1}{d} I_d)$. By Lemma 5 and Lemma 6, if $\Sigma = P \Lambda P^\top$ and $\theta' = P^\top \theta^*$ are as defined in Table 1 (where $P$ is an orthogonal matrix and $\Lambda = \text{diag}(\lambda_1 I_{d_1}, \ldots, \lambda_m I_{d_m}) \in \mathbb{R}^{d \times d}$ is a diagonal matrix), for fixed $\theta^*$ we have

$$\mathbb{E}_{X_n, \mathbf{y}_n} \left\| \hat{\theta}_{\lambda, n} - \theta^* \right\|_\Sigma^2 = \mathbb{E}_{X_n, \mathbf{y}_n} \left[ \|\Lambda^{1/2} \left( I_d + \frac{1}{n\lambda} \Lambda^{1/2} Z Z^\top \Lambda^{1/2} \right)^{-1} \theta'\|_2^2 \right]$$
$$+ \sigma^2 \mathbb{E}_{X_n, \mathbf{y}_n} \left[ \|\Lambda Z \left( \lambda n I_n + Z^\top \Lambda Z \right)^{-1} \|_2^2 \right] ,$$

where every entry of $Z \in \mathbb{R}^{d \times n}$ follows i.i.d. $\mathcal{N}(0, 1)$. If $\theta^* \sim \mathcal{N}(0, \frac{1}{d} I_d)$, we have $\theta' \sim \mathcal{N}(0, \frac{1}{d} I_d)$. Since the variance term $\sigma^2 \mathbb{E}_{X_n, \mathbf{y}_n} \left[ \|\Lambda Z \left( \lambda n I_n + Z^\top \Lambda Z \right)^{-1} \|_2^2 \right]$ does not depend on $\theta^*$, the two variance terms cancel out and we get

$$\mathbb{E}_{X_n, \mathbf{y}_n} \left\| \hat{\theta}_{\lambda, n, d} - \theta^* \right\|_\Sigma^2 - \mathbb{E}_{\theta^* \sim \mathcal{N}(0, \frac{1}{d} I_d)} \mathbb{E}_{X_n, \mathbf{y}_n} \left\| \hat{\theta}_{\lambda, n, d} - \theta^* \right\|_\Sigma^2$$
$$= \mathbb{E}_{X_n, \mathbf{y}_n} \left[ \|\Lambda^{1/2} \left( I_d + \frac{1}{n\lambda} \Lambda^{1/2} Z Z^\top \Lambda^{1/2} \right)^{-1} \theta'\|_2^2 \right]$$
$$- \mathbb{E}_{X_n, \mathbf{y}_n, \theta' \sim \mathcal{N}(0, \frac{1}{d} I_d)} \left[ \|\Lambda^{1/2} \left( I_d + \frac{1}{n\lambda} \Lambda^{1/2} Z Z^\top \Lambda^{1/2} \right)^{-1} \theta'\|_2^2 \right]$$

For $U \sim \bigoplus_{i \in [m]} \text{Unif}(O(d_i))$, we have

$$\left( I_d + \frac{1}{n\lambda} \Lambda^{1/2} Z Z^\top \Lambda^{1/2} \right)^{-1} \stackrel{d}{=} \left( I_d + \frac{1}{n\lambda} \Lambda^{1/2} U Z Z^\top U^\top \Lambda^{1/2} \right)^{-1} = U \left( I_d + \frac{1}{n\lambda} \Lambda^{1/2} Z Z^\top \Lambda^{1/2} \right)^{-1} U^\top .$$

By Lemma 13, for $\theta^*$ (and thereby $\theta'$) specified in Equation (18), we get

$$\mathbb{E}_{X_n, \mathbf{y}_n} \left[ \|\Lambda^{1/2} \left( I_d + \frac{1}{n\lambda} \Lambda^{1/2} Z Z^\top \Lambda^{1/2} \right)^{-1} \theta'\|_2^2 \right] = \mathbb{E}_{X_n, \mathbf{y}_n, \phi} \left[ \|\Lambda^{1/2} \left( I_d + \frac{1}{n\lambda} \Lambda^{1/2} Z Z^\top \Lambda^{1/2} \right)^{-1} \phi\|_2^2 \right] ,$$

where

$$\phi \sim \bigoplus_{i \in [m]} \text{Unif}\left( S^{d_i - 1}(\|\theta_i'\|_2) \right) = \bigoplus_{i \in [m]} \text{Unif}\left( S^{d_i - 1}\left( \sqrt{d_i/d} \right) \right) .$$

In the Bayesian setting, if $\theta' \sim \mathcal{N}(0, \frac{1}{d}I_d)$, then $U^\top\theta' \sim \mathcal{N}(0, \frac{1}{d}I_d)$. We have

$$\mathbb{E}_{\theta' \sim \mathcal{N}(0,\frac{1}{d}I_d), X_n, \mathbf{y}_n} \left\| \hat{\theta}_{\lambda,n} - \theta^* \right\|_\Sigma^2 = \mathbb{E}_{\theta' \sim \mathcal{N}(0,\frac{1}{d}I_d), X_n, \mathbf{y}_n} \left[ \left\| \Lambda^{1/2} U \left( I_d + \frac{1}{n\lambda} \Lambda^{1/2} ZZ^\top \Lambda^{1/2} \right)^{-1} U^\top\theta' \right\|_2^2 \right]$$

$$= \mathbb{E}_{X_n, \mathbf{y}_n, \psi} \left[ \left\| \Lambda^{1/2} \left( I_d + \frac{1}{n\lambda} \Lambda^{1/2} ZZ^\top \Lambda^{1/2} \right)^{-1} \psi \right\|_2^2 \right],$$

where $\psi = U^\top\theta' \sim \mathcal{N}(0, \frac{1}{d}I_d)$.

Next, we want to couple $\phi$ and $\psi$. Let $s_i \overset{\text{i.i.d.}}{\sim} \mathrm{Unif}\left(S^{d_i-1}(1)\right)$, $h_i \overset{\text{i.i.d.}}{\sim} \chi^2(d_i)$, and define

$$\phi = \begin{bmatrix} \sqrt{d_1/d}s_1 \\ \vdots \\ \sqrt{d_m/d}s_m \end{bmatrix}, \quad \psi = \begin{bmatrix} \sqrt{h_1/d}s_1 \\ \vdots \\ \sqrt{h_m/d}s_m \end{bmatrix}.$$

We have $\|\phi\|_2 = 1$ and

$$\|\psi\|_2 = \sqrt{\sum_{i=1}^m \frac{h_i}{d}} = \sqrt{\sum_{i=1}^m \frac{d_i}{d} \cdot \frac{h_i}{d_i}},$$

$$\|\phi - \psi\|_2 = \sqrt{\sum_{i=1}^m \frac{d_i}{d}\left(1 - \sqrt{\frac{h_i}{d_i}}\right)^2}.$$

By the strong law of large numbers, $\lim_{d_i \to +\infty} h_i/d_i = 1$ almost surely. Thus we get $\lim_{d_i \to +\infty, d_i/d \to \nu_i} \|\psi\|_2 = \sqrt{\sum_{i=1}^m \nu_i}$ and $\lim_{d_i \to +\infty, d_i/d \to \nu_i} \|\phi - \psi\|_2 = 0$ almost surely (recall that we will let $d_i \to +\infty$ and $d_i/d \to \nu_i$ for some constant $\nu_i > 0$. ). Because $\left\| \Lambda^{1/2} \right\|_2 \lesssim 1$ and $\left\| \left( I_d + \frac{1}{n\lambda} \Lambda^{1/2} ZZ^\top \Lambda^{1/2} \right)^{-1} \right\|_2 \leq \|I_d\|_2 = 1$, we bound the norm of $Q \triangleq \Lambda^{1/2} \left( I_d + \frac{1}{n\lambda} \Lambda^{1/2} ZZ^\top \Lambda^{1/2} \right)^{-1}$ as follows

$$\|Q\|_2 \leq \left\| \Lambda^{1/2} \right\|_2 \left\| \left( I_d + \frac{1}{n\lambda} \Lambda^{1/2} ZZ^\top \Lambda^{1/2} \right)^{-1} \right\|_2 \lesssim 1.$$

It follows that

$$\left| \mathbb{E}_{X_n, \mathbf{y}_n, \phi} \left[ \|Q\phi\|_2^2 \right] - \mathbb{E}_{X_n, \mathbf{y}_n, \psi} \left[ \|Q\psi\|_2^2 \right] \right|$$

$$\leq \mathbb{E}_{X_n, \mathbf{y}_n, \phi, \psi} \left| \|Q\phi\|_2^2 - \|Q\psi\|_2^2 \right|$$

$$= \mathbb{E}_{X_n, \mathbf{y}_n \phi, \psi} \left( \|Q\phi\|_2 + \|Q\psi\|_2 \right) \left| \|Q\phi\|_2 - \|Q\psi\|_2 \right|$$

$$\lesssim \mathbb{E}_{X_n, \mathbf{y}_n \phi, \psi} \left[ (\|\phi\|_2 + \|\psi\|_2) \|Q(\phi - \psi)\|_2 \right]$$

$$\lesssim \mathbb{E}_{X_n, \mathbf{y}_n \phi, \psi} \|\phi - \psi\|_2,$$

where the last inequality is because $\|\phi\|_2 + \|\psi\|_2 \lesssim 1$ for all sufficiently large $d_i$. We know that $\lim_{d_i \to +\infty, d_i/d \to \nu_i} \|\phi - \psi\|_2 = 0$ almost surely. To apply Lebesgue's dominated convergence theorem, we need to find a dominating integrable random variable. In fact, $1 + \|\psi\|_2$ dominates $\|\phi - \psi\|_2$:

$$\|\phi - \psi\|_2 \leq \|\phi\|_2 + \|\psi\|_2 = 1 + \|\psi\|_2.$$

It is integrable because $\mathbb{E}\|\psi\|_2 = \mathbb{E}\left[\sqrt{\frac{\chi^2(d)}{d}}\right] \leq \sqrt{\frac{\mathbb{E}[\chi^2(d)]}{d}} = 1$. Application of Lebesgue's dominated convergence theorem yields

$$\lim_{d_i \to +\infty, d_i/d \to \nu_i} \left| \mathbb{E}_{X_n, \mathbf{y}_n, \phi} \left[ \|Q\phi\|_2^2 \right] - \mathbb{E}_{X_n, \mathbf{y}_n, \psi} \left[ \|Q\psi\|_2^2 \right] \right| = 0.$$

Therefore, we conclude that

$$\lim_{d_i \to +\infty, d_i/d \to \nu_i} \left| \mathbb{E}_{X_n, \mathbf{y}_n} \left\| \hat{\theta}_{\lambda,n} - \theta^* \right\|_\Sigma^2 - \mathbb{E}_{\theta^* \sim \mathcal{N}(0, \frac{1}{d} I_d)} \mathbb{E}_{X_n, \mathbf{y}_n} \left\| \hat{\theta}_{\lambda,n,d} - \theta^* \right\|_\Sigma^2 \right| = 0$$

and this convergence is uniform in $n$ and $\lambda \in (0, \infty)$. It follows that

$$\lim_{\substack{n, d_i \to \infty \\ n/d_i \to \gamma_i}} \left| \mathbb{E}_{X_n, \mathbf{y}_n} \left\| \hat{\theta}_{\lambda,n,d} - \theta^* \right\|_\Sigma^2 - \mathbb{E}_{\theta^* \sim \mathcal{N}(0, \frac{1}{d} I_d)} \mathbb{E}_{X_n, \mathbf{y}_n} \left\| \hat{\theta}_{\lambda,n,d} - \theta^* \right\|_\Sigma^2 \right| = 0$$

and this convergence is uniform in $\lambda \in (0, \infty)$.

By Lemma 8 (the proof is similar when we replace $\alpha \to +\infty$ by $n, d_i \to \infty, n/d_i \to \gamma_i$), we have

$$\lim_{\substack{n, d_i \to \infty \\ n/d_i \to \gamma_i}} \left| \inf_{\lambda > 0} \mathbb{E}_{X_n, \mathbf{y}_n} \left\| \hat{\theta}_{\lambda,n,d} - \theta^* \right\|_\Sigma^2 - \inf_{\lambda > 0} \mathbb{E}_{\theta^* \sim \mathcal{N}(0, \frac{1}{d} I_d)} \mathbb{E}_{X_n, \mathbf{y}_n} \left\| \hat{\theta}_{\lambda,n,d} - \theta^* \right\|_\Sigma^2 \right| = 0 . \quad (51)$$

Define $f_\alpha(\lambda) = \mathbb{E}_{\theta^* \sim \mathcal{N}(0, \frac{1}{d} I_d)} \mathbb{E}_{X_n, \mathbf{y}_n} \left\| \hat{\theta}_{\lambda,n,d} - \theta^* \right\|_\Sigma^2$. We use $\alpha$ to denote the indices $n, d_i$. By Lemma 5 and Lemma 6, we have

$$\mathbb{E}_{\theta^* \sim \mathcal{N}(0, \frac{1}{d} I_d)} \mathbb{E}_{X_n, \mathbf{y}_n} \left\| \hat{\theta}_{\lambda,n,d} - \theta^* \right\|_\Sigma^2 \lesssim \mathbb{E}_{\theta^* \sim \mathcal{N}(0, \frac{1}{d} I_d)} \| \theta^* \|_2^2 = 1 .$$

Therefore $\{ f_\alpha(\lambda) \}$ is uniformly bounded for $\lambda > 0$. Since $\left| \frac{d}{d\lambda} \mathbb{E}_{X_n, \mathbf{y}_n} \left\| \hat{\theta}_{\lambda,n} - \theta^* \right\|_\Sigma^2 \right| \lesssim \| \theta^* \|_2^2$ and $\mathbb{E}_{\theta^* \sim \mathcal{N}(0, \frac{1}{d} I_d)} \| \theta^* \|_2^2 = 1$, we have

$$\left| \frac{d}{d\lambda} \mathbb{E}_{\theta^* \sim \mathcal{N}(0, \frac{1}{d} I_d)} \mathbb{E}_{X_n, \mathbf{y}_n} \left\| \hat{\theta}_{\lambda,n} - \theta^* \right\|_\Sigma^2 \right| = \left| \mathbb{E}_{\theta^* \sim \mathcal{N}(0, \frac{1}{d} I_d)} \frac{d}{d\lambda} \mathbb{E}_{X_n, \mathbf{y}_n} \left\| \hat{\theta}_{\lambda,n} - \theta^* \right\|_\Sigma^2 \right| \lesssim 1 .$$

As a result, $\{ f_\alpha(\lambda) \}$ is uniformly equicontinuous for $\lambda > 0$, and in particular $\lambda \in (0, M]$ for any $M > 0$. Therefore $\{ f_\alpha(\lambda) \}$ can be extended continuously to $[0, M]$ and the family of extended functions is still uniformly bounded and uniformly equicontinuous. Recall that if $\theta^* \sim \mathcal{N}(0, \frac{1}{d} I_d)$, we have $\theta' \sim \mathcal{N}(0, \frac{1}{d} I_d)$. As in Equation (19), write $\theta'$ in a row-partitioned form

$$\theta' = \begin{pmatrix} \theta'_1 \\ \vdots \\ \theta'_m \end{pmatrix} ,$$

where $\theta'_i \in \mathbb{R}^{d_i}$. Then $\| \Pi_i \theta^* \|_2 = \| \theta'_i \|_2 \sim \sqrt{\frac{\chi^2(d_i)}{d}} = \sqrt{\frac{\chi^2(d_i)}{d_i} \cdot \frac{d_i}{d}} \to \sqrt{\nu_i}$ as $n, d_i \to +\infty$ and $n/d_i \to \gamma_i$, where $\nu_i = \left( \gamma_i \sum_{j \in [m]} \frac{1}{\gamma_j} \right)^{-1}$. By Theorem 1, $\{ f_\alpha(\lambda) \}$ converges pointwise, say, to $h(\lambda, \gamma_1, \ldots, \gamma_m)$. By the Arzela-Ascoli theorem, $\lim_\alpha f_\alpha(\lambda) = h(\lambda, \gamma_1, \ldots, \gamma_m)$ uniformly on $\lambda \in [0, M]$. Therefore, as $n, d_i \to \infty$ and $n/d_i \to \gamma_i$, by Lemma 8, we have

$$\inf_{\lambda \in [0, M]} \mathbb{E}_{\theta^* \sim \mathcal{N}(0, \frac{1}{d} I_d)} \mathbb{E}_{X_n, \mathbf{y}_n} \left\| \hat{\theta}_{\lambda,n} - \theta^* \right\|_\Sigma^2 \to \inf_{\lambda \in [0, M]} h(\lambda, \gamma_1, \ldots, \gamma_m) .$$

Recalling $\hat{\theta}_{\text{Bayes}} (X_n, \mathbf{y}_n) = \arg\min_\theta \left( d \| \theta \|_2^2 + \frac{\| X_n \theta - \mathbf{y}_n \|_2^2}{\sigma^2} \right) = \hat{\theta}_{\frac{\sigma^2 d}{n}, n}$ and

$$R_n = \mathbb{E}_{\theta^* \sim \mathcal{N}(0, \frac{1}{d} I_d), X_n, \mathbf{y}_n} \left[ \left\| \hat{\theta}_{\text{Bayes}} (X_n, \mathbf{y}_n) - \theta^* \right\|_\Sigma^2 \right] = \inf_{\lambda \geq 0} \mathbb{E}_{\theta^* \sim \mathcal{N}(0, \frac{1}{d} I_d), X_n, \mathbf{y}_n} \left\| \hat{\theta}_{\lambda,n,d} - \theta^* \right\|_\Sigma^2 .$$

For all $M > C_M := 2\sigma^2 \sum_{i \in [m]} \frac{1}{\gamma_i} \geq \frac{\sigma^2 d}{n}$ (recall $\frac{d}{n} \to \sum_{i \in [m]} \frac{1}{\gamma_i}$), we have

$$\inf_{\lambda \in [0, M]} \mathbb{E}_{\theta^* \sim \mathcal{N}(0, \frac{1}{d} I_d), X_n, \mathbf{y}_n} \left\| \hat{\theta}_{\lambda,n,d} - \theta^* \right\|_\Sigma^2 = \inf_{\lambda \geq 0} \mathbb{E}_{\theta^* \sim \mathcal{N}(0, \frac{1}{d} I_d)} \mathbb{E}_{X_n, \mathbf{y}_n} \left\| \hat{\theta}_{\lambda,n,d} - \theta^* \right\|_\Sigma^2$$

$$\to \inf_{\lambda \in [0, M]} h(\lambda, \gamma_1, \ldots, \gamma_m) .$$

The uniqueness of limits implies that $\inf_{\lambda \in [0,M]} h(\lambda, \gamma_1, \ldots, \gamma_m)$ is independent of $M$ as long as $M > \sigma^2$. As a result, if $M > C_M$, we have $\inf_{\lambda \in [0,M]} h(\lambda, \gamma_1, \ldots, \gamma_m) = \inf_{\lambda \geq 0} h(\lambda, \gamma_1, \ldots, \gamma_m)$, which yields

$$\inf_{\lambda \geq 0} \mathbb{E}_{\theta^* \sim \mathcal{N}(0, \frac{1}{d} I_d), X_n, \mathbf{y}_n} \left\| \hat{\theta}_{\lambda, n, d} - \theta^* \right\|_{\Sigma}^2 \to \inf_{\lambda \geq 0} h(\lambda, \gamma_1, \ldots, \gamma_m). \tag{52}$$

Equation (50) implies $\inf_{\lambda \geq 0} h(\lambda, \gamma_1, \ldots, \gamma_m)$ is decreasing in every $\gamma_i$. Combining Equation (51) and Equation (52) gives

$$\inf_{\lambda \geq 0} \mathbb{E}_{X_n, \mathbf{y}_n} \left\| \hat{\theta}_{\lambda, n} - \theta^* \right\|_{\Sigma}^2 \to \inf_{\lambda \geq 0} h(\lambda, \gamma_1, \ldots, \gamma_m).$$

