# OpenReview forum: "Spectral Multiplicity Entails Sample-wise Multiple Descent"
_ICLR.cc/2022/Conference — ICLR 2022 Submitted_

### Official Review · Reviewer_6s7y · 2021-10-31

**Correctness:** 4
**Technical Novelty And Significance:** 1
**Empirical Novelty And Significance:** Not applicable
**Recommendation:** 3
**Confidence:** 4

**Main Review:**

This authors devoted a lot of effort to rigorously prove the generalization error formulas in the proportional limit.
Unfortunately, my impression is that most of the theorems in this submission are essentially re-derivation of existing results from the following papers (some of these prior results actually allow for weaker assumptions):
* Dobriban and Wager, Annals of Statistics 2017. https://arxiv.org/pdf/1507.03003.pdf.
* Wu and Xu, NeurIPS 2020. https://arxiv.org/pdf/2006.05800.pdf.
* Richards et al., AISTATS 2021. https://arxiv.org/pdf/2006.06386.pdf.
* Amari et al., ICLR 2021. https://arxiv.org/pdf/2006.10732.pdf.

Note that all of these papers have been publicly available for more than one year, so I think it is no longer the case that this new submission can be considered as concurrent work.

Specifically,
1. in terms of the risk formula, [Wu and Xu] and [Richards et al.] derived the limiting risk under the same problem setup for general regularization parameter $\lambda$, whereas [Amari et al.] focused on the bias-variance decomposition in the ridgeless limit. All these works built upon a generalized Marchenko-Pastur theorem, as well as [Dobriban and Wager], which showed that the limiting variance can be related to the Stieltjes transform and its derivative (sometimes referred to as the "derivative trick").
These results cover Theorem 1 and 2 in the current submission.
2.  The two block ($m=2$) case was already studied in Section 3 of [Richards et al.], Appendix A.4 of [Amari et al.], and Section 5 of [Wu and Xu]. In particular, the closed-form expression of the Stieltjes transform (and therefore the limiting variance) has been derived in Appendix A.6 of [Richards et al.] and Appendix D.10 of [Amari et al.], and the resulting "multiple descent" has also been reported. These results cover Theorem 3 and Corollary 2 in the current submission.
3. Theorem 4 in this submission is analogous to Proposition 7 in [Wu and Xu], which states that under an isotropic prior on the true parameters $\theta^*$, the optimally-tuned ridge regression estimator has monotonically decreasing generalization error with respect to $\gamma=n/d$, and thus avoids multiple descent; this is basically equivalent to the projection formulation in Theorem 4 (the bias term in the generalization error only depends on the projection of $\theta^*$ onto the eigenspace of $\Sigma$).


**Additional Comments**

- At a high level, I personally do not find counting the number of peaks in the risk curve to be that interesting of a research problem, as least in the case of minimum $\ell_2$-norm interpolation. It is rather intuitive that one can engineer the features such that the sample covariance matrix becomes ill-conditioned once in a while, and it is not clear if this specific phenomenon has wider implications in practical machine learning problems.

- (minor) Note that the Gaussian assumption has already been relaxed to bounded moments in the Stieltjes transform-based analysis, e.g., [Dobriban and Wager] and [Wu and Xu]. Hence I'm not sure if there's any benefit of introducing CGMT to this problem (which limits the analysis to Gaussian features).

- (minor) In Corollary 2, why is $\varrho\to\infty$ a well-defined limit? Intuitively, wouldn't this violate the condition that the eigenvalues do not depend on $d$ which goes to infinity?

**Summary Of The Paper:**

Derived the asymptotic generalization error (squared loss) of the minimum $\ell_2$-norm interpolator under general Gaussian design. The precise asymptotic formulas reveal that the risk curve can be non-monotonic and exhibit multiple peaks (in the variance term), depending on the spectral properties of the feature covariance matrix $\Sigma$.

**Summary Of The Review:**

Due to the significant overlap with prior results, I cannot recommend acceptance. I'm happy to adjust my score if the authors can clarify how the current submission differs from (and improves upon) all the aforementioned papers.

---

> ### Author Response · Authors · 2021-11-23
> **Response to Reviewer 6s7y**
>
> Thank you so much for your time and insightful feedback! We have incorporated the changes into the revised paper, and address the issues in detail here.
>
> Q1: Our contributions and comparison to [Wu and Xu], [Richards et al.], [Amari et al.], and [Dobriban and Wager].
>
> A1: We would like to thank the reviewer for the references. In what follows, we highlight our contributions and differences from them. We cited the papers and included the following discussion in the revision.
>
> 1. Our major contribution is providing a rigorous proof for the existence of sample-wise (test error vs. the number of training samples) double and multiple descent in linear regression. However, [Richards et al.] only mentioned parameter-wise double descent (test error vs. model capacity) in their related work (Section 1.1 Related Literature). [Amari et al.] only mentioned epoch-wise (test error vs. training time) double descent in Appendix A.2. Neither [Richards et al.] nor [Amari et al.] mentioned multiple descent or sample-wise double/multiple descent.
>
> 2. We made and theoretically proved the observation that an ill-conditioned covariance matrix is a sufficient condition for the existence of sample-wise multiple descent. To the best of our knowledge, our work is the first paper that pointed this out.
>
> 3. We solved the Stieltjes transform explicitly and derived explicit formulae for the risk and variance in our setup. In addition, we also provided rigorous treatment to the ridgeless setting ($\lambda=0$) and also obtained explicit formulae for it.
>
> 4. There is another difference between our paper and the papers that the reviewer mentioned.  All of [Wu and Xu], [Richards et al.], [Amari et al.], and [Dobriban and Wager] assumed a prior on the true linear model $\beta^*$ and takes expectation over the prior. In our paper, we do NOT assume a prior on the true linear model and our risk does NOT take the expectation over a random true linear model.
>
> - For [Wu and Xu], see their abstract: "Under ... anisotropic prior on the true coefficients $E[\beta_* \beta_*^\top] =\Sigma_\beta$,
> we provide an exact characterization of the prediction risk ..." and the introduction "We generalize the previous random effects hypothesis and place the following prior on the true coefficients: $E\beta_*\beta_*^\top = \Sigma_{\beta}$".
>
> - For [Richards et al.], see their Equation (3): $E[\beta^*] = 0$, $E[\beta^* (\beta^*)^\top] = \frac{r^2}{d}\Phi(\Sigma)$.
>
> - For [Amari et al.], see the first paragraph of Section 3.2: "we consider a more general prior on $\theta^*: E[\theta^* \theta^{*\top}] = d^{-1} \Sigma_{\theta}$".
>
> - For [Dobriban and Wager], see the second paragraph of Section 1.1 (the paragraph right before their Theorem) "We consider a more general random-effects hypothesis where $\mathbb{E}[w]=0$, and $\mathrm{Var}[w] = p^{-1}\alpha^2 I_{p\times p}$".
>
> Q2: At a high level, I personally do not find counting the number of peaks in the risk curve to be that interesting of a research problem, as least in the case of minimum $\ell_2$-norm interpolation. It is rather intuitive that one can engineer the features such that the sample covariance matrix becomes ill-conditioned once in a while, and it is not clear if this specific phenomenon has wider implications in practical machine learning problems.
>
> A2: The practical importance of our results in machine learning problems is that when practitioners find that the distribution of data may have an ill-conditioned covariance matrix, our results remind them to be careful of possible performance degradation when adding more training data. Our results provide a possible and provable explanation for the samplewise non-monotonicity observed in machine learning practice.
>
> Q3: (minor) Note that the Gaussian assumption has already been relaxed to bounded moments in the Stieltjes transform-based analysis, e.g., [Dobriban and Wager] and [Wu and Xu]. Hence I'm not sure if there's any benefit of introducing CGMT to this problem (which limits the analysis to Gaussian features).
>
> A3: The benefit of introducing CGMT is that it is easier to yield a tractable closed-form formula for the two-block case (our Theorem 3) and theoretically establish the existence of sample-wise multiple descent.
>
> Q4: (minor) In Corollary 2, why is $\varrho\to \infty$ a well-defined limit? Intuitively, wouldn't this violate the condition that the eigenvalues do not depend on $d$ which goes to infinity?
>
> A4: $\varrho$ and therefore the eigenvalues do not depend on $d$.
>
> Thank you so much again for your detailed comments. If you have any further questions, please do not hesitate to comment.

---

> > ### Comment · Reviewer_6s7y · 2021-11-24
> > **Followup**
> >
> > Thanks for the detailed response, which does not address my concerns unfortunately. The authors should be more explicit and honest in acknowledging prior results.
> > I won't comment on everything in the rebuttal; just a few key points:
> >
> > 1. *Sample- vs. model-wise double (multiple) descent*.
> >
> > This is just a matter of terminology. For the studied linear model in the proportional limit, one may interpret the increase of ratio $d/n$ as larger model capacity or smaller sample size — I don't think this changes any technical content. Whether the prior works explicitly mentioned the term "sample-wise double descent" is irrelevant to the theoretical contribution.
> > For example, Figure 5 in [[Wu and Xu]](https://proceedings.nips.cc/paper/2020/file/72e6d3238361fe70f22fb0ac624a7072-Paper.pdf), Figure 3 in [Richards et al.], and Figure 15 in [Amari et al.] can all be viewed as either sample-wise or model-wise multiple descent.
> >
> > If the authors want to claim that their contribution is to rigorously prove multiple descent (instead of demonstrating it in the simulations), then it should also be acknowledged that this is currently only done for the variance term in the two-block case — this result is rather incremental (see point below).
> >
> > 2. *"We solved the Stieltjes transform explicitly and derived explicit formulae for the risk and variance."*
> >
> > Again, the closed-form expression is only derived for the variance term in the two-block case (of course the underparameterized regime is already well-understood).
> > The variance under general $\Sigma$ was first obtained in [Dobriban and Wager], and for the two-block case, the explicit expression of the Stieltjes transform has been derived in [Richards et al.] and [Amari et al.] (see point 2 in the original review) — this immediately gives the closed-form formula for the variance (e.g., see Appendix A.6 of [Richards et al.]).
> > I don't see anything new unless the authors can correct me that the two-block model they analyzed is significantly different than those in prior works. Consequently, I also don't see any new insights from the use of CGMT.
> >
> > 3. *"All of [Wu and Xu], [Richards et al.], [Amari et al.], and [Dobriban and Wager] assumed a prior on the true linear model... our risk does NOT take the expectation over a random true linear model."*
> >
> > This is incorrect. [Wu and Xu] clearly covers the case where $\beta^*$ is fixed, for which $\Sigma_\beta$ is a rank-1 matrix.
> > Therefore, my impression is that both the bias and variance derived in this paper are already covered by [Wu and Xu] (which does not require Gaussian data). Furthermore, as mentioned in my original review, the result on risk monotonicity (Theorem 4 in this submission) also largely overlaps with Proposition 7 in [Wu and Xu] — this is still not acknowledged in the revision.
> >
> > I hope the authors could point out where I misunderstood their response.
> > Otherwise my evaluation remains unchanged: this submission is below the acceptance bar, and I might consider further lowering my score.

---

### Official Review · Reviewer_Xsqg · 2021-11-02

**Correctness:** 4
**Technical Novelty And Significance:** 2
**Empirical Novelty And Significance:** Not applicable
**Recommendation:** 6
**Confidence:** 2

**Main Review:**

Strengths of the paper:
1.	The paper tackles an important problem very relevant to the ICLR community.
2.	It provides a deep theoretical analysis of a specific setting and the proofs appear non-trivial. The main contributions of the paper are novel.


Weaknesses of the paper:
Overall the paper is reasonably well-written but the writing can improve in certain aspects. Some comments and questions below.
1.	It is not apparent to the reader why the authors choose an asymptotic regime to focus on. My understanding is that the primary reason is easier theoretical tractability. It would help the reader to know why the paper focuses on the asymptotic setting.
2.	It is unclear in the write-up if sample-wise descent occurs only in the over-parameterized regime or not. Pointing this explicitly in the place where you list your contributions would help. More broadly, it is important to have a discussion around these regimes in the main body and also a discussion around how they are defined in the asymptotic regime would help.
3.	The paper is written in a very technical manner with very little proof intuition provided in the main body. It would benefit from having more intuition on the tools used and the reasons the main theorems hold.
4.	Given that prior work already theoretically shows that sample-wise multiple descent can occur in linear regression, the main contribution of the paper appears to be the result that optimal regularization can remove double descent even in certain anisotropic settings. If this is not the case, the paper should do a better job of highlighting the novelty of their result in relation to prior results.

I am not too familiar with the particular techniques and tools used in the paper and could not verify the claims but they seem correct.

**Summary Of The Paper:**

The paper builds on a very recent line of work studying the “multiple descent” phenomenon.
Multiple descent is known to occur along many axes: (i) number of parameters, (ii) train time, (iii) amount of train data. The current paper focuses on studying the theoretical rates for (iii) in the asymptotic regime where the number of samples and the dimensionality of data tend to infinity at similar rates.  This covers both under and over-parameterized settings.
In this limiting regime, the main results of the paper are:
1.	Computing formulae for the excess risk of the least squares estimator and the ridge regression estimator in a specific setting.
2.	Using 1, they show that if the covariance matrix of data is ill-conditioned, then multiple descent can happen.
3.	They also show that under an assumption on the scaling of the projections of the true parameter to the eigen-spaces of the eigenvalues of the covariance matrix, an optimal amount of regularization ridge regression estimator avoids multiple descent even in anisotropic settings which is novel.



**Summary Of The Review:**

The problem of multiple descent is an important scientific question which is relevant to the ICLR community. The contributions of the current paper are novel. The techniques used appear novel as well. However the degree of significance of the results is not fully clear. The paper would benefit from a deeper discussion around this.

---

> ### Author Response · Authors · 2021-11-23
> **Response to Reviewer Xsqg**
>
> Thank you so much for your time, insightful feedback, positive evaluation :-) We have incorporated the changes into the revised paper, and address the issues in detail here:
>
> Q1: It is not apparent to the reader why the authors choose an asymptotic regime to focus on. My understanding is that the primary reason is easier theoretical tractability. It would help the reader to know why the paper focuses on the asymptotic setting.
>
> A1: Yes, the primary reason is easier theoretical tractability. In fact, the asymptotic regime is an approximation for large $n,d$ and can also shed light on practical machine learning problems. We clarified this in the revision.
>
> Q2: It is unclear in the write-up if sample-wise descent occurs only in the over-parameterized regime or not. Pointing this explicitly in the place where you list your contributions would help.
>
> A2: We showed in Theorem 4 that the samplewise descent occurs across the under- and over-parameterized regimes when we use the optimally-regularized estimator. Without regularization, there will be a blow-up in expected excess risk when $n=d$ (the linear model exactly interpolates the data) and therefore, there is no samplewise descent across the under- and over-parameterized regimes. As per the reviewer's suggestion, we pointed out this explicitly in the revised Section 1.2.
>
> Q3: More broadly, it is important to have a discussion around these regimes in the main body and also a discussion around how they are defined in the asymptotic regime would help.
>
> A3: Thank you for the suggestion. In the revision, we added a discussion around these regimes in the main body and a discussion around how they are defined in the asymptotic regime.
>
> Q4: The paper is written in a very technical manner with very little proof intuition provided in the main body. It would benefit from having more intuition on the tools used and the reasons the main theorems hold.
>
> A4: In the revision, we added more intuition on the high-level ideas of the proofs at the beginning of Section 3.1.
>
> Q5: Given that prior work already theoretically shows that sample-wise multiple descent can occur in linear regression, the main contribution of the paper appears to be the result that optimal regularization can remove double descent even in certain anisotropic settings. If this is not the case, the paper should do a better job of highlighting the novelty of their result in relation to prior results.
>
> A5: To the best of our knowledge, prior work proved parameter-wise (test error vs. model complexity) multiple descent theoretically, but not sample-wise (test error vs. the number of training samples) multiple descent. It would be great if the reviewer would point us to the related references.

---

> > ### Comment · Reviewer_Xsqg · 2021-11-30
> > **Thanks for the response to my questions**
> >
> > Thank you for responding to my questions and for incorporating the suggestions in the revised draft. The reference I had in mind showing sample-wise multiple descent is this one: https://arxiv.org/abs/1912.07242. I realize now that it only shows double descent and not multiple descent.
> >
> > Overall, the paper's contributions as pointed out by reviewer 6s7y seem incremental given prior work. I will retain my score.

---

### Official Review · Reviewer_7ii8 · 2021-11-05

**Correctness:** 4
**Technical Novelty And Significance:** 3
**Empirical Novelty And Significance:** 3
**Recommendation:** 6
**Confidence:** 2

**Main Review:**

Strengths
* The paper is well written
* Technical contributions are sound and somewhat interesting

Weaknesses
* Some assumptions are unclear if they are mild or why they are assumed in the first place. For instance, assuming fixed eigenvalues looks like a rather strange assumption at first glance, and authors do not provide any intuition why this is assumed and how restrictive it is. To me, that assumption is an artifact for the proof techniques, which undermines the contributions of the paper.


**Summary Of The Paper:**

Authors study the generalization risk of ridge and ridgeless linear regression. It is assumed that the data follows a multivariate normal distribution and there is also an assumption on the spectrum of the covariance matrix. Among the contributions, the authors show a formula for the limiting bias and variance, show that sample-wise multiple descent happens when the covariance matrix is highly ill-conditioned, and study optimal regularization.

**Summary Of The Review:**

After my first read of the paper, I am recommending a weak accept. My concerns are more about the assumptions and the lack of discussions about them. I would appreciate it if the authors can comment on this matter.

---

> ### Author Response · Authors · 2021-11-23
> **Response to Reviewer 7ii8**
>
> Thank you so much for your time, insightful feedback, positive evaluation :-) We have incorporated the changes into the revised paper, and address the issues in detail here:
>
> Q1: Some assumptions are unclear if they are mild or why they are assumed in the first place. For instance, assuming fixed eigenvalues looks like a rather strange assumption at first glance, and authors do not provide any intuition why this is assumed and how restrictive it is. To me, that assumption is an artifact for the proof techniques, which undermines the contributions of the paper.
>
> A1: Sorry for the confusion. First, in practice, there are natural random variables that satisfy our assumption. For example, assume that we want to use machine A to measure the length of several objects and use machine B to measure their temperature. The measured lengths and temperatures follow i.i.d. Gaussian distribution. However, the variance of measurement of machine A is different from that of machine B. Then we consider the random vector formed by the measurements (length of object 1, length of object 2, ..., length of object $n$, temperature of object 1, temperature of object 2, ..., temperature of object $n$). This results in a block-structured covariance matrix. When we measure more objects, the size of the covariance matrix tends to infinity. Second, the motivation came from (Nakkiran et al., 2021). Nakkiran et al. (2021) observed  empirically in their Figure 2 that when the covariance matrix has a block structure (specifically, there are only two fixed different eigenvalues $10$ and $1$), the expected excess risk exhibits multiple descent. We quantitatively studied this observation and obtained the related formulae. We stated this in the third item in Section 1.2. We further clarified in the revision.

---

### Official Review · Reviewer_RwFH · 2021-11-08

**Correctness:** 4
**Technical Novelty And Significance:** 3
**Empirical Novelty And Significance:** 3
**Recommendation:** 8
**Confidence:** 4

**Main Review:**

The paper is well-written and technically sound as far as I can tell. I enjoy reading Theorem 1, which uses two different approaches to derive the same limits in a way that can mutually corroborate each other, and believe this closed-form result will have future impact on other topics. The theoretical result is further supported by clean experiments up to high precision. I believe it is a mathematically strong paper that is among top quality in this venue.

However, I have reservations regarding the motivation, at least it is not nicely explained.
1. As the title suggests, the authors are working on sample-wise multiple descents. They made this clear in the INTRODUCTION by citing (Nakkiran et al.,2021;Chen et al., 2020b; Min et al., 2021). However, I believe the authors are working on something else by their setup in section 1.1 'Asymptotic Regime' as $n,d\to\infty$. In words, not only the sample size goes to infinity but also the feature size goes to infinity in proportion. First of all, this is not sample-wise descents in prior works, where $d$ is fixed! A better description of this work is on sample-feature-wise multiple descents. Second, this seems to limit the application as in most tasks the number of features is fixed. Whether the theoretical results in this work shed any light on real cases needs at least empirical evidence.
2. Can the authors give some motivation on why to assume $\|\Pi_i\theta^*\|_2→\eta_i$. I understand its use in theory but what does this assumption mean in words?

Other minor points: the paper seems very sloppy, e.g. Page 4 'Bias-Variance Decomposition of Expected Excess Risk', the first line of equations misses a square; in CONCLUSION section, the first sentence should have ridge regression besides ridgeless; in the same section, "risk curve.Moreover"->"risk curve. Moreover"; there are multiple places where the lines are out of margin, e.g. Equation (8); the appendices are not well-referenced, e.g. after reading Theorem 1, I tried to look for the proofs but there is no hyperlink that should link to Appendix C. Same thing happens to other results wherever appendices should be clearly referenced.

**Summary Of The Paper:**

This paper studies the generalization risk of ridge and ridgeless linear regression under Gaussian data and Gaussian noise settings. The limiting bias and variance are described in closed-form. The analysis leverages random matrix theory and CGMT. The authors show that sample-wise multiple descent phenomenon can take place or be avoided if optimal regularization is applied.

**Summary Of The Review:**

This paper is technically very strong and well-supported by experiments. I believe it is among top 20% of accepted ICLR papers. However, the motivation of the problem is less clear which requires non-trivial improvement.

---

> ### Author Response · Authors · 2021-11-23
> **Response to Reviewer RwFH**
>
> Thank you so much for your time, insightful feedback, positive evaluation :-) We have incorporated the changes into the revised paper, and address the issues in detail here:
>
> Q1: As the title suggests, the authors are working on sample-wise multiple descents. They made this clear in the INTRODUCTION by citing (Nakkiran et al.,2021;Chen et al., 2020b; Min et al., 2021). However, I believe the authors are working on something else by their setup in section 1.1 'Asymptotic Regime' as $n,d\to \infty$. In words, not only the sample size goes to infinity but also the feature size goes to infinity in proportion. First of all, this is not sample-wise descents in prior works, where $d$ is fixed! A better description of this work is on sample-feature-wise multiple descents. Second, this seems to limit the application as in most tasks the number of features is fixed. Whether the theoretical results in this work shed any light on real cases needs at least empirical evidence.
>
> A1: We considered the asymptotic regime where $n,d\to \infty$ proportionally and studied how the expected excess risk varies with the ratio $n/d$. The asymptotic regime characterizes the situation where $d$ is very large and we vary $n$ from $1$ to a large number. Therefore, in practice, the asymptotic result is particularly useful when $d$ is large. We called it ``samplewise multiple descent'' because we vary the ratio $n/d$, in other words, the relative number of samples.
>
> Q2: Can the authors give some motivation on why to assume $\|\Pi_i \theta^*\|_2\to \eta_i $. I understand its use in theory but what does this assumption mean in words?
>
> A2: This means that the norm of the true linear model on each eigenspace of the covariance matrix tends to be a constant.

---

### Decision · Program_Chairs · 2022-01-20

**Decision:**

Reject

**Comment:**

The authors analyze linear regression with gaussian covariates in an
asymptoptic setting, where the number of examples and the number of
covariates go to infinity together.  They identify conditions
on the covariance under which "multiple descent" occurs, and
conditions under which a regularization removes this effect.

Concerns were raised that the overlap between this paper and previous
research was too substantial for it to be published in ICLR.  These
persisted after the authors' response and the discussion period.